# LATS1 controls CTCF chromatin occupancy and hormonal response of 3D-grown breast cancer cells

Julieta Ramírez-Cuéllar [ID] [1,2,8], Roberto Ferrari [ID] [1,7,8], Rosario T Sanz[3], Marta Valverde-Santiago[3], Judith García-García[3], A Silvina Nacht[1], David Castillo[4], Francois Le Dily[1], Maria Victoria Neguembor [ID] [1], Marco Malatesta [ID] [5], Sarah Bonnin[1], Marc A Marti-Renom[1,2,4,6], Miguel Beato [ID] [1,2] & Guillermo P Vicent [ID] [1,3 ✉]

## Abstract

The cancer epigenome has been studied in cells cultured in two-dimensional (2D) monolayers, but recent studies highlight the impact of the extracellular matrix and the three-dimensional (3D) environment on multiple cellular functions. Here, we report the physical, biochemical, and genomic differences between T47D breast cancer cells cultured in 2D and as 3D spheroids. Cells within 3D spheroids exhibit a rounder nucleus with less accessible, more compacted chromatin, as well as altered expression of ~2000 genes, the majority of which become repressed. Hi-C analysis reveals that cells in 3D are enriched for regions belonging to the B compartment, have decreased chromatin-bound CTCF and increased fusion of topologically associating domains (TADs). Upregulation of the Hippo pathway in 3D spheroids results in the activation of the LATS1 kinase, which promotes phosphorylation and displacement of CTCF from DNA, thereby likely causing the observed TAD fusions. 3D cells show higher chromatin binding of progesterone receptor (PR), leading to an increase in the number of hormone-regulated genes. This effect is in part mediated by LATS1 activation, which favors cytoplasmic retention of YAP and CTCF removal.

**Keywords** Three-dimensional Cell Growth; Breast Cancer; CTCF; LATS1 Kinase and Hormonal Response
**Subject Categories** Cancer; Signal Transduction

## Introduction

Over the years, two-dimensional (2D) cultures have been used to improve our understanding of cellular signaling pathways and to decode the mechanisms deregulated in many diseases, including cancer. Yet, in recent years increasing evidence showed that 2D cultures do not fully reflect the complexity of the microenvironment encountered by cells as part of tissues or organs, endorsing the use of three-dimensional (3D) culture conditions generated by the presence of more physiological extracellular milieu (Kim et al, 2020). Within tissues, cells are exposed to a complex environment, including blood-circulating molecules, neighboring cells, and the extracellular matrix (ECM). 3D cell culture systems have been experimented for decades (Barcellos-Hoff et al, 1989) and the importance of the ECM in cell behavior and gene expression is now well accepted.

The Hippo pathway is one of the several routes that have been reported to respond to the cell's environment. In mammals, the core Hippo pathway is characterized by serine/threonine kinases; mammalian Sterile 20-related 1 and 2 kinases (MST1 and MST2), and Large Tumor Suppressor 1 and 2 kinases (LATS1 and LATS2). The transcriptional co-activators Yes-associated protein (YAP) and the Transcriptional coactivator with PDZ-binding motif (TAZ) act as nuclear relays of mechanical signals exerted by ECM rigidity and cell shape (Dupont et al, 2011). It has been previously described that on soft substrates, YAP is inactivated by LATS-dependent phosphorylation on serine 127 and later tagged for degradation in the cytoplasm (Meng et al, 2016). The unphosphorylated YAP translocates into the cell nucleus followed by the regulation of gene transcription (Zanconato et al, 2016). LATS1 and LATS2 (LATS) have emerged as central regulators of cell fate, and modulate the functions of numerous oncogenic or tumor suppressive effectors, including YAP/TAZ, the Aurora mitotic kinase family and the tumor suppressor transcription factor p53 (Furth and Aylon, 2017). Except for YAP, not many substrates have been described for LATS, however, it was recently reported that LATS can phosphorylate the architectural protein CTCF within its zinc finger (ZF) linkers, reducing their affinity for DNA (Luo et al, 2020).

In breast cancer cells, steroid hormones are key regulators of cell differentiation and proliferation. The steroid hormones estrogens and progesterone acting via their specific receptors (ERα and PR, respectively) control the proliferation of breast cancer cells in a very

[1]Center for Genomic Regulation (CRG), Barcelona Institute for Science and Technology (BIST) Barcelona, Barcelona, Spain. [2]Universitat Pompeu Fabra (UPF), Barcelona, Spain. [3]Molecular Biology Institute of Barcelona, Consejo Superior de Investigaciones Científicas (IBMB-CSIC), C/ Baldiri Reixac, 4-8, 08028 Barcelona, Spain. [4]CNAG-CRG, Centre for Genomic Regulation, The Barcelona Institute of Science and Technology, Baldiri Reixac 4, Barcelona 08028, Spain. [5]Department of Chemistry, Life Sciences and Environmental Sustainability, University of Parma, Parma, Italy. [6]ICREA, Barcelona, Spain. [7]Present address: Department of Chemistry, Life Sciences and Environmental Sustainability, University of Parma, Parma, Italy. [8]These authors contributed equally: Julieta Ramirez Cuellar, Roberto Ferrari. ✉E-mail: gvmbmc@ibmb.csic.es

different way. While estrogens are inducers of cell proliferation that activate cyclin D1 (Musgrove et al, 1994; Planas-Silva et al, 1999) and decrease the expression of CDK inhibitors (Prall et al, 1997), progestins promote a single-cell cycle followed by proliferation arrest at G1/S, that correlates with a delayed activation of CDK inhibitor p21$^{WAF1}$ (Owen et al, 1998).

The activated receptors of estrogens and progesterone regulate gene expression mainly by interacting with specific DNA sequences in chromatin and recruiting chromatin remodeling complexes and transcriptional coregulators (Beato et al, 1995). In addition, a minor fraction of ERα and PR is attached to the cell membrane via palmitoylation of a conserved cysteine. Binding of the hormone to this membrane-attached receptors can activate the Src/Erk/Msk1 and CyclinA/CDK2 signaling cascades (Migliaccio et al, 1996; Vicent et al, 2006; Vicent et al, 2011), leading to PR activation via phosphorylation at S294, binding to progesterone responsive elements, and modulation of chromatin structure as prerequisite for gene regulation (Beato and Vicent, 2011). One limitation of these results is that they were obtained from breast cancer cells cultured as monolayer ignoring other physiological signaling pathways, such as the Hippo pathway. Although progress has been made toward understanding the roles of LATS in tumorigenesis, the kinase substrates or downstream target proteins mediating LATS function remain largely unknown.

In this work, we try to overcome this limitation by cultivating T47D breast cancer cells as spheroids in Matrigel (3D) and compared the results with those obtained with the same cells grown in monolayer on plastic (2D). We found that compared to 2D-grown cells, the nucleus of the 3D cells is more rounded and exhibits a larger surface, a more compact chromatin and a distinctive set of 1700 downregulated genes. Using Hi-C assays, we did not detect strong differences at the genome structural level between cells grown in 3D and those gown in 2D, but only a modest increase in the B-compartment regions and a tendency to TAD fusions. However, we did observe an increase in LATS1 kinase activity in 3D-grown cells, which triggered the displacement of the architectural protein CTCF from 75% of its genomic binding regions, including the promoters of genes regulated in 3D. Despite less accessible chromatin in 3D, hormone regulation was more pronounced compared to 2D, with more responsive genes and more extensive PR binding.

## Results

### Characterization of T47D-derived spheroids

To explore how the extracellular matrix (ECM) and cell-cell contacts impact on the structure and function of the cell nucleus, we grew breast tubular epithelial cells on Matrigel, to enable the formation of three-dimensional (3D) structures. We cultured breast cancer cell lines MCF10A, T47D, MCF-7; BT-474, and ZR-75-1 on plastic (2D) or embedded on Matrigel (3D), for 10 days, the time required to form multicellular acini (Appendix Fig. S1A). While T47D cells formed spheroids containing proliferating cells, the non-tumorigenic MCF10A cells formed round structures with a polarized acinar internal pattern (Fig. 1A,B). The other tumorigenic cell lines (MCF-7 and BT-474) cultured on Matrigel also form spheroids, as previously reported (Lee et al, 2007) (Appendix Fig. S1A). In this study, we focused on PR + T47D cells as a model

system. These cells are known for their robust responsiveness to the hormone progesterone, and their genomic structure, the PR cistrome, and the associated signaling pathway network have been extensively characterized (Ballare et al, 2013; Le Dily et al, 2014; Le Dily et al, 2019; Vicent et al, 2006; Vicent et al, 2010; Zaurin et al, 2021).

We performed a functional and phenotypic characterization of the T47D spheroids. First, we assessed the physical properties of the cell nuclei in the spheroids compared to 2D by implementing IMARIS and ImageJ tools (Martin et al, 2021). The nuclear volume of 3D cells, measured after DAPI staining, was significantly enlarged compared to the 2D cell nuclei (488.7 μm$^3$ vs. 408.5 μm$^3$, $P$ value < 0.001, Appendix Fig. S1B, left panel) while the diameter of the 3D nuclei was smaller (9.17 μm vs.15.56 μm, p value < 0.001, Appendix Fig. S1B, middle panel). This translates into an increased sphericity coefficient of the 3D nuclei (0.74 vs. 0.57 in 2D, $P$ value < 0.001, Appendix Fig. S1B, right panel) and a larger surface area (643.3 μm$^2$ vs. 381.6 μm$^2$, $P$ value < 0.001, Appendix Fig. S1B, lower panel).

We also evaluated the proliferation capacity of T47D cells cultured in 3D compared to cells grown in 2D. We found that 3D cells proliferated at a rate 2.6 times faster than its 2D counterpart (Fig. 1C, left panel). Within 10 days of culture the 3D cells formed spheroids with an average diameter of 100 μm, typically comprising 119 ± 12 cells (Fig. 1C, right panel). The increased cell proliferation detected in 3D cells was in accordance to the high levels of Ki67-positive cells detected in the spheres, also evident by using a live/dead cell imaging Kit (Appendix Fig. S2A).

Next, we probed the signaling pathways known to be regulated in cells grown in 3D conditions, such as the Hippo and the Focal Adhesion Kinase (FAK/ PTK2) pathways. In 3D-cultured breast cancer cells, YAP displayed higher level of phosphorylation on S127 compared to 2D, while total YAP levels remained unchanged (Fig. 1D, left panel). YAP was also found largely in the cytoplasm of 3D-grown cells (Fig. 1D, right panels), as previously described (Zanconato et al, 2019). In contrast, in 2D cells, the Hippo pathway was switched off and YAP mainly found into the nucleus (Fig. 1D, top right panels). When probing the FAK/PTK2 pathways, we found p-FAK severely deregulated in 3D cells, as indicated by its significant reduction in FAK-Y397 phosphorylation, compared to the levels found in 2D cells (Fig. 1E), despite no changes in total FAK levels (Fig. 1E). Therefore, in 3D-grown cells the presence of a less stiff environment is sensed, integrated and transduced leading to both repression of focal adhesion and activation of the Hippo pathway.

Increased levels of phospho-LATS (part of the Hippo pathway) in response to 3D cell growth was also observed in ER$^+$ MCF-7 cells. However, this activation was less pronounced in comparison to T47D cells (Appendix Fig. S1C compared to Fig. 5B). Notably, there was no significant increase in pYAP or decrease in p-FAK in MCF-7 cells, where YAP expression notably remained low in comparison to other mammary cell lines (Appendix Fig. S1C, lower panel). This observation indicates a partial conservation of the cytosolic activation of the Hippo pathway in response to 3D in MCF-7 cells. (Appendix Fig. S1C). In addition, it is worth mentioning that reduced ER levels in MCF-7 3D cells (Appendix Fig. S1C) had an impact on their proliferative capacity.

To investigate whether these changes in signaling, shape, surface and volume of the 3D nucleus impact on gene activity, we performed

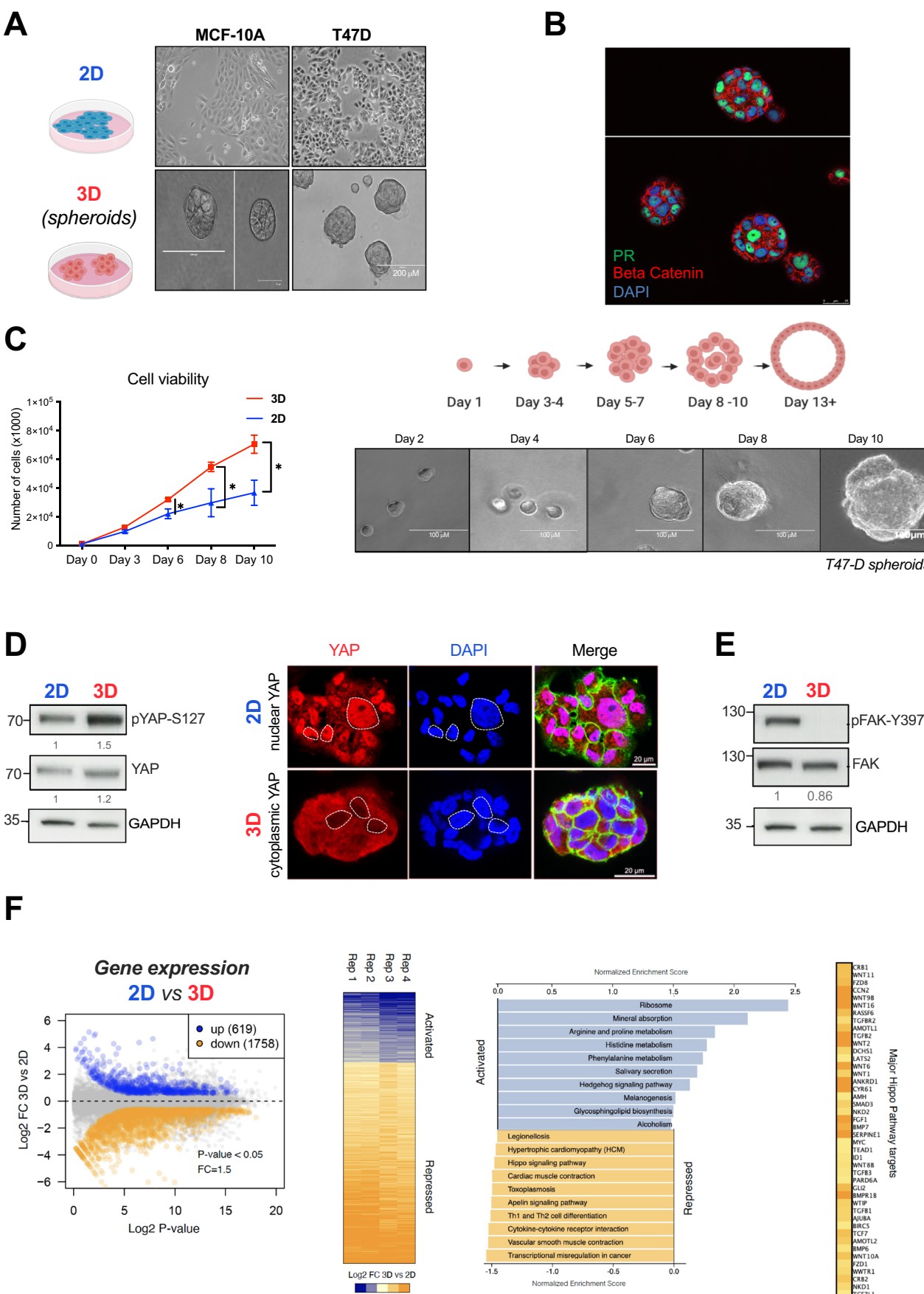

◄ **Figure 1.** **Evaluation of cell growth, signaling pathways, and differentially expressed genes in T47D cells grown in 3D and 2D conditions.**

(**A**) Comparison of MCF10A and T47D breast cell lines grown on the plastic dish 2D and as 3D culture in Matrigel. (**B**) Immunofluorescence analysis showed that PR expression is conserved in 3D T47D cells. (**C**) Cell proliferation assays performed in 2D and 3D cells (left panel). The growth of 3D T47D cells over a period of 10 days to reach an estimated size of 100 μm is depicted (right panel). (**D**) Increased levels of YAP protein phosphorylation (pYAP-S127) in 3D was assessed by western blot (left panel). Immunostaining of Hippo pathway effector YAP in 2D and 3D T47D cells. When cells are grown in 3D conditions YAP is phosphorylated and re-localized to the cytoplasm (right panel). (**E**) T47D cells in grown in a 3D downregulate phosphorylation of FAK and its downstream pathways. (**F**) RNA-seq Differential Expression Analysis (DEA), performed in 2D and 3D T47D cells using [log2FC = 2 and *P* value < 0.05], the experiment reveals over 2300 genes deregulated when cells were grown in 3D (left panel). Using the DEA results 3D/2D, enriched signaling pathways (KEGG) are shown, FDR < 0.05 (right panel). Source data are available online for this figure.

RNA-seq analysis in T47D cells grown in 2D and 3D. The transcriptional profiles of 2D and 3D samples, yielded statistically significant changes for 2377 transcripts: 619 were upregulated and 1758 were downregulated in 3D-cultured cells compared to 2D cells [log2FC = 2 and *P* value < 0.05] (Fig. 1F). Gene Ontology analysis of biological processes and cellular compartments showed that cultivating cells in 3D lead to upregulation of genes enriched in terms associated to neurogenesis, nervous system development, axon genesis and neuronal projection. While 3D downregulated genes were overrepresented in cell and focal adhesion, cell junction and cell periphery (Appendix Fig. S3A). These results suggest that cells cultured in 3D possess a unique gene expression program that recapitulates neuronal growth and, additionally, decreases the expression of genes associated to adhesion, most likely due to the presence of a less stiff environment compared to 2D. The similarities between growth as 3D spheres and the nervous system has been previously proposed (Caswell and Zech, 2018). Further analysis of KEGG-exclusive 3D pathways showed terms related to *Ribosome*, *Mineral Absorption* and *Amino acid Metabolism* for 3D upgenes (Fig. 1F, right panel). The physical characteristics of the surrounding structure play a pivotal role in coordinating this regulatory interplay. Concerning *Amino acid Metabolism*, the stiffness of the extracellular matrix regulates the activation of the Hippo pathway and, consequently, the nuclear localization of YAP/TAZ. YAP/TAZ play a critical role in controlling genes involved in amino acid synthesis, transport and metabolism (Ge et al, 2021). Therefore, in response to the mechanical properties of their environment, cells adjust amino acid metabolism to acquire the energy necessary for specific cellular functions, such as enhanced cell proliferation, as illustrated in Fig. 1C.

Extracellular matrix stiffness also influences *Mineral Absorption*, as previously documented (Derricks et al, 2015). Cells on softer substrates (4 kPa) demonstrate increased responsiveness to VEGF, while cells on stiffer substrates (125 kPa) exhibit a diminished response.

The enrichment of genes related to *Ribosomes* suggests that protein synthesis is differentially regulated in 3D environments. This may be attributed to the unique demand for specialized proteins involved in cell-cell interactions, the integration of external signals, and mechano-transduction, especially in cells growing as spheres with low stiffness. In summary, the regulation of genes associated with *Amino acid Metabolism*, *Mineral Absorption*, and *Ribosomes* in 3D environments underscores the significant influence of extracellular matrix stiffness on various cellular processes. This intricate interplay allows cells to adapt to their mechanical surroundings, ultimately affecting crucial aspects of cell biology and physiology.

The 3D-repressed genes exhibit an enrichment in genes associated with *Transcriptional deregulation in Cancer*, the *Hippo*

and *Apelin pathways*, among others. The Apelin pathway is not well-documented in breast cancer, with only one report associating high levels of Apelin with postmenopausal breast cancer (Salman et al, 2016). However, genes involved in vasodilation and *Muscle Contraction* are significantly repressed, which could be explained as an adaptation to the softer 3D environment. The suppression of these pathways could also be linked to the upregulation of *Mineral Absorption* pathway, as calcium ions (Ca2+) can regulate the contractility of vascular smooth muscle cells (Brozovich et al, 2016).

Next, we asked whether the changes in gene expression that we observed in 3D-grown cells (Fig. 1F) were due to changes in histone acetylation. To this end, cells grown in 2D and 3D were subjected to ChIP-seq of H3K27ac and H3K18ac, two marks strongly associated to gene expression (Ferrari et al, 2012; Wang et al, 2008). In general, a notable variation, ranging from 20% to 44%, was observed in the two marks when cells transitioned from 2D to 3D culture conditions (Appendix Fig. S4A,B). These alterations were particularly concentrated in genes influenced by the growth environment. Consequently, a substantial reduction in H3K27ac was detected for genes downregulated and upregulated in 3D (Appendix Fig. S4C), along with an increase in H3K18ac for genes upregulated in 3D (Appendix Fig. S4D, right panel), as compared to their 2D counterparts.

RNA polymerase 2 (pol2) which globally did not show drastic changes (Appendix Fig. S4E) was significantly reduced at the TSS and gene body of 3D downregulated genes (Appendix Fig. S4F, left panel) and increased in 3D upregulated genes (Fig. S4F, right panel) as visualized at the *HSPA8* and *CXCL12* loci, two representative genes of each category (Appendix Fig. S4G).

We then tested whether the levels of repressive marks such as H3K27me3 and H3K9me3 are affected on genes that change their expression with the culture condition. In 3D downregulated genes an increase in H3K27me3 was detected (Fig. EV1A) while H3K9me3 levels remained unchanged (Fig. EV1B). Surprisingly, a significant increase in H3K9me3 was found in 3D upgenes (Fig. EV1B).

Given that Polycomb operates through a digital mechanism with two opposing expression states signifying its presence or absence on target genes (Menon et al, 2021), assigning the modest increase in H3K27me3 detected in 3D downgenes directly to its final impact on transcription is challenging. Instead, it is likely a complex interplay of various histone modifications, along with other transcriptional and architectural factors, that contributes to this phenomenon.

We also evaluated whether the change in cell culture condition affects the presence and distribution of super-enhancers (SEs) (Hnisz et al, 2013). To date, 214 genomic regions were identified as SEs in T47D breast cancer cells (Jiang et al, 2019). The overlap

**A**

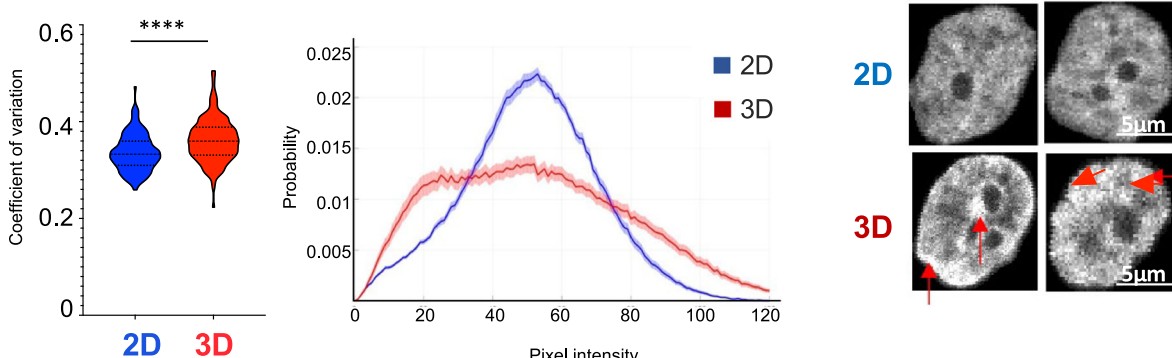

**B**

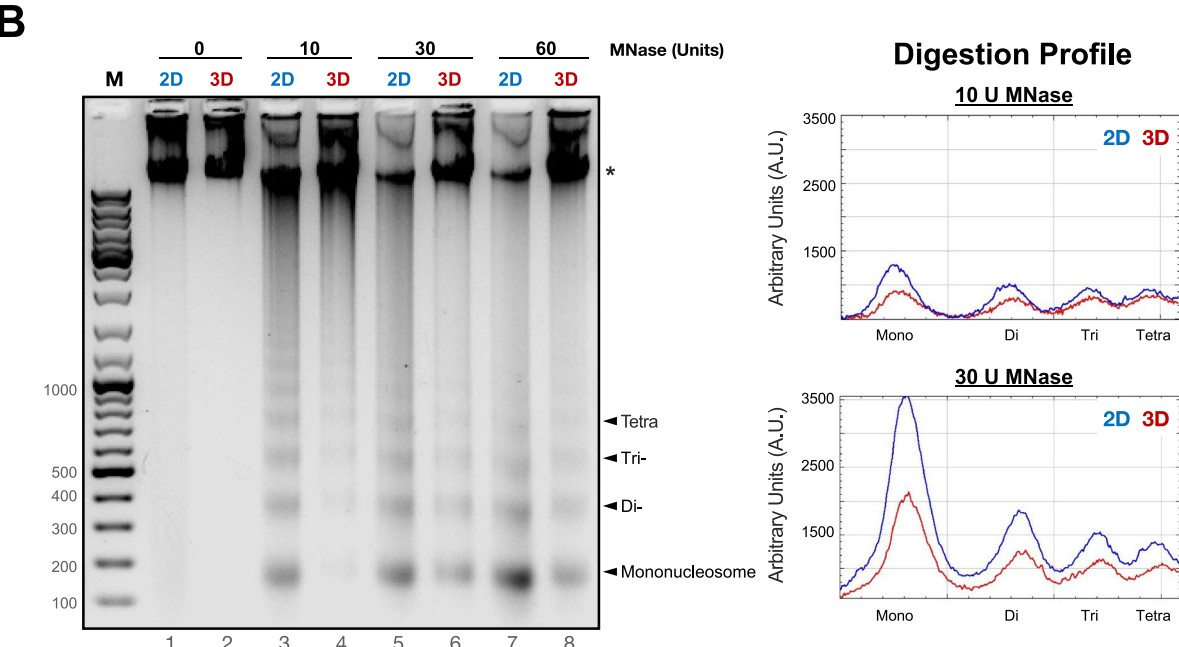

**C**

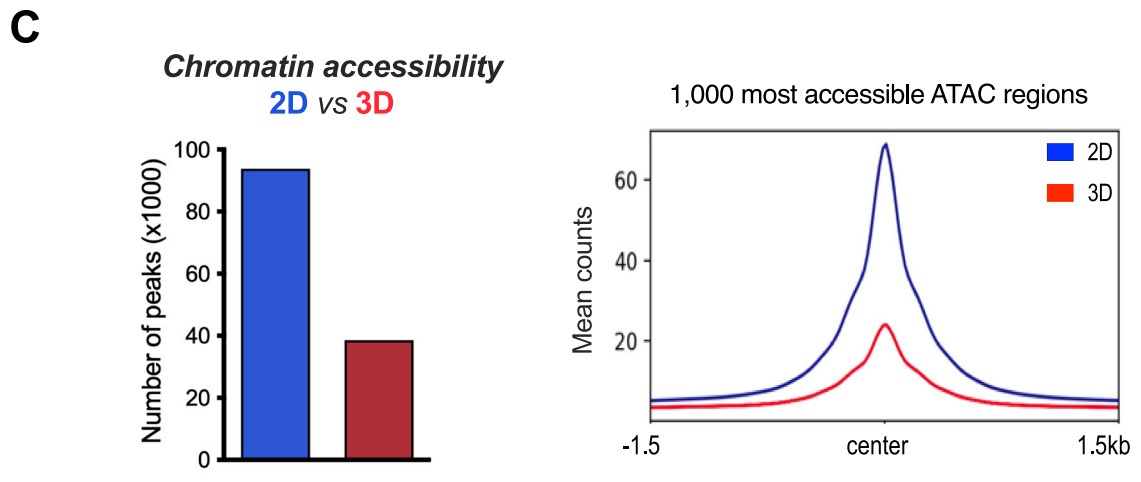

**Figure 2. Chromatin organization and accessibility in 2D and 3D-grown cells.**

(A) Coefficient of variation of DAPI staining distribution between T47D cells grown in 2D and in 3D, $n = 100$ cells per condition. Violin plot (left panel). Probability distribution of the pixel intensities obtained in 2D and 3D (middle panel). T47D nuclei stained with DAPI exhibiting a clearer pattern of condensed chromatin clusters in 3D-grown cells is shown (right panel, red arrows indicate highly compacted regions). Scale bar: 5 μm. (B) T47D cells grown in 2D and 3D conditions were isolated, nuclei were prepared and digested with different concentrations of MNase. Digestion products were resolved in 1.2% agarose gel electrophoresis (left panel) and the intensity of the bands was quantified by ImageJ software (right panel). (C) Number of peaks per sample of ATAC-seq experiments (up left panel). The 1000 peaks with the highest score for 2D and 3D condition were plotted (bottom left panel). The overlap between ATAC-seq replicates is shown (right panel). Source data are available online for this figure.

between the super-enhancers detected in 2D and 3D according to the H3K27ac signal (579 and 564, respectively) was close to 80% (Fig. EV1C). This observation would support that cells fundamentally maintain their regulatory gene network in both of these environments (Fig. EV1D). Nonetheless, it is essential to acknowledge that 20% of super-enhancers are exclusively activated in the 3D environment. The question of whether these variances have implications for the alteration of cell identity remains open.

To address this point, by using a proximity-based script (Hnisz et al, 2013), we found 129 and 95 genes associated to 2D and 3D SEs, respectively (Fig. EV2A). The 2D-exclusive genes are related to terms like abnormal embryo size, abnormal development, and hematopoiesis, which are closely linked to the Hippo pathway (as reported in (Yu et al, 2015)) (Fig. EV2C). Conversely, in the case of genes exclusively regulated in the 3D condition by SEs, many of them appear to be artifacts, as illustrated with the HSPA8 gene (Fig. EV2B). This hampers the identification of significant associated terms.

When attempting to establish a connection between super-enhancers and alterations in genome structure, such as the A to B transition, it becomes evident that the quantity of SEs is reduced, and this transformation exhibits subtler changes, predominantly shifting from A to less degree of A, or toward emerging B, and vice versa (Fig. EV2D). Snapshots from the genome browser illustrating these transitions for two 3D downregulated genes, SLC26A7 and RUNX1T1, and two 3D upregulated genes, PI15 and ITGAL are shown (Fig. EV2D).

## Impact of 3D growth on nuclear structure

To explore how 3D culture impacts on nuclear structure, we measured the distribution of the chromatin throughout the nucleus by determining the coefficient of variation of DNA stained with DAPI (Martin et al, 2021). We found that the chromatin is organized differently, presenting a more heterogenous pattern in 3D-grown cells, with an increased coefficient of variation compared to cells grown in 2D (Fig. 2A, left panel), a broader distribution of DAPI signal intensity and the appearance of highly dense foci in 3D-grown cells (Fig. 2A, middle and right panels, respectively).

In order to assess chromatin accessibility within the cell nucleus of 2D and 3D-grown cells, we conducted a comprehensive approach that includes: (i) MNase experiments, which detect the compaction of chromatin fibers at a large scale, potentially indicating changes in the presence of linker histones and heterochromatin, and (ii) ATAC-seq experiments, which measure focal nucleosome depletion at sites where sequence-specific transcription factors, such as CTCF could bind, as well as the nucleosome-depleted region at transcription start sites (TSS).

To estimate the accessibility of the chromatin at large scale, we performed micrococcal nuclease (MNase) digestion assays. Isolated nuclei obtained from cells grown in 2D and 3D were treated with increasing concentrations of MNase, and the products of digestion were analyzed by gel electrophoresis (Fig. 2B). We found that the accessibility to MNase was compromised in 3D compared to its 2D counterpart at the range of MNase concentrations tested. (Fig. 2B). In addition, the amount of undigested DNA present in 3D was higher compared to 2D, even at 60U MNase (Fig. 2B, left panel). These results support that chromatin is less accessible in nuclei from 3D growing cells.

To assess whether the chromatin accessibility changed at nucleosome level with the growth conditions, we performed an Assay for Transposase-accessible chromatin followed by sequencing (ATAC-seq), which is a technique used to assess genome-wide chromatin accessibility (Buenrostro et al, 2013).

Nuclei obtained from cells grown in 2D and 3D conditions were incubated with 2.5 U of the tagmentase Tn5. The average number of ATAC peaks was significantly higher in 2D than in 3D growing cells (93,850 vs. 38,542 peaks for 2D and 3D, respectively) (Fig. 2C), pointing to a general decrease of accessibility in nuclei from 3D-grown cells. When comparing the 1000 most accessible ATAC regions detected in both 2D and 3D-grown cells, we observed that 3D peaks contained less reads/peak (Fig. 2C, right panel). Most of the peaks detected in 3D-grown cells also appeared in 2D (96.88%; 37,342). Only 547 new ATAC peaks were detected in 3D-grown cells, while 24,242 sites were lost when comparing all replicates. Hence, the chromatin structure in 3D-grown T47D cells exhibited a noticeably less accessible conformation, both at the level of larger fibers and at nucleosome resolution, in contrast to the 2D cells. These findings align with the results from gene expression analysis and imaging studies (Figs. 1F and 2A–C).

To characterize the regions that become more accessible in 2D and 3D we conducted a more in-depth analysis of the ATAC-seq data (Fig. EV3A,B). Our results suggest that the sites that become more accessible in 2D and 3D are enriched in the TF motifs of CTCF, TEAD, FOXA1, and PR motifs (Fig. EV3B). However, when we combined our analysis with ChIP-seq data (Zheng et al, 2019), word cloud plots indicated that open sites in 3D are enriched in the estrogen pathway through the ER itself, alongside pioneer factors FOXA1 and GATA3; the cofactor GREB1 and the coactivator P300 (Fig. EV4B,C, lower panels), which could explain the increased proliferation observed in 3D cells (Fig. 1C). In contrast, the sites that become closed in 3D (2D-exclusive) are enriched in CTCF, an architectural factor that we have shown to be displaced when we grow the cells as spheroids (Fig. 4A). This would imply that in 3D cells a more estrogenic program is turned on along with CTCF being displaced, if these two events are connected require further investigation.

**A**

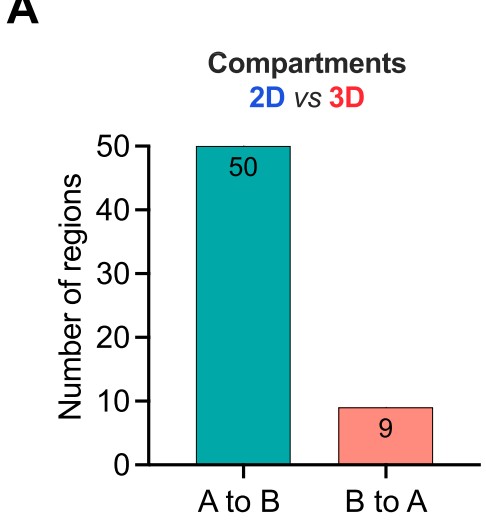

**B**

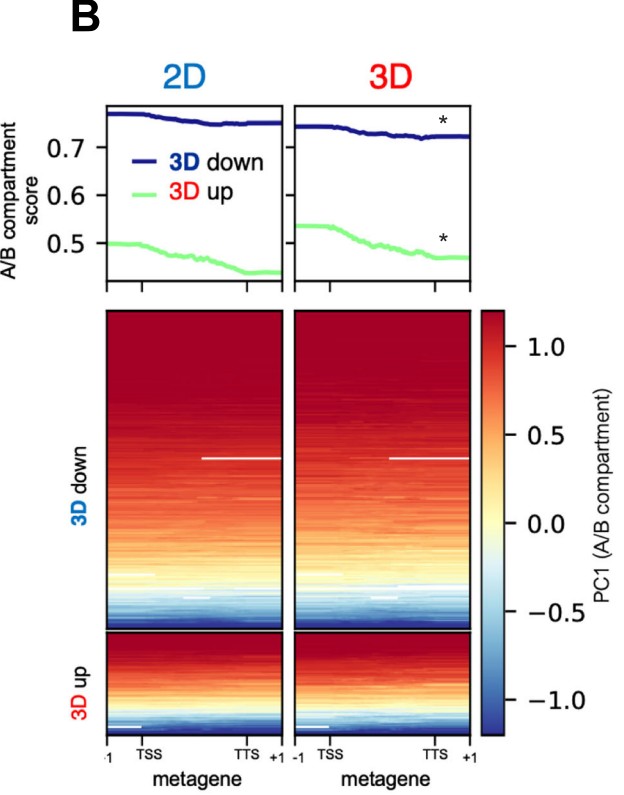

**C**

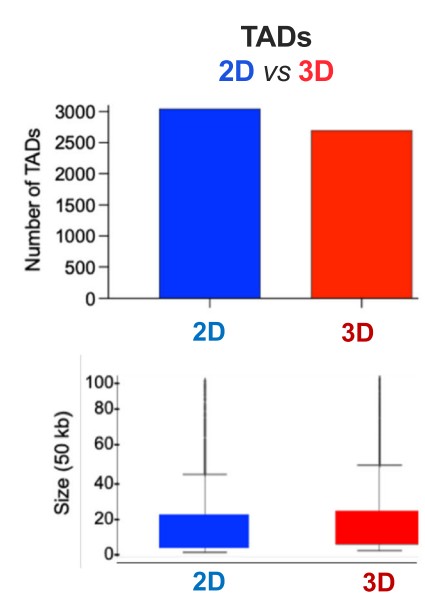

**E**

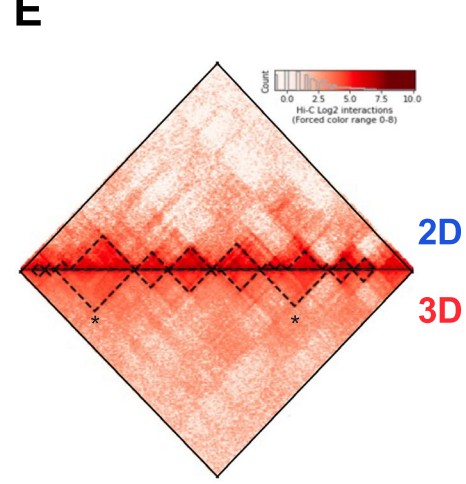

Region: chrX, resolution: 25kb

**D**

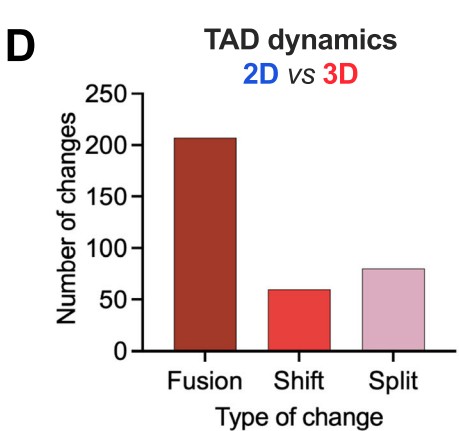

**Figure 3. Genome topology in 2D and 3D-grown cells.**

(A) Total number of compartments changing from 2D to 3D cells (bin = 50 kb). (B) Metagene analysis of A/B compartments versus gene expression. Metagene profiling of A/B compartments for 3D down and 3D upgenes is shown both as average profile (top) and heatmap (bottom). Both sets of genes show significant changes in compartment score (* two-tailed *t* test *P* values < 0.001). (C) Changes in total number (top) and size of TADs (bottom) between 2D and 3D-grown cells. (D) Type of TAD changes detected in 3D-grown cells. (E) A Hi-C heatmap of a region located in the X chromosome illustrating TADs fusions in 3D.

Furthermore, the chromatin accessibility of 2D and 3D-exclusive genes was also assayed, and we detected changes. As expected, 2D-upgenes exhibit greater accessibility within the gene body in 2D condition compared to their 3D counterpart, whereas in 3D condition the expected inverse trend is not observed (Fig. EV3D). However, we found increased accessibility in distantly located regulatory regions (enhancers) associated to 166 genes upregulated in 3D. To illustrate this finding, the TN2Cx and PTPRD genes are shown (Fig. EV3E).

## Effect of 3D growth on genome architecture

Next, we addressed the overall 3D genome organization performing Hi-C experiments with cells grown in 2D and 3D conditions.

First, we explored the chromosome compartments by analyzing the correlation between the eigenvectors obtained from the interaction of Hi-C matrices in 2D and 3D-cultured cells at 1 Mb resolution. This analysis showed no substantial changes in the genome structure at this resolution ($R = 0.99$). We next evaluated the genome compartmentalization at 100 kb resolution and found a limited number of changes between cells grown in 2D and in 3D. We identified 50 regions that changed from A (active) to B (inactive) compartment in 3D-grown cells, while only nine regions changed from B to A compartment (Fig. 3A). In fact, when the A/B-compartment transitions were evaluated by calculating an A/B score using Hi-C homer (Heinz et al, 2018), we observed a significant increase of 3D upgenes in A compartments and conversely, decrease in A and increase in B compartments for downregulated genes in 3D (Fig. 3B). These findings suggest that in the 3D condition, the changes detected in gene expression sustain gradual transitions between the A and B compartments.

Finally, we assessed the behavior of topologically associating domains (TADs). As depicted in Fig. 3C, in 3D-grown cells there is a decrease in total number of TADs (3041 and 2860 in 2D and 3D, respectively) accompanied by a small increase in their size. This is due to the prevalence of TAD fusions (200) over splitting (60) or shifting (50) (Fig. 3D). A Hi-C heatmap plot at chromosome X depicting two TAD fusions detected in 3D-grown cells is shown (Fig. 3E).

Since the discovery of TADs, it became clear that the boundary regions separating topological domains are enriched in CTCF along with the structural maintenance of chromosomes (SMC) cohesin complex, housekeeping genes, and SINE elements (Barutcu et al, 2018; Dixon et al, 2012; Nora et al, 2012).

CTCF is a highly conserved zinc finger protein and is best known as a transcription factor. It can function as a transcriptional activator, repressor or as an insulator protein, blocking the communication between enhancers and promoters (Kim and Wirtz, 2015). Thus, targeted degradation of CTCF can affect either enhancer-promoter looping or local insulation, promoting TAD fusions. To map the genome distribution of CTCF in both 2D and 3D conditions, we performed ChIP-seq experiments. Even though CTCF protein levels remain unchanged in 2D and 3D growing conditions (Fig. 4A, bottom right panel), we found a 75% loss of CTCF binding in 3D cells (Fig. 4A, top right panel).

To rule out that the antibody against CTCF might not be properly recognizing CTCF in 3D, we perform chromatin fractionation assays in 2D and 3D-grown cells. Our results showed that about half of the CTCF is bound to 3D chromatin fraction compared to the 2D condition (Appendix Fig. S5A).

This led us to hypothesize that the loss of CTCF binding could be responsible for the loss of TAD borders resulting in their fusion detected in 3D-grown cells (Fig. 3E).

We found regions where TAD fusions and CTCF displacement overlap in 3D (Fig. 4B). However, the loss of CTCF binding was global and distributed throughout the entire genome, impacting on all TADs, irrespective of whether they change or not in 3D (Appendix Fig. S5B–D). Similar loss of CTCF was found at fused, shifted or split TADs (Fig. 4C). In fact, we identified several regions where the loss of CTCF did not have impact on the TAD structure at all (Appendix Fig. S5E). Around 95% of the TAD borders were conserved in 2D and 3D conditions.

To assess whether changes in the 3D TADs could be assigned to differential loading of cohesin components in these regions, we conducted RAD21 ChIP-seq experiments under 2D and 3D growth conditions. Our findings indicate that there are no significant alterations in RAD21 occupancy within both fused and random TADs (Appendix Fig. S6).

Interestingly, HOMER motif analysis of the ATAC-seq peaks showed that the motif for CTCF tops the rank of enrichment in sites where accessibility is reduced in 3D (Appendix Fig. S5F). It is important to note that even though BORIS (CTCFL) is ranked second, this protein is not present in breast cancer cells. Several reports show that displacement of CTCF leads to a decrease in ATAC-seq signal and gene regulation (Franke et al, 2021; Xu et al, 2021). This would suggest that in 3D, CTCF displacement results in a more compact chromatin at those sites and could affect gene activity.

## CTCF displacement and gene regulation in T47D 3D cells

Although the displacement of CTCF observed in the chromatin of 3D-grown T47D cells does not always lead to changes in TADs boundaries examined by Hi-C, we wanted to investigate whether the absence of CTCF is associated to changes in gene regulation.

Combined analyses of RNA-seq and ChIP-seq data showed that 3D up and downregulated genes showed a significant depletion in CTCF around the TSS compared to random genes (Fig. 4D). Therefore, the genes activated or repressed as a result of growing cells in 3D, present less CTCF around the transcription start site, which would imply a reduction of long-distance interactions ("looping") between regulatory regions (enhancers, silencers) and

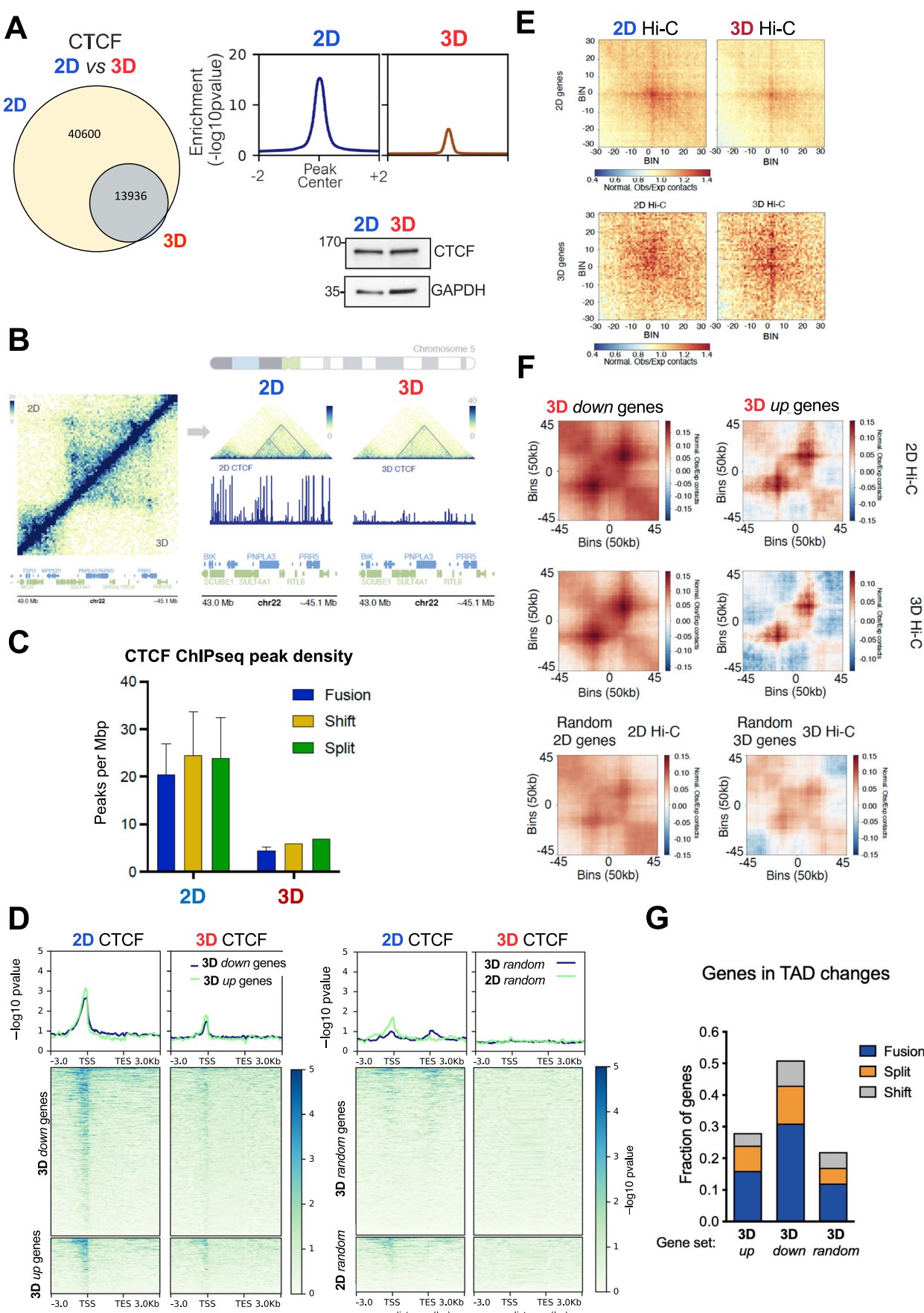

◄

**Figure 4. The architectural protein CTCF is displaced from target chromatin in 3D-grown cells.**

(A) Cells grown in 2D and 3D were lysed and total extracts were analyzed by western blot using CTCF and GAPDH-specific antibodies (left panel). ChIP-seq of CTCF performed in 2D and 3D conditions showed differential CTCF binding regions genome-wide (middle and right panels). (B) Snapshot of a region presenting a 3D-exclusive TAD fusion event that overlaps with the loss of CTCF binding. (C) The density of CTCF peaks in 3D-exclusive fused, split and shifted TADs is shown. (D) The presence of CTCF in 3D up- and downgenes (left panels) and in random genes (right panels) are shown. (E) Hi-C explorer aggregate plots. Long-distance interactions among 3D down- and upregulated genes grown in 3D and 2D conditions. The genomic coordinates of the 3D down- and upregulated genes are centered between half the number of bins and the other half the number of bins. Plotted are the submatrices of the aggregated contact frequency for 20 bins (1.5 kb bin size, 35 kb in total) in both upstream and downstream directions. Color bar scale with increasing red shades of color stands for higher contact frequency. (F) Aggregate FAN-C plots depict a region that is three times the size of the TAD located in its center. TADs are selected on the basis of containing 3D up- or downgenes (upper panels) or random genes (lower panels). High signal is located in the center, especially at the TAD corner, where the corner loops are typically located. (G) CTCF displacement impacts in 3D-regulated genes, by changing their contact environment through TADs fusion. The percentages of up, down or random genes in 3D condition that fall into fused, split or shifted TADs are shown. Source data are available online for this figure.

their target genes. In fact, this loss in interactions detected in 3D resulted into a decrease of Hi-C contacts, particularly evident in 3D downregulated genes (P value <0.001) (Fig. 4E and S6G).

As mentioned above, CTCF is also enriched at TADs borders. We asked whether long range CTCF-interactions and therefore TADs structure as a whole, could be particularly affecting those TADs specifically enriched in 3D deregulated genes. To address this point, we used FAN-C aggregate plots using the TADs containing genes differentially expressed in 3D-grown cells and measure the levels of CTCF by ChIP-seq. Our results showed that for genes downregulated in 3D, the global interactions between two CTCF sites that formed a TAD corner were drastically reduced (P value <0.001) (Fig. 4F, left panels), whereas for genes activated in 3D cells we could detect a slight decrease, but not statistically significant (P value >0.001) (Fig. 4F, right panels). No difference was found for a random set of genes and TADs as a control (Fig. 4F, bottom panels). Therefore, CTCF displacement could be involved in the prevailing 3D-gene repression. This implies changes at the level of intra-TAD interactions as well as at the level of TAD borders, decreasing its ability to isolate and regulate genes.

We then assessed whether the observed differences in CTCF and TAD structure associated to 3D genes (Fig. 4E,F) could lead to any preferential change of TADs in 3D condition. We found that 17% of up and 30% of downgenes in 3D are included in 3D fused TADs, while the percentages were significantly lower in split and shift TADs (7 and 13% for split and 3.5 and 7% for shifted TADs, P <0.001) as well as in all categories for random genes (Fig. 4G). Thus, CTCF displacement particularly impacts in 3D genes, by changing their contact environment through TADs fusion as shown for the 3D down gene *DUSP1* whose expression is associated with an increased risk of metastasis and shorter overall survival in breast cancer (Candas et al, 2014) (Appendix Fig. S7A–C).

## LATS activity and CTCF binding to chromatin in T47D grown in 3D cells

In 3D condition, we have shown that the Hippo pathway is activated, resulting in the phosphorylation of YAP at serine 127 by the LATS kinase (Fig. 1D).

A recent study reported that LATS can phosphorylate CTCF at T347 and S402 and disables its DNA-binding activity (Luo et al, 2020). In that system, loss of CTCF binding was able to disrupt local chromatin domains and downregulate genes located in the neighborhood (Luo et al, 2020). Therefore, we tested whether the displacement of CTCF found in 3D-grown cells could be due to increased LATS-dependent phosphorylation of CTCF.

As no commercial phos-CTCF antibody is available, we performed a Phos-tag gel, that would allow detection of phos-CTCF as slow-migrating bands depending on their state of phosphorylation. When extracts from 2D and 3D-grown cells were analyzed in Phos-tag gels, we detected a retarded band in 3D, which was not present in 2D extracts (Fig. 5A). To confirm that this band corresponded to a phosphorylated version of CTCF, we prepared 3D extracts by using a buffer lacking phosphatase inhibitors. Under this condition, the CTCF band ran faster in Phos-tag gel as a nonphosphorylated version of CTCF (compare lanes 2 and 3 in Fig. 5A), supporting that phosphorylated-CTCF is preferentially found in extracts from 3D-grown cells.

The direct impact of LATS on CTCF phosphorylation was tested using TRULI, a potent ATP-competitive inhibitor of LATS kinases (Kastan et al, 2021). Incubation for 24 h with 10 μM TRULI decreased the phosphorylated (activated) version of YAP at S127 (Fig. 5B, left panel).

To have a formal proof that LATS1 phosphorylates CTCF in 3D cells, we performed immunoprecipitation of CTCF in 3D-grown cells treated or not with TRULI and then probed with specific antibodies recognizing the phosphorylated RXXpS/T residues that match the changes observed in T374 and S402 as previously described (Luo et al, 2020). Our results showed that the phospho-CTCF signal in 3D is reduced by 40% in the presence of TRULI, confirming the raised hypothesis (Fig. EV4A).

To confirm that T374 and S402 of CTCF are the phosphorylation targets of LATS, and to assess their impact on chromatin binding in our experimental system, we transfected T47D cells with both wild-type and the phospho-mimetic T374E/S402E CTCF variant. Subsequently, we evaluated their chromatin binding abilities. The T374E/S402E mutant displayed a diminished capacity to bind to chromatin when compared to the wild-type CTCF (Fig. EV4B). In addition, we carried out chromatin fractionation and ChIP assays in T47D and HEK293 cells, treated or not with the LATS1 activator Latrunculin B, and transfected with the wild-type CTCF and the phospho-mutant. While the wild-type CTCF was displaced in the presence of LatB, the mutant remained bound regardless of LATS1 activation. However, these results should be taken with caution as these residues could also participate in the stability of CTCF, which makes the scenario more complex and challenges for definitive interpretations.

It has been reported that the overexpression of CTCF impedes the proliferation and metastasis of breast cancer cells by deactivating the nuclear factor-kappaB pathway in breast cancer cells (Wu et al, 2017). To elucidate the function T474 and S402, we conducted a functional assay by evaluating cell growth in T47D cells

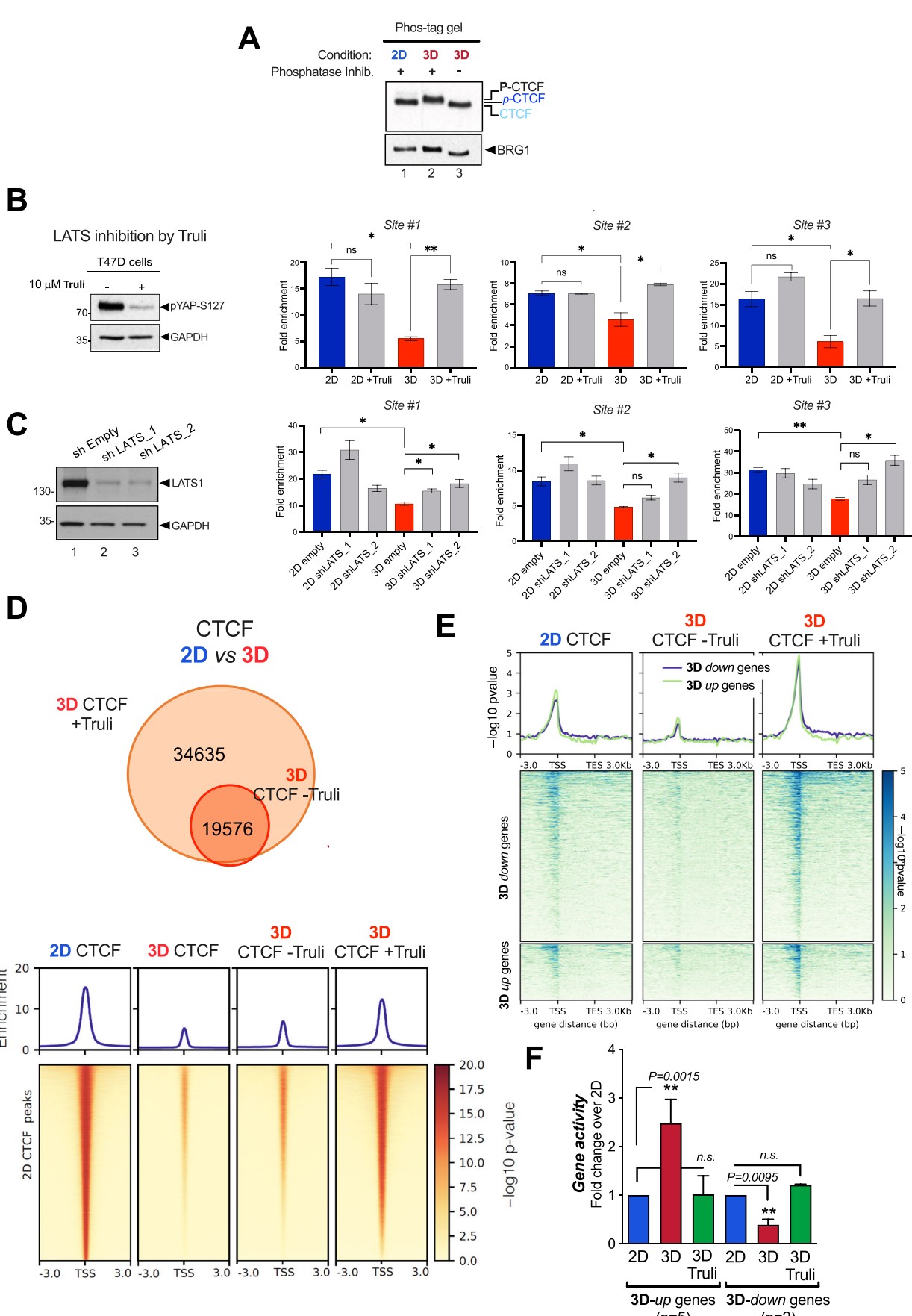

**Figure 5. Inhibition or depletion of the Hippo pathway LATS kinase compromises the loss of CTCF detected in 3D-grown cells.**

(A) Cells grown in 2D and 3D were lysed and total extracts were analyzed by Phos-tag gel which identifies putative phosphorylated bands of CTCF exclusively in 3D. (B) Cells grown in 2D or 3D conditions and treated or not with the LATS inhibitor TRULI were subjected to ChIP-qPCR of CTCF in three regions where CTCF binding is lost in cells grown in 3D ($n = 2$, $P$ value < 0.05) (right panel). T47D cells treated or not with TRULI were lysed and total extracts were analyzed by western blot using pYAP-S127 and GAPDH-specific antibodies (left panel). (C) Effect of LATS1 depletion on CTCF binding. Left panel: quantification of LATS1 by western blot. Right panel: ChIP-qPCR analysis of CTCF in extracts derived from control cells transfected with shEmpty and two distinct shLATS. (D) Cells grown in 3D conditions treated or not with the LATS inhibitor TRULI were subjected to CTCF ChIP-seq. Venn diagrams showed the overlapping in CTCF binding. (E) Cells grown in 3D conditions and treated or not with TRULI were subjected to CTCF ChIP-seq. The enrichment of CTCF in 3D up- and downgenes as well as in random genes is shown. The heatmaps, illustrating CTCF ChIP-seq results under both 2D and 3D untreated conditions, are the same as those previously presented in Fig. 4D, left panels. (F) Cells grown in 2D and 3D conditions and treated or not with TRULI as indicated, were submitted to gene activity assays. Several up and downregulated genes in 3D were tested. Results are represented as mean and SD from three experiments performed in duplicate. The $P$ value was calculated using the Student's $t$ test. Source data are available online for this figure.

expressing both wild-type (WT) and a phospho-mimetic variant of CTCF T374E/S402E (Fig. EV4C). The overexpression of WT CTCF in T47D cells inhibited cell growth compared to the control transfection (Fig. EV5C). Conversely, in the T374E/S402E mutant in which chromatin binding is compromised (see Fig. EV4B), this effect was not appreciated (compare WT vs. mutant in Fig. EV5C). This suggests that LATS1-dependent phosphorylation of CTCF induced its displacement and participates in 3D cell growth, as demonstrated in Fig. 1C.

Therefore, we performed qPCR-ChIP of CTCF in 2D and 3D-grown cells in the absence or in the presence of TRULI. As expected, a decrease in CTCF binding was detected in 3D compared to 2D in the three different genomic regions tested (Fig. 5B, right panels). In the presence of TRULI, CTCF binding was significantly recovered (Fig. 5B, right panels).

To support these findings and to discard indirect effects of TRULI, we performed similar ChIP experiments in cells deficient of LATS1 (Fig. 5C, left panel). As with TRULI, knockdown of LATS1 rescues CTCF binding in 3D cells (Fig. 5C, right panels). Interestingly, these results point to LATS1 as responsible for the observed effects.

The potential involvement of LATS2 was assessed using a specific antibody that recognizes phosphorylation at T1079 and T1041 present in both LATS1 and LATS2, in wild-type cells and LATS1-depleted cells. The dual phosphorylation signal in 3D cells decreased proportionally with the reduction of LATS1 (Fig. 5C), supporting that LATS1 is the Hippo kinase involved in our system (Fig. EV4D).

To extend our conclusions, we performed ChIP-seq of CTCF in 3D cells in the presence and absence of TRULI. We found that binding of CTCF was recovered across the genome in the presence of the LATS inhibitor (Fig. 5D). In fact, in this condition, we observed that CTCF binding was preferentially recovered at 3D-dependent genes (Fig. 5E).

The sensitivity of CTCF ChIP-seq experiments is notably influenced by the particular protocol employed. This results in varying outcomes regarding the detected 3D shift. Notably, using a stringent ChIP protocol, which includes multiple washes and the use of DNA purification columns (ChIP-IT, Active Motif), revealed a 75% displacement of CTCF when cells were cultured in a 3D environment, as depicted in Fig. 4A.

In contrast, using a standard protocol with gentler wash steps, as outlined by Vicent et al, 2014 (Vicent et al, 2014), and incorporating an internal Drosophila spike-in control, reduced this displacement to 40% (Appendix Fig. S8A–C). Notably, this reduction remained statistically significant when compared to cells

cultured in a 2D monolayer setting. Moreover, in spike-in-controlled CTCF ChIP-seq experiments conducted in the presence of the LATS inhibitor TRULI, CTCF binding was restored (refer to Appendix Fig. S8C). These findings align with our earlier results; the average of all three CTCF ChIP-seq replicates obtained from 2D and 3D cells is depicted in Fig. S8D.

We next asked whether the activity of the LATS kinase and its effect on CTCF binding was required for the proper regulation of 3D genes. We tested 5 genes up and 2 genes down in 3D and we found that TRULI significantly changed their activity, approaching to that detected in 2D (Fig. 5F).

Moreover, we also evaluate the effect of TRULI on 3D growth. In the presence of TRULI, 3D cells grew less and the spheres were smaller compared to untreated 3D cells (Appendix Fig. S9A,B). In agreement with the gene activity, cell growth in the presence of TRULI turned out to be similar to that observed in 2D, coinciding with the presence of CTCF bound to the target genes and the activity exhibited in both conditions (Fig. 5E,F; Appendix Fig. S9).

## Hormone-dependent gene regulation in 2D and 3D T47D cells

The exposure to progestins produces multiple effects in breast cancer cells which are mediated by the activation of the Progesterone Receptor (PR). The hormone-activated PR translocates to the nucleus where it actively regulates gene transcription. To explore the response to progestins of T47D cells grown in 3D, we measured the level of expression of different transcription factors known to be involved in PR activation in cells cultured in 2D conditions. The levels of PR, ERα and FOXA1 remained similar between 2D and 3D-grown cells (Appendix Fig. S10A). A slight increase in activated PR phosphorylated at S294 (pPRS294) levels was detected in 3D compared to 2D in the absence and in the presence of 10 nM R5020 (Appendix Fig. S10A,B). However, no differences either in the extent of pPRS294 signal induced by hormone or in the percentage of cells responding to the hormonal stimulus were found (Appendix Fig. S10B). The pPRS294 signal increased with hormone and its expression was detected homogeneously distributed throughout the cells of the spheroid (Appendix Fig. S10C, upper panels), similar to the pattern observed for total PR (Appendix Fig. S10D).

To globally evaluate the hormone-regulated genes in the spheroids and compare them to the 2D model, we performed RNA-seq experiments. Briefly, T47D cells grown in 2D or 3D were exposed to solvent or to 10 nM R5020 for 6 h, followed by RNA extraction, mRNA library preparation, and massive sequencing.

Almost twice as many genes were regulated by hormone in 3D-grown cells compared to 2D cells (7654 vs 4681, log2FC > 1, adj *P* value < 0.01). Upregulated genes increased by around 35%, while downregulated genes increased by 100% (Fig. 6A). Most of the genes regulated in 3D were not regulated in 2D (4749 vs 2961 genes, respectively) (Fig. 6B).

Interestingly, among the GO categories regulated by hormone exclusively in 3D we found terms such as *cytoskeleton, actin filament organization, cell growth*, and *cell–cell junction* (Appendix Fig. S11A,B). These terms were also overrepresented when the extracellular matrix was the only variable incorporated into the analysis (Appendix Fig. S3). Thus, when exposed to hormone, T47D breast cancer cells grown as spheroids responded differently to those grown as monolayers.

### PR binding in 2D and 3D-grown T47D cells

To explore whether the environmental-dependent changes detected in gene expression are associated with changes on PR binding to the genome, we carried out ChIP-seq experiments of PR in 2D and 3D-grown cells exposed to 10 nM R5020.

In line with the increased number of genes regulated in 3D-grown cells, we also found more PR-binding sites (PRbs) in cells grown as spheroids. Almost all PRbs found in 2D were also present in 3D, but 10,226 new PRbs were exclusively found in 3D-grown cells (Fig. 6C). The genomic distribution of total PRbs in 3D turned out to be very similar to 2D (Appendix Fig. S12A). Compared to 2D-exclusive, the distribution slightly changed in the 3D-exclusive PRbs, with less PRbs found in introns and an increased number in promoter, UTR and exons (Appendix Fig. S12A).

The majority of the PRbs in 2D-grown cells are localized far from the target genes, in enhancers (Ballare et al, 2013). To map the hormone-dependent active enhancers we overlapped 3D-exclusive PRbs (10,226 regions) with H3K27ac peaks obtained from a ChIP-seq performed in 3D-grown cells. We measured the distances between these peaks to the nearest significantly regulated gene in 3D-grown cells or to random genes set. The 3D-exclusive genes were significantly closer to 3D PRbs enriched in H3K27ac (enhancers), compared to random genes (Fig. 6D). Thus, in 3D-grown cells a program is implemented aiming at the regulation of a distinct group of genes that involves the specific binding of PR to 3D-exclusive active enhancers.

### Activation of the Hippo pathway in 3D-grown cells impacts in hormone-dependent PR binding

As YAP is a transcriptional coactivator that lacks DNA-binding activity, it could act on gene expression via interaction with other DNA-binding factors. Reported YAP binding partners include TEAD, p73, Runx2, and the ErbB4 cytoplasmic domain (Li et al, 2010). However, only TEAD has been demonstrated to be important for the growth-promoting function of YAP (Zhao et al, 2008). Interestingly, the 3D-exclusive PR-binding sites turned out to be enriched in the TEAD DNA-binding motif beyond the classical HRE DNA-binding motif (Appendix Fig. S12C). As in 3D-grown cells the Hippo pathway is activated and YAP is phosphorylated in S127 by LATS1 and tagged for degradation in the cytoplasm (Fig. 1D), we hypothesize that decreased levels of nuclear YAP would increase the proportion of free TEAD sites,

thus, accounting for the enhanced PR binding detected upon hormone in 3D cells.

To address this point, we performed ChIP-seq of PR in siControl and siYAP T47D cells treated with hormone in 2D conditions. We found that 32.7% of PR-binding sites that appeared only when YAP is depleted in 2D (10,168 regions) overlap with 3D-exclusive PR-binding sites (3381 regions, *P* value < 2.2e$^{-16}$) (Fig. 6E). These new sites could regulate a new set of genes associated to a "3D spheroid" condition. Interestingly, PR binding increased globally in the absence of YAP compared to control cells (Fig. 6F).

According to this model, YAP and PR would compete for binding to 3D-exclusive chromatin regions. As YAP does not bind to DNA directly, but rather via TEAD, the role of TEAD1, TEAD4 and TAZ in this proposed mechanism should be further elucidated.

Recently in our lab, a classification of PR-binding sites has been established according to their accessibility at various progestin concentrations. The lowest progestin concentration that allows detection of ligand-dependent PRbs was 50 pM. At this physiological concentration, 2848 PRbs, termed "Highly Accessible PR binding sites", (HAs) were identified (Zaurin et al, 2021) (Fig. EV5A).

As HAs constitute essential regulatory elements in hormone-dependent growth of breast cancer cells, we decided to evaluate whether the Hippo pathway also impacted on their function. First, we tested whether HAs-dependent genes required LATS1 activity in 3D-grown cells. Our results showed that hormone-dependent regulation of *EGFR, STAT5A, TIPARP, PGR, BCAS1*, and *IGFBP5* was compromised in the presence of TRULI (Fig. EV5B). Thus, LATS1 impacts in hormonal response affecting a significant proportion of 3D-exclusive and HAs-associated genes. Unveiling the molecular mechanism by which LATS participates in hormone-dependent gene regulation requires further research.

Therefore, the global impact of the Hippo pathway -through p-LATS activation- in 3D cells could be explained at least in two ways: (i) at basal conditions, via CTCF phosphorylation inducing its displacement and (ii) via YAP phosphorylation and inactivation, releasing "hidden" TEAD sites for PR binding allowing regulation of 3D unique genes (Appendix Fig. S13). In fact, silencing YAP in 2D partially recapitulate the pattern of PR binding in 3D conditions.

## Discussion

By culturing tubular epithelial breast cancer cells as spheroids (3D), we aimed to explore novel cell signaling pathways, gene networks, transcription factors, chromatin remodelers that might have been overlooked in previous experiments performed in cells cultured as monolayer in plastic dishes (2D). The work reported here on the hormone-responsive T47D cell line is a first step in the description and characterization of a more appropriate and physiological in vitro model for breast cancer that should include various epithelial cell types, the supporting fibroblasts and blood cell environment.

Morphologically, T47D cells grown as spheroids in matrigel presented an increase in nuclear volume coupled with a reduction in nuclear diameter (Appendix Fig. S1B). These seemingly conflicting alterations might be attributed to the underlying physicochemical modifications that dictate the irregular shapes of

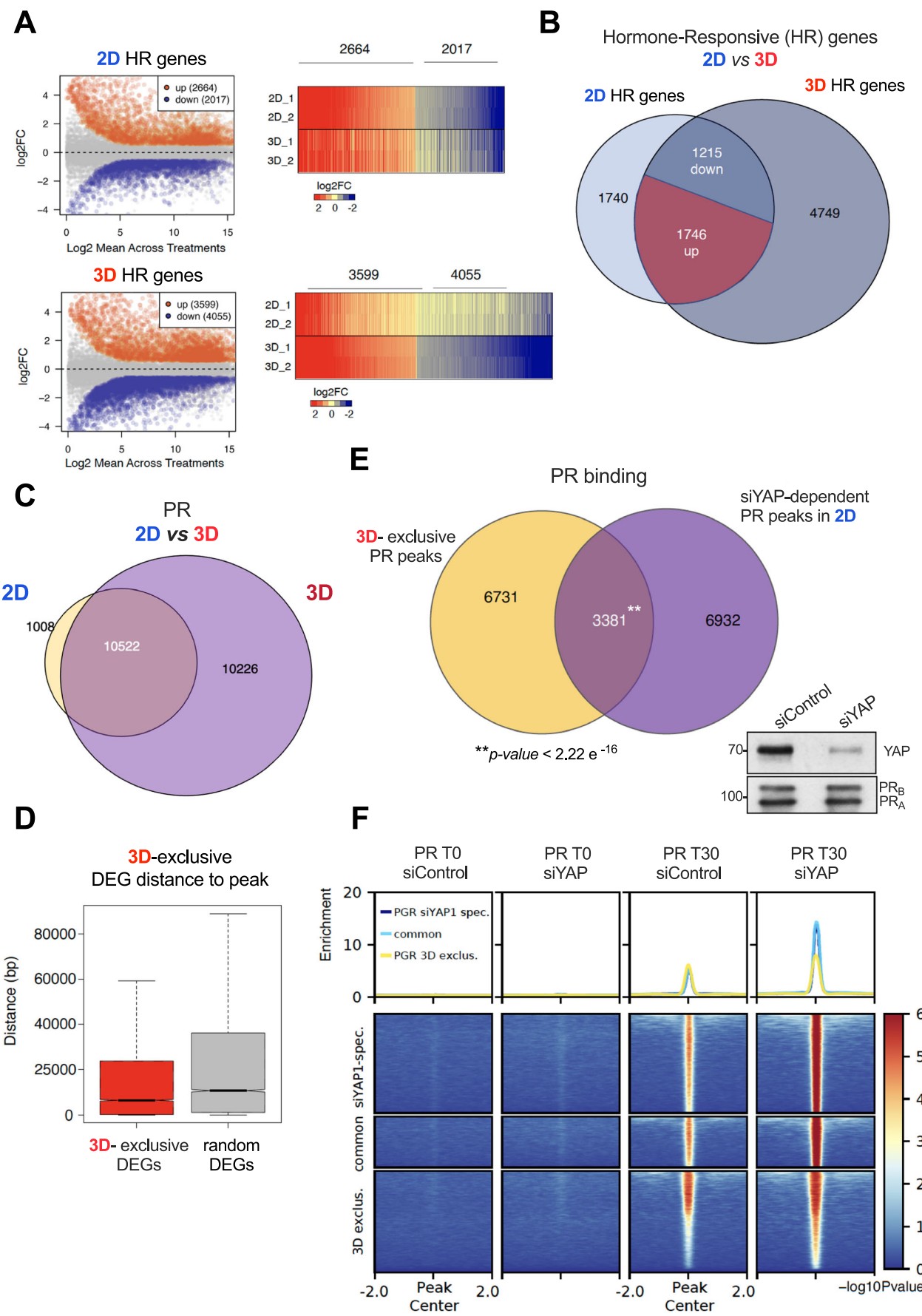

◀

**Figure 6.  3D-grown cells showed an increased response to the hormone.**

(A) T47D cells grown in 2D and 3D conditions in the presence or absence of 10 nM R5020 during 6 h were subjected to RNA-seq assays (log2FC > 1, P adj <0.01). The number of genes regulated by hormone in each condition is depicted (left panel). The volcano plot of the distribution of genes regulated in both conditions is shown (right panel). (B) Venn diagrams of up- and downregulated genes detected in 2D and 3D cells are shown. (C) T47D cells grown in 2D and 3D conditions in the presence or absence of 10 nM R5020 during 30 min were subjected to PR ChIP-seq. (D) Distance between enhancer-associated PRbs (those which overlapped with H3K27ac ChIP-seq signal) to the most proximal differentially expressed gene (3D-exclusive). The distance to random genes is used as control. (E) Venn diagram depicting the overlap of PR-binding sites from ChIP-seq experiments performed in siYAP cells treated with hormone and the subset of 3D-exclusive PR peaks. Thus, a total of 3381 PR-binding sites of 3D-exclusive PR peaks may be a result of YAP displacement detected in the 3D nucleus (P value < 2.2e-16). (F) Heatmap plots of the PR ChIP-seq experiments performed in siControl and siYAP T47D cells. Source data are available online for this figure.

the cell nucleus. These processes encompass mechanical forces acting through microtubules, actin filaments, and the osmotic pressure within the cytoplasm (Kim et al, 2015).

It has been widely reported that modifying the stiffness and composition of the used matrix can have an impact on cell growth, cell cycle, differentiation, and the activation of specific signaling pathways, such as the Hippo pathway (Garreta et al, 2019; Uroz et al, 2018). It is known that exposure of cells to a stiffer environment such as plastic implies a force transmission through focal adhesions leading to nuclear flattening and stretching of nuclear pores, reducing their mechanical resistance to molecular transport, increasing YAP nuclear import (Elosegui-Artola et al, 2017), and indirectly affecting gene expression. Conversely, on soft substrates, the nucleus is mechanically uncoupled from the matrix and not submitted to strong forces, inducing a balance between nuclear import and export of YAP through the nuclear pores (Elosegui-Artola et al, 2017).

Beyond this mechanical connection between the shape of the nucleus and YAP distribution, we found that 3D cells presented an increased LATS kinase activity which preferentially phosphorylates YAP, reducing its presence in the 3D nucleus (Figs. 1D and 5B).

## Impact of 3D growth on chromatin structure and gene regulation

The differential distribution of DAPI nuclear staining in 2D and 3D-grown cells pointed to an increased and more structured heterochromatin in 3D-grown cells. MNase and ATAC-cleavage experiments provided confirmation that chromatin in 3D-cultured cells is characterized by increased compaction, affecting both the larger chromatin fibers and the localized regions where transcription factors interact with nucleosomes. This heightened compaction results in reduced accessibility (Fig. 2B,C). In line with these results, we found an increased number of downregulated genes in 3D-grown cells (Fig. 1F) and terms related to neurogenesis, lamellipodia, and axon guidance are enriched, suggesting higher mobility to migrate in matrigel. In 3D-grown cells, we detected protrusions emerging from the cell membrane (pseudopodia) and filopodia extending out from lamellipodia, both clear indicators of cell–matrix interactions. These protrusions are associated with cellular sensing mechanisms, involving cell adhesion and cytoskeleton organization strategies (Caswell and Zech, 2018), as supported by the GO terms of the regulated genes.

As T47D cells exhibit an upregulation of genes related to neurogenesis (Appendix Fig. S3A), we sought to determine whether this phenomenon was exclusive to breast cancer cells. To address this, we conducted a comparative analysis using non-tumorigenic MCF10A cells cultured as monolayers and in 3D spheres (Maguire et al, 2016).

When we examined the genes upregulated in the 3D environment in both cell lines and categorized them by tissue-associated genes, we consistently found an enrichment of brain-associated genes, followed by those associated with the lung, mammary gland, and blood (see Appendix Fig. S14A). This suggests that culture conditions have a significant influence on gene expression patterns, rather than being solely dependent on the tumor type itself.

However, upon closer examination of the expression of upregulated genes in the 3D environment and their clustering, distinctive differences emerged. Notably, in cluster 2, we identified an exclusive overexpression of the term "nervous system development" in T47D cells compared to MCF10A cells (see Appendix Fig. S14B). Additionally, we observed significant variations in the expression of genes related to cell and focal adhesion, particularly in cluster 3, which turned out to be distinct between tumor and non-tumoral cells cultured in the 3D environment (see Appendix Fig. S14B). Therefore, the distinct expression of neurogenesis-related genes in a 3D environment appears to be associated with a more cancerous behavior. However, further investigations involving a broader range of both tumor and normal cells are required to confirm this observation.

Our results suggest that 3D-gene repression program is driven by a reduction of H3K27ac signal and a concomitant increase in its trimethylation, while gene activation is led by an increase in H3K18ac, surprisingly, accompanied by H3K9me3 (Appendix Fig. S4; EV1). In this regard, it has been previously reported that H3K9me3 can be found in promoters of repressed genes, as well as at some active genes (Barski et al, 2007).

In addition, T47D cells grown in 3D exhibit a higher overlap of the nuclear lamina with H3K9me3 and H3K27me3 heterochromatin marks compared to 2D-grown cells. A tempting hypothesis is that the presence of a more consistent heterochromatin at the Lamina Associated Domains (LADs) might be responsible for the increased gene repression detected in 3D condition. This would imply a 3D-dependent repositioning of genes close to the nuclear lamina, as previously shown in other systems (Reddy et al, 2008). However, more research is required to find the mechanistic and functional basis that support this hypothesis.

## The hippo kinase LATS promotes CTCF displacement from chromatin in 3D-grown cells

Despite the changes in chromatin compaction detected by microscopy and nuclease accessibility (Fig. 2B,C), when the topology of the genome was assessed through Hi-C experiments, no significant differences in the contact matrices were found at 1 Mb resolution (Fig. 3A). However, at higher resolution we detected changes between cells grown in 2D and in 3D at the level

of compartments and TAD structure. We found subtle differences indicative of changes in gene repression (more transitions from A to B compartment) (Fig. 3B,C). These results along with the identification of the active SEs (Fig. EV1D) suggest that the cell identity is maintained in cells grown in 3D (Flyamer et al, 2017; Stadhouders et al, 2019; Vilarrasa-Blasi et al, 2021).

However, we detected moderate changes in genome structure. Compared to cells grown on plastic, cells grown in 3D cells exhibit a higher number of TAD fusions over splits or shifts. Even though the levels of architectural CTCF protein are similar in 2D and 3D-grown cells, we detected by ChIP-seq less CTCF bound to chromatin in 3D-grown cells (Fig. 4A). A possible explanation of this finding could be associated to the increased Hippo pathway activity detected in 3D-grown cells, which is translated into a more active p-LATS1 (Fig. 5B).

It has been reported that activated p-LATS can phosphorylate CTCF resulting in reduced binding to a subset of genomic binding sites (Luo et al, 2020). In our system CTCF is phosphorylated in 3D-grown cells and inhibition or depletion of LATS restores CTCF binding to target regions. Therefore, the Hippo pathway is turned on in 3D-grown cells impacting on the cell nucleus through the reduction of CTCF binding, and thus promoting TAD fusions and changes in gene regulation of the neighboring genes. As the dissociation of CTCF we observed in 3D is more global -and not limited to fused TADs-, than previously reported (Luo et al, 2020), the role of p-LATS would be critical and CTCF binding would be more dependent on the Hippo pathway in 3D-grown cells. In fact, inhibition of p-LATS by TRULI compromised 3D cell proliferation and reduced the size of the spheroids (Appendix Fig. S9A,B).

It is worth to mention, that given the high molecular weight of the CTCF protein (140 KDa) along with the few LATS-dependent phosphorylation sites found (Luo et al, 2020), the effect of TRULI on CTCF phosphorylation could not be detected in Phos-tag gels, even though different conditions were assayed.

It has been reported in breast tissue that LATS activity modulates estrogen receptor (ERα) activity (Lit et al, 2013). More recently, LATS1/2 kinases have been shown to restrict the activity of ERα by binding and promoting its degradation (Britschgi et al, 2017). These studies implicate a nuclear function of LATS kinases in cell lineage commitment and in the malignant progression of breast and prostate cancers (Britschgi et al, 2017; Powzaniuk et al, 2004).

Although limited at the level of genome structure, the effect of CTCF depletion detected in 3D had consequences on the activity of 3D-specific genes. Thus, the role of CTCF as a general transcription factor involved in enhancer-promoter looping rather than its function as insulator at TAD borders, would be more relevant in 3D culture cells.

Regarding the uncoupling between displacement of CTCF detected in 3D-grown cells and changes in genome architecture, previous work carried out in different systems has shown that CTCF may not be as essential in establishing genome structure/TADs (Barutcu et al, 2018), at least in the mammalian genome, but this is still a matter of debate. In fact, collectively, our findings suggest that the 70% of CTCF displacement detected in 3D chromatin is either not sufficient to change the arrangement of TADs or other factors may be involved and minimize the effect of such substantial depletion. As we previously mentioned, the association of housekeeping genes with boundary regions extends previous studies in yeast and insects (Duan et al, 2010; Ulianov et al, 2016) and suggests that non-CTCF factors may also be involved in insulator/barrier functions in mammalian cells. Although it is a debatable topic, it has been recently reported that very strong loss of CTCF is required (>99%) to detect changes in TADs (Cummings and Rowley, 2022). In fact, additional experiments carried out on compatible systems are needed in order to reach more precise conclusions.

## A more sensitive response to hormone is achieved in 3D cells

Cells grown as spheroids are surrounded by other cells, interact differentially with the ECM and receive nutrients and growth factors in a very heterogeneous manner.

However, in response to hormone, the number of hormone-regulated genes is increased twofold in 3D-grown cells compared to 2D, particularly in repressed genes (2017 vs 4055 downregulated, and 2664 vs 3599 upregulated, in 2D and 3D, respectively). In cells grown in 3D, progestins regulate a group of breast cancer-associated genes that are also regulated in 2D including *CDH10, PGR, CHEK2, LSP1, TERT, SDPR*, but also a new set of 3D-exclusive genes, associated to the ECM including members of the integrin family, Laminins, and *AKT3*.

The response to progestins is more sensitive in 3D-grown cells than in cells grown in 2D, as more genes are regulated by hormone. Moreover, the increase in the number of hormone-regulated genes in 3D was accompanied by an increase in the number of PR-binding sites detected by ChIP-seq and many of the 3D-exclusive genes are closer to 3D-exclusive PR peaks (Fig. 6D). Most of these PR-binding sites correspond to enhancers, as determined by H3K27 acetylation.

Moreover, LATS1 activity turned out to be essential for high accessible PR-binding sites (HAs) (Zaurin et al, 2021) function, as the presence of TRULI compromised hormone-dependent response of HA-associated genes (Fig. EV5B). How LATS1 participates in hormonal gene regulation requires further investigation.

Our data support that in 3D-grown cells the chromatin is organized differently compared to monolayer cultured 2D cells. DAPI staining, MNase digestion and ATAC-seq assays confirm that the genome is less accessible in 3D (Fig. 2), which seems somewhat paradoxical to the increased response to hormone (Fig. 6). We found that 3D-activated signals such as the Hippo pathway LATS kinase, impact on the cell nucleus in at least two ways: the LATS kinase phosphorylates both CTCF and YAP promoting the displacement of the first and cytoplasmic retention of the latter. The absence of these two proteins in the 3D nucleus determines the activity of a subset of genes specifically regulated in 3D and in turn, enhances and assures the proper response to hormone (Appendix Fig. S13).

In summary, mechanical and chemical signals in 3D-grown cells "protect" the nucleus by making it more compact, restricting accessibility to certain regions in the genome and facilitating access to others, thus creating a more sensitive platform for the response to external hormonal cues.

## 3D spheroids: a system that recapitulates the physiological environment of the tumor cell

Cancer cells grown in 3D culture systems exhibit physiologically relevant cell-cell and cell–matrix interactions, gene expression and signaling pathway profiles, heterogeneity and structural complexity

that reflect *in vivo* tumors (Nath and Devi, 2016). Actually, the non-cellular components of the tumor microenvironment (TME), those that are preserved in spheroids, as the extracellular matrix (ECM), growth factors, cytokines, and chemokines, play a significant role in cancer progression (Paszek et al, 2005). We found that breast cancer 3D spheroids showed high expression of key genes involved in tumor growth, as well as LATS1-dependent nuclear receptor binding features shared with PDXs but absent in 2D cells (Appendix Figs. S3 and S12B). In line with this, LATS1 has been linked to cancer cell plasticity and increased resistance to hormone therapy in breast tumors (Furth and Aylon, 2017).

To assess whether our 3D system recapitulate the resistance of breast cancer cells to CDK4/6 inhibitors and endocrine therapy, which is associated with low FAT1 levels and Hippo pathway activation, we conducted a comparative analysis of gene expression data. Specifically, we compared the gene expression profiles of our 3D breast cancer cell model with those from a previous study (Li et al, 2018). Our analysis revealed that cells resistant to CDK4/6 inhibitors and endocrine therapy exhibited reduced levels of FAT1 and demonstrated activation of the Hippo pathway, which matched the characteristics observed in our 3D-grown cells (as depicted in Appendix Fig. S15). However, the resistant cells had lower progesterone receptor (PR) status (Appendix Fig. S15A), leading to decreased expression of genes controlled by progestin, both in terms of activation and repression (Appendix Fig. S15D). Thus, although our 3D model partially replicated the characteristics of resistant cells, it did not entirely reproduce the diminished hormonal response observed in cells lacking FAT1 expression (Appendix Fig. S15E). In fact, in our 3D model, this response was enhanced (Fig. 6A).

Regarding the heterogeneity, gene expression analysis showed that 3D spheroids are enriched in terms associated to nervous system development compared to 2D cells (Fig. 1F; Appendix Fig. S3A). In fact, increasing evidence suggests that the nervous system itself, as well as neurotransmitters and neuropeptides present in the tumor microenvironment, play a role in orchestrating tumor progression (Fernandez-Nogueira et al, 2016). Thus, 3D cells accurately recapitulate the intratumor heterogeneity facilitating tumor progression and fostering the adaptation and survival of the different tumor cells to the different microenvironments in which a tumor resides.

Our results highlight the importance of the 3D system as a reliable model lacking cross-species incompatibilities for breast cancer studies.

# Methods

## Cell culture and hormone treatments

T47D breast cancer cells were routinely grown in RPMI 1640 medium supplemented with 10% FBS, 2 mM L-glutamine, 100 U/ml penicillin, and 100 μg/ml streptomycin. BT-474, ZR-75, and MCF10A cells were obtained from ATCC and cultured in 2D and 3D in the recommended medium.

For the experiments, cells were plated in RPMI medium without phenol red supplemented with 10% dextran-coated charcoal-treated FBS (DCC/FBS) and 48 h later medium was replaced by fresh medium without serum. After 24 h in serum-free conditions, cells were incubated with 10 nM R5020 for different times at 37 °C.

MCF10A cells were routinely grown in DMEM/F12 medium with 5% horse serum, 20 ng/ml EGF, 100 U/ml penicillin and 100 mg/ml streptomycin, 0.5 mg/mL hydrocortisone, 100 ng/ml cholera toxin, and 10 mg/ml insulin.

## 3D cell culture on Matrigel

Prechilled p60 plates were coated with a thin layer of Matrigel (Corning Life Sciences) and then incubated for 20–30 min at 37 °C to allow polymerization without over-drying. In total, 200,00 cells trypsinized cells were resuspended and plated on top of the Matrigel without disrupting the capsule. Culture was maintained for 10 days changing the medium every 2–3 days. For hormonal induction on the 3D spheroids, T47D cells were plated on top of phenol red-free Matrigel and after 7 days of culture medium was replaced by RPMI medium without phenol red supplemented with 10% dextran-coated charcoal-treated FBS (DCC/FBS) and 48 h later medium was replaced by fresh medium without serum. After 24 h in serum-free conditions, cells were incubated with 10 nM R5020 for different times at 37 °C.

### Spheroids extraction from 3D matrix

Dishes containing the spheroids grown for 10 days were rinsed twice with PBS, followed by the addition of 3 ml ice-cold PBS-EDTA. Matrigel, including the 3D spheroids, was carefully detached with a plastic scraper and left on ice for 30 min to allow complete depolymerization of the gel. The liquid solution was transferred to a conical tube, centrifuged for 5 min at $112 \times g$, and rinsed twice with 0.5 volume of PBS-EDTA. The cell pellet was then ready for further processing.

## Chromatin immunoprecipitation (ChIP) in 2D- and 3D-cultured cells

ChIP assays were performed as described (Strutt and Paro, 1999) using anti-PR (H190 SC-7208, Santa Cruz,); anti-RNApol II (#2629, Cell Signaling); anti-CTCF (07-729, Merck); anti-H3K27ac (ab4729, Abcam); anti-H3K18ac (#39693, Active Motif). Quantification of chromatin immunoprecipitation was performed by real-time PCR using Roche Lightcycler (Roche). The fold enrichment of target sequence in the immunoprecipitated (IP) compared to input (Ref) fractions was calculated using the comparative Ct (the number of cycles required to reach a threshold concentration) method with the Eq. (2) $^{Ct(IP)-Ct(Ref)}$. Each of these values was corrected by the human β-globin gene and referred as relative abundance over time zero. Primers sequences for target regions are available on request.

## RNA interference experiments

Stable LATS-depleted T47D cells were generated by using lentiviral shRNAs obtained from Sigma (MISSION) shRNA Lentiviral Transduction Particles: TRCN0000001777_LATS and TRCN0000001779_LATS.

## Cell proliferation

T47D cells ($1 \times 10^3$) were plated in a non-transparent-walled 96-well plate in RPMI medium or in charcolized medium in the presence or absence of 10 nM R5020. The TiterGlo reagent

(Promega) was added, and cells were then incubated for 2 min at RT with agitation, followed by 10 min incubation at RT. Bioluminescence was detected in a Barthold luminometer system allowing 0.25 s per well. The experiments were performed in quintuplicate.

## Flow cytometry

T47D cells were plated into duplicate wells of six-well plastic dishes and preincubated as described. After 24 h 10 nM R5020 was added. Cells were harvested at the start of treatment (control, zero time) and after 24 h of hormone addition. The cell suspension was pelleted, stained with propidium iodide and treated with ribonuclease (RNase). Samples were cooled to 4 °C, and 10,000 cells were analyzed on a BD FACSCanto analyser flow cytometer.

## Live/dead assay

To visualize the number of viable cells in a 3D spheroid the LIVE/DEAD™ Cell Imaging Kit was used according to the manufacturer protocol.

## Immunofluorescence

For immunostaining, $1 \times 10^3$ cells per well were seeded on Matrigel precoated eight-well LabTek (10 mm). 2D and 3D cells grown for 3 and 10 days, respectively, were washed two times with PBS and fixed for 10 min with 4% fresh Paraformaldehyde (PFA), before being permeabilized during 30 min with 0.5% Triton X-100 in PBS. Samples were then blocked during 1.5 h with 5% BSA solution. Then, the cells were incubated o/n at 4 °C with the corresponding antibody diluted in the IF solution.

The following day, cells were washed three consecutive times with IF solution during 10 min each. The samples were then incubated during 1 h at 4 °C with the secondary antibody in a humid dark chamber.

The secondary antibodies used were: AlexaFluor 488 anti-rabbit IgG (1:1000; raised in donkey) and AlexaFluor 546 anti-mouse (1:1000; raised in goat). After the incubation with the secondary antibody, cells were washed once with IF solution, incubated during 10 min with 0.1 mg/ml DAPI in PBS, and then washed with PBS three times, before mounting in Mowiol 4-88 Mounting Medium for imaging. Images were collected sequentially on a Leica SP8-STED confocal laser-scanning microscope using the software Leica Application Suite X. All collected images conserved an optical thickness of 0.25 µm. Image analysis for marker distribution, quantification, and colocalization were performed using ImageJ (Schindelin et al, 2012).

## Preparation of cell extracts and western blot

Matrigel-free cell pellets were collected and washed two times with PBS-EDTA and lysed with RIPA buffer followed by incubation at 95 °C for 10 min. For western blotting, pellets were centrifuged and quantified by Bradford before being loaded and run in SDS-acrylamide gels. The following antibodies were used: CTCF (07-729, Millipore); ERα (H20, Santa Cruz); PR (H190, Santa Cruz); FAK (PTK2) (3285, Cell Signaling); LATS (3477, Cell Signaling); LATS-S909(9157, Cell Signaling); pFAKT397 (ab81298, Abcam);

pPRs294 (ab61785, Abcam); pYAP127 (4911, Cell Signaling) and YAP (sc-101199, Santa Cruz).

## Phos-tag gels

Whole-cell lysates were separated on a 5% SDS/PAGE containing 20 µM Phos-tag (NARD Institute), followed by western blotting with anti-CTCF antibody.

## MNase digestion

2D and 3D-grown T47D cells were washed once with PBS, collected in 2 ml cold PBS + PIC, and centrifuged 5 min at $900 \times g$ at 4 °C. The cell pellet was then gently resuspended in 50 µl RBS buffer (10 mM Tris-HCl pH 7.4, 10 mM NaCl, 3 mM CaCl2.) followed by the addition of 1.3 ml RBS buffer + 0.1% NP40. Cells were centrifuged again for 10 min at $500 \times g$ at 4 °C and the nuclei were then resuspended in RBS buffer for counting. An amount of 600,000 nuclei obtained from 2D and 3D-grown cells were treated with 0, 30, 45, 90 Units of MNase to obtain differential digestion patterns. The MNase reaction was carried out in 500 µl final volume reaction. The nuclei were then incubated for 2 min at 37 °C. The reaction was stopped with 40 mM EDTA 0.5 M and then treated with RNAseA 10 mg/ ml for 30 min at 37 °C and Proteinase K (1.2 µg/µl) for 1 h at 45 °C. The DNA was purified with Phenol/Chloroform, and 600 ng of material was loaded in a 1.2% agarose gel.

## RNA-seq

RNA was extracted from T47D cells grown in 2D and 3D conditions in RPMI 1640 medium. To evaluate the effect of the hormone, 2D and 3D T47D cells were treated or not for 6 h with 10 nM R5020 and submitted to massive sequencing using the Solexa Genome Analyzer. The protocol followed to analyze the RNA-seq data can be found in the Supplementary Methods section.

## ChIP-seq

ChIP-DNA was purified and subjected to deep sequencing using the Solexa Genome Analyzer (Illumina, San Diego, CA). The protocol followed to analyze the ChIP-seq data can be found in the Supplementary Methods section.

## ATAC-seq

ATAC experiments were performed as described (Buenrostro et al, 2013) using nuclei obtained from 2D or 3D-grown T47D cells. Extended bioinformatics methods can be found in the Supplementary information.

## Hi-C experiments

Hi-C libraries were generated from 2D and 3D-grown T47D cells treated or not with R5020 for 60 min according to the previously published Hi-C protocol with minor adaptations (Lieberman-Aiden et al, 2009). Hi-C libraries were generated independently in both conditions using HindIII and NcoI restriction enzymes. Hi-C libraries were controlled for quality and sequenced on an Illumina

Hiseq2000 sequencer. The Illumina Hi-seq paired-end reads were processed by aligning to the reference human genome (GRCh37/hg19) using BWA.

## Data availability

The raw sequencing data from this study (ChIP-seq, ATAC-seq, RNA-seq, and Hi-C) were submitted to the NCBI Gene Expression Omnibus (GEO; http://www.ncbi.nlm.nih.gov/geo/) repository, and the accession number is GSE247777.

## Peer review information

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

## Acknowledgements

The authors would like to thank Jianrong Lu (University of Florida College of Medicine, USA) for generously sharing the CTCF constructs. Additionally, we also would like to thank Dr. Julia Ponomarenko (Bioinformatics Unit, CRG) for their valuable advice, technical assistance and involvement in processing the RNA-seq, ChIP-seq, and ATAC-seq data. The experimental work was supported by grants from the Departament d'Innovació Universitat i Empresa (DIUiE), the Spanish Ministry of Economy and Competitiveness (SAF2016-75006-P, PID2019-105173RB-I00, and PID2022-137045OB-I00) and Consejo Superior de Investigaciones Científicas (Ref# 2018201131), 'Centro de Excelencia Severo Ochoa 2013-2017', SEV-2012-2018 and ERC Synergy Grant "4DGenome" nr: 609989. MAM-R acknowledges support by the Spanish Ministerio de Ciencia e Innovación (PID2020-115696RB-I00). This work has benefited from the equipment and framework of the COMP-HUB and COMP-R Initiatives, funded by the 'Departments of Excellence' program of the Italian Ministry for University and Research (MIUR, 2018-2022 and MUR, 2023-2027) and from the HPC (High-Performance Computing) facility of the University of Parma, Italy.

## Author contributions

**Julieta Ramirez Cuellar**: Investigation; Methodology. **Roberto Ferrari**: Formal analysis; Investigation. **Rosario T Sanz**: Investigation; Methodology. **Marta Valverde-Santiago**: Investigation; Methodology. **Judith García García**: Formal analysis; Investigation; Methodology. **A Silvina Nacht**: Investigation; Methodology. **David Castillo**: Formal analysis; Investigation; Methodology. **Francois Le Dily**: Formal analysis; Investigation. **Maria Victoria Neguembor**: Formal analysis; Investigation; Methodology. **Marco Malatesta**: Formal analysis; Investigation; Writing—review and editing. **Sarah Bonnin**: Formal analysis; Supervision; Investigation; Methodology; Writing—original draft; Writing—review and editing. **Marc A Marti-Renom**: Formal analysis; Investigation; Methodology. **Miguel Beato**: Investigation; Methodology; Writing—review and editing. **Guillermo P Vicent**: Formal analysis; Supervision; Investigation; Methodology; Writing—original draft; Writing—review and editing.

## Disclosure and competing interests statement

The authors declare no competing interests.

# Expanded View Figures

**Figure EV1.  Profiles of H3K27me3, H3K9me3 and super-enhancers detected in 2D and 3D T47D breast cancer cells.**

(**A**) The H3K27me3 profiles in 3D-repressed and activated genes (blue and green lines, respectively) obtained in both conditions (first and second panels, from the left) is shown. Third and fourth panel from the left: profiles of H3K27me3 in 2D and 3D random genes. Right panel: Genome browser view of H3K27me3 ChIP-seq data in the NRCAM gene. (**B**) The H3K9me3 profiles in 3D-repressed and activated genes (blue and green lines, respectively) obtained in both conditions (first and second panels, from the left) is depicted. Third and fourth panels from the left: profiles of H3K9me3 in 2D and 3D random genes. (**C**) The enrichment of the H3K27ac signal in super-enhancers obtained from 2D and 3D cells is shown. (**D**) Venn diagram corresponding to the super-enhancers detected in T47D cells grown in 2D and 3D conditions.

▶

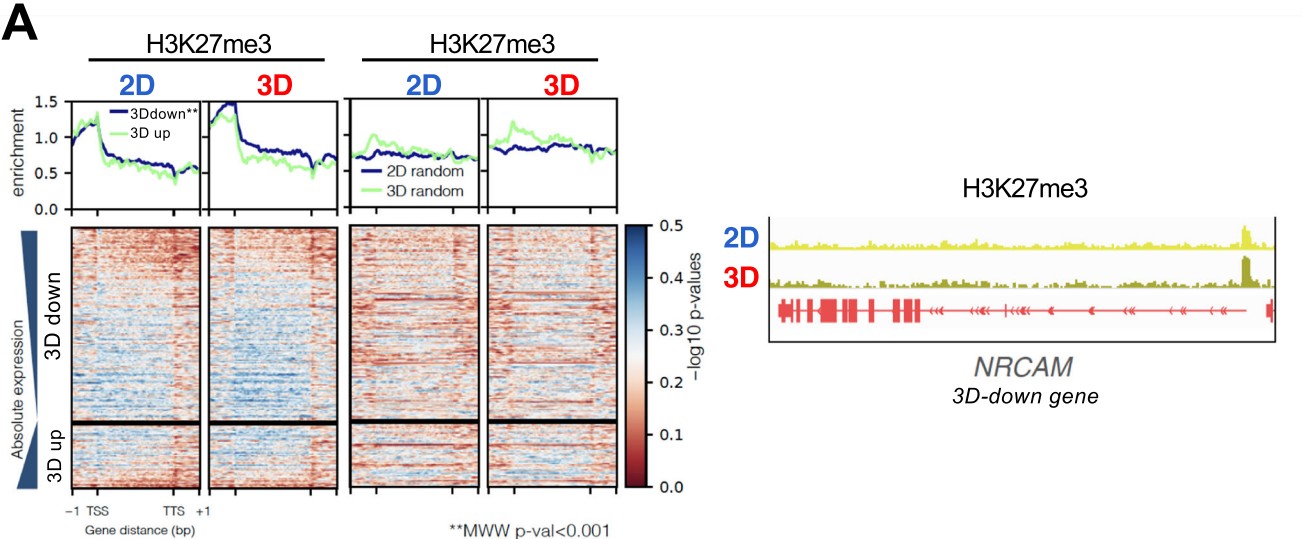

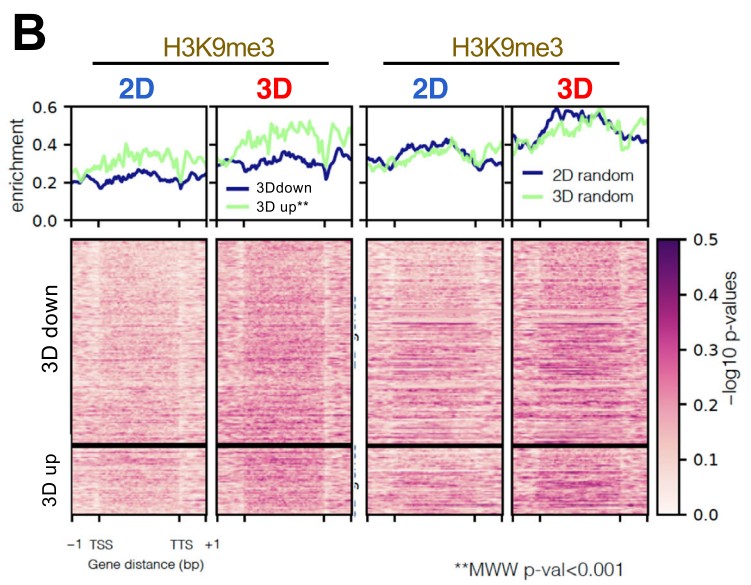

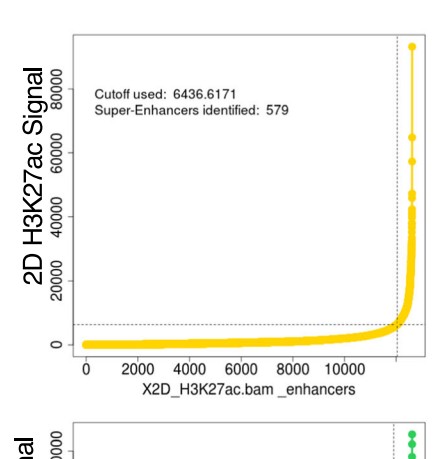

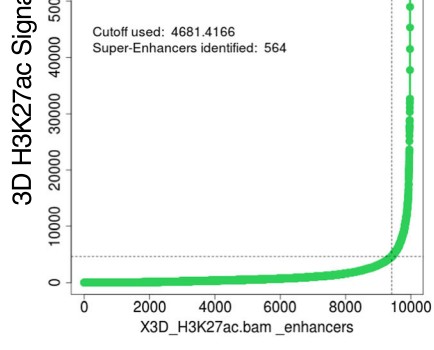

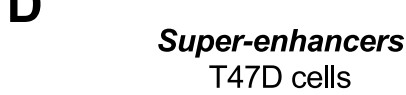

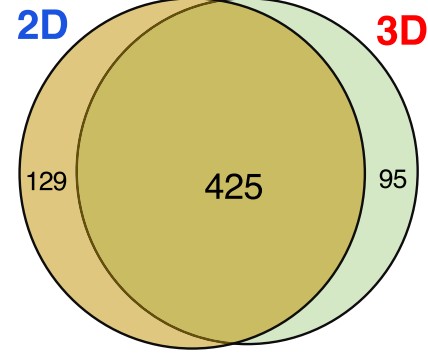

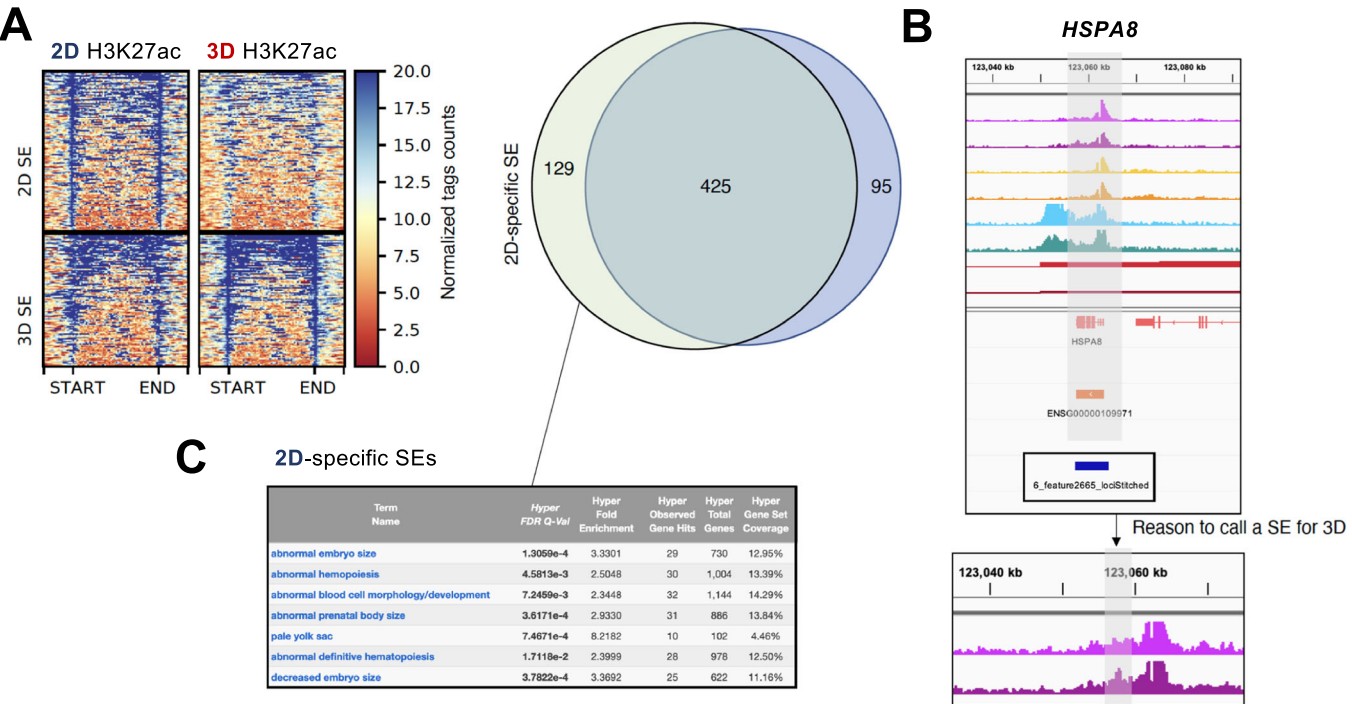

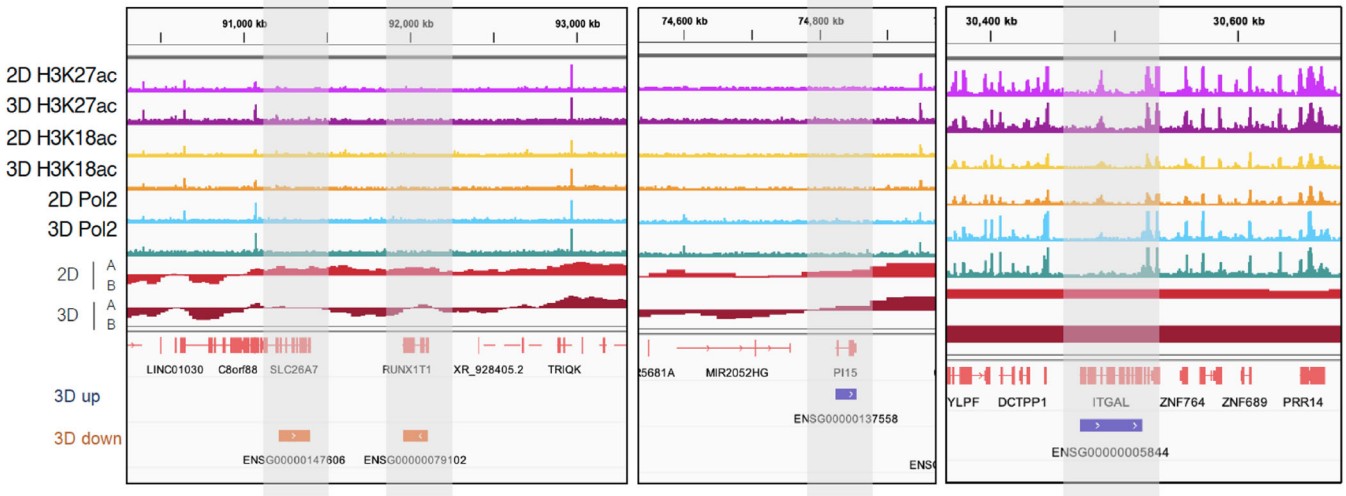

**Figure EV2. Characterization of super-enhancers identified in 2D and 3D cells.**

(A) The enrichment of the H3K27ac signal in super-enhancers obtained from 2D and 3D cells is shown. Venn diagram corresponding to the super-enhancers detected in T47D cells grown in 2D and 3D conditions. By using a proximity-based script (Hnisz et al, 2013), we found 129 and 95 genes associated to 2D and 3D SEs, respectively (right panel). In the case of genes exclusively regulated in the 3D condition by SEs, many of them appear to be artifacts, as illustrated with the HSPA8 gene (B). (C) The 2D-exclusive genes are related to terms like abnormal embryo size, abnormal development, and hematopoiesis. (D) Snapshots from the genome browser illustrating the transitions for two 3D downregulated genes, SLC26A7 and RUNX1T1, and two 3D upregulated genes, PI15 and ITGAL are shown.

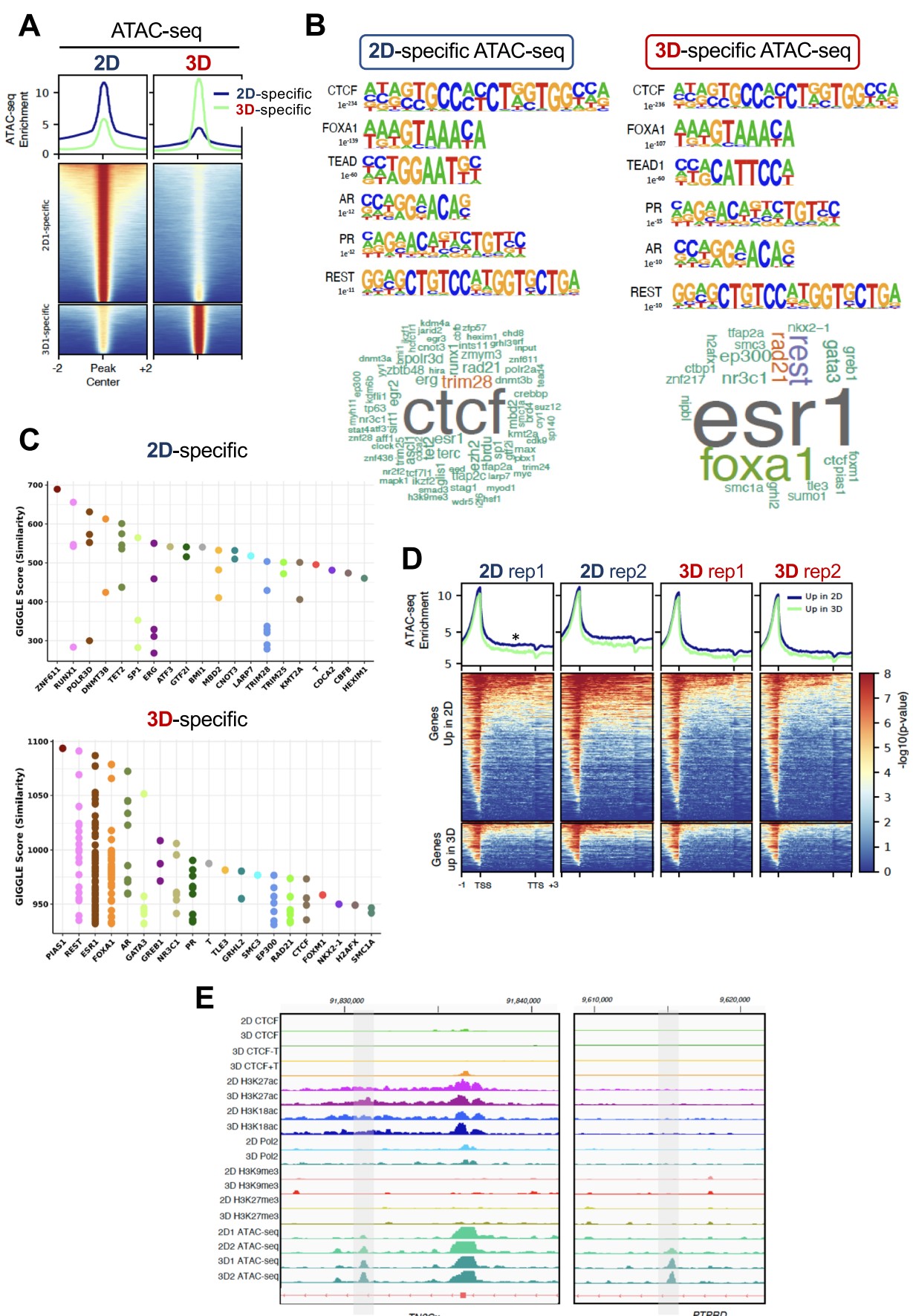

◀  **Figure EV3.   Regions more accessible in 3D are enriched in estrogenic signaling.**

(**A**) Heatmaps of ATAC-seq data performed in 2D and 3D T47D cells. The exclusive regions belonging to each condition is highlighted. (**B**) Homer motif analysis of 2D and 3D-exclusive ATAC-seq regions (upper panel). When the same regions are contrasted with available ChIP-seq data, the CTCF and ESR1 terms appear enriched (bottom panels). (**C**) Giggle score of the data presented in (**B**). (**D**) Heatmaps of the ATAC-seq signal around 2D and 3D upgenes. (**E**) Snapshot of the genome browser around TN2Cx and PTPRD genes showing the profiles of H3K27ac, RNAPol2, H3K9me3, H3K27me3 and ATAC-seq.

## A

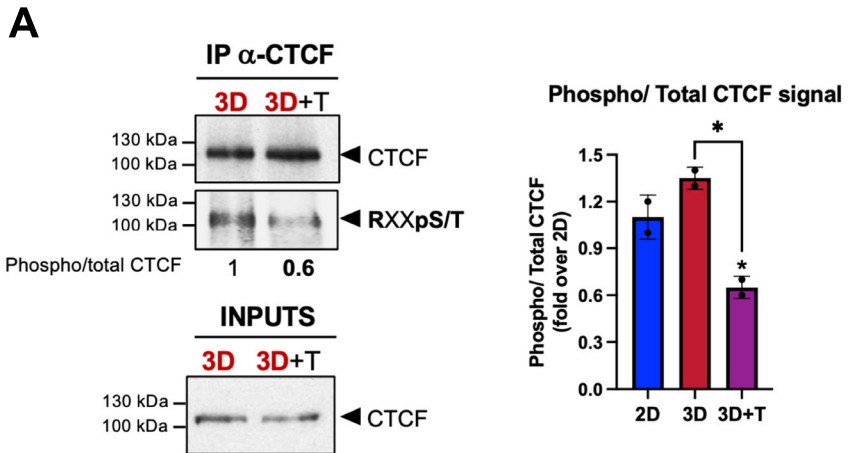

**IP α-CTCF**

Phospho/total CTCF    1    0.6

**INPUTS**

**Phospho/ Total CTCF signal**

## B

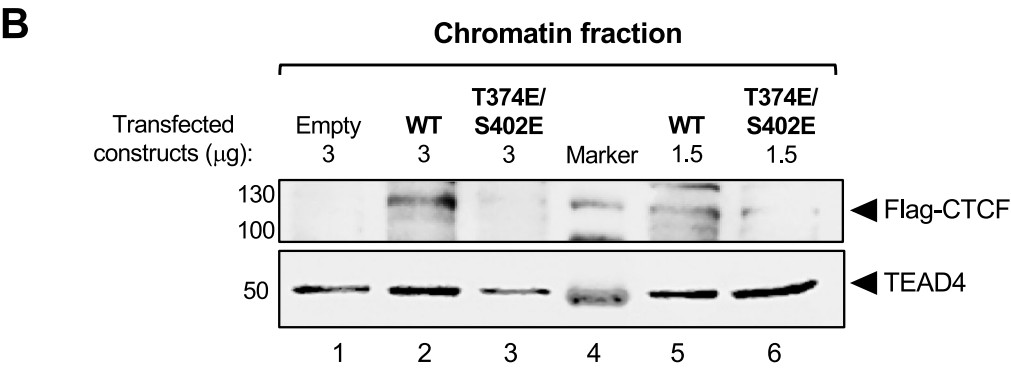

**Chromatin fraction**

## C

## D

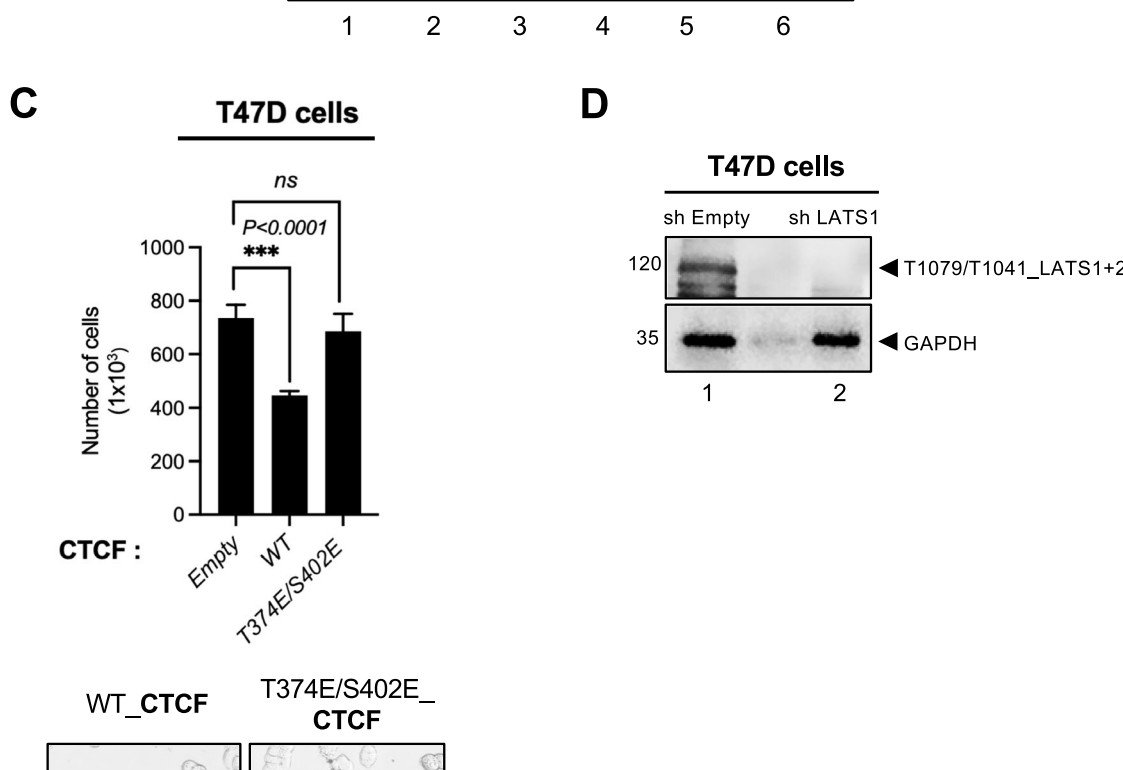

**T47D cells**

WT_CTCF          T374E/S402E_CTCF

◀ **Figure EV4.  CTCF phosphorylation is reduced in the presence of LATS inhibitor.**

(**A**) 2D, 3D and 3D treated with TRULI T47D cells were cultured and subjected to immunoprecipitation with anti-CTCF antibodies and the immunoblotting for phospho-RxxS/T was performed. For quantification, the intensity of phospho-CTCF versus total CTCF signal in control samples is shown (right panel). (**B**) T47D cells were transiently transfected with wild-type (WT) and T374E/S402E (phospho-mimetic) CTCF flag-tagged constructs. Subsequently, cells were lysed, and the chromatin fraction was isolated to display the presence of flag-CTCF bound to the chromatin fraction. (**C**) Cell growth assays performed in T47D cells expressing both wild-type (WT) and a phospho-mimetic variant of CTCF T374E/S402E. (**D**) The levels of T1079/T1041p LATS1 + 2 signal in both shEmpty (control) and shLATS1 cells is shown. The noticeable decrease in the phospho T1079/T1041 signal and LATS1 (Fig. 5C) suggests that the predominant portion of detected p-LATS corresponds to LATS1 in the 3D model. GAPDH is used as loading control. Source data are available online for this figure.

# A

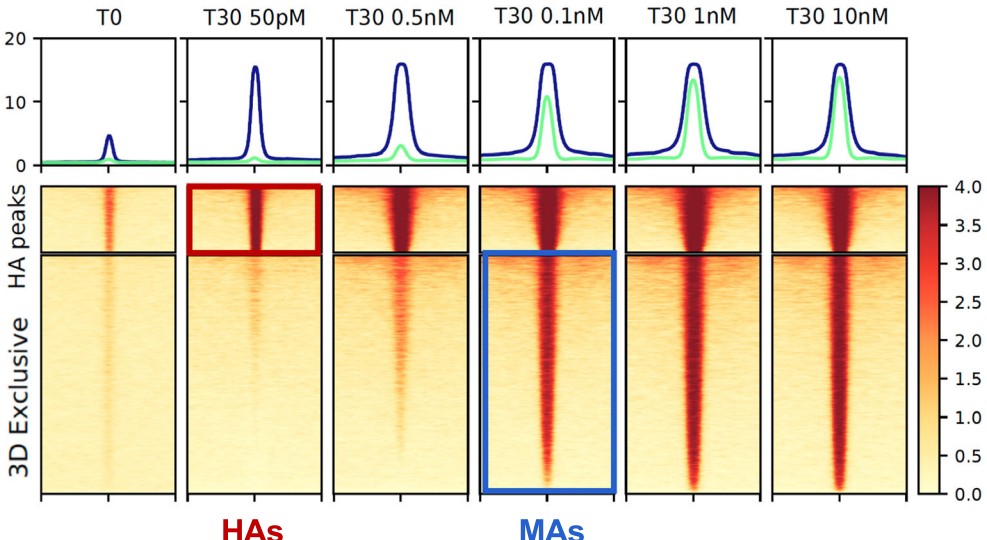

# B

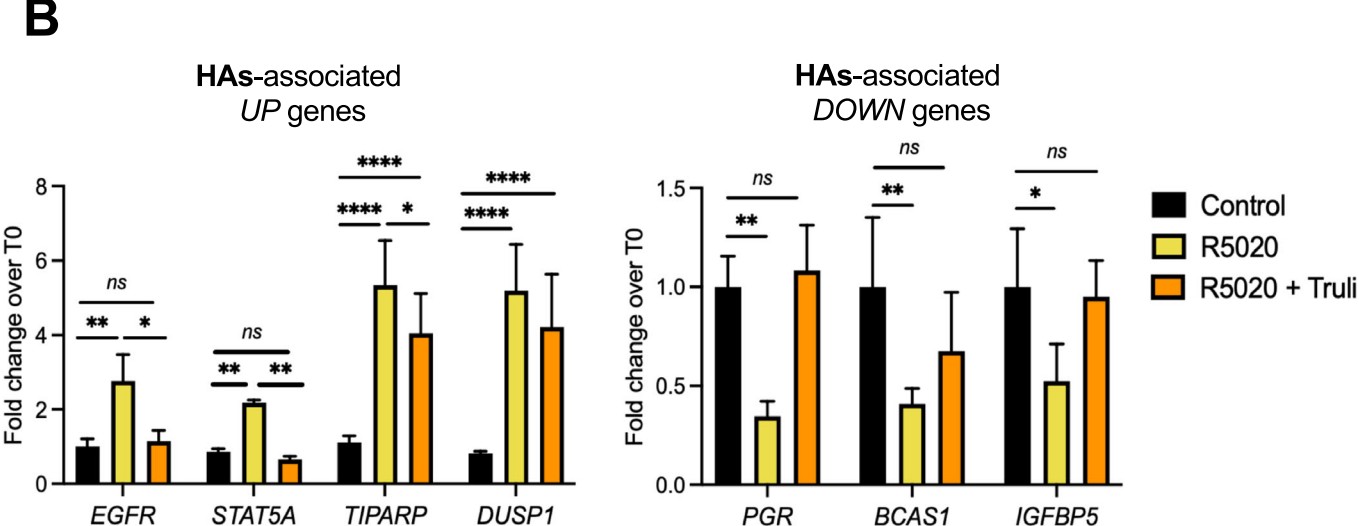

**Figure EV5.  Hormone-dependent CTCF recruitment to High accessible PR binding sites (HAs) requires LATS1 activity.**

(A) Heatmaps of PR ChIP-seq signal obtained at different concentrations of R5020 and corresponding to HAs and 3D-exclusive PR-binding sites are shown. (B) Cells grown in 3D conditions and treated or not with R5020 and TRULI as indicated, were submitted to gene activity assays. Four up and three down-HAs-associated genes were tested. Results are represented as mean and SD from two experiments performed in duplicate. The *P* value was calculated using the Student's *t* test.

