## [Peer Review File · The EMBO Journal]

LATS1 controls CTCF chromatin occupancy and hormonal response of 3D-grown breast cancer cells

Julieta Cuellar, Roberto Ferrari, Rosario Sanz, Marta Valverde-Santiago, Judith Garcia-Garcia, A. Silvina Nacht, David Castillo, François Le Dily, Maria Neguembor, Marco Malatesta, Sarah Bonnin, Marc A. Marti-Renom, Miguel Beato, and Guillermo Vicent

Corresponding author(s): Guillermo Vicent (gymbmc@ibmb.csic.es)

Review Timeline:

Submission Date:	15th May 23
Editorial Decision:	22nd May 23
Appeal Received:	25th May 23
Editorial Decision:	30th May 23
Revision Received:	30th May 23
Editorial Decision:	21st Jul 23
Appeal Received:	22nd Nov 23
Editorial Decision:	23rd Dec 23
Appeal Received:	17th Jan 24
Editorial Decision:	17th Jan 24
Revision Received:	5th Feb 24
Accepted:	27th Feb 24

Editor: Cornelius Schneider

Transaction Report:

Dear Dr. Vicent,

thank you for submitting your manuscript " LATS1 controls CTCF chromatin occupancy and hormonal response of 3D-grown breast cancer cells" to the EMBO Journal. I have now carefully read your manuscript and discussed the work with other members of the editorial team. I am sorry to say that our conclusion from all these considerations is that we cannot offer to consider publication of the manuscript in the EMBO Journal.

We appreciate your detailed investigation of the similarities and differences in nuclear architecture and chromatin organization between breast cancer cells grown in 2D - and 3D cell culture. We also appreciate that the manuscript proposes a model how the observed differences in signaling and transcription regulation can be linked back to the different cellular environments which the cells find themselves in under 2D or 3D cell culture. We are concerned however that the manuscript does not show or discuss if the 3D specific properties more faithfully recapitulate in vivo tumor cells and their physiological environment. We recognize the importance and the need for better in cellulo experimental systems to study cancer biology but, given that it remains unexplored if and in what regard and to what extend 3D culturing is closer to in vivo tumor cells, we unfortunately concluded that your manuscript in its current form is not a sufficiently strong candidate for publication in The EMBO Journal.

Thank you again for giving us the opportunity to consider this manuscript. I am sorry that I could not share a better outcome this time, and I hope for a timely publication of the manuscript at another venue of your choice.

Yours sincerely,

Cornelius Schneider

Cornelius Schneider, PhD
Editor
The EMBO Journal
c.schneider@embojournal.org

** As a service to authors, EMBO Press provides authors with the possibility to transfer a manuscript that one journal cannot offer to publish to another EMBO publication or the open access journal Life Science Alliance launched in partnership between EMBO Press, Rockefeller University Press and Cold Spring Harbor Laboratory Press. The full manuscript and if applicable, reviewers' reports, are automatically sent to the receiving journal to allow for fast handling and a prompt decision on your manuscript. For more details of this service, and to transfer your manuscript please click on Link Not Available. **

MINISTERIO
DE CIENCIA
E INNOVACIÓN

Instituto de Biología Molecular de Barcelona
(IBMB)

Barcelona, May 25th 2023

Dear Dr Cornelius Schneider
Editor
The **EMBO Journal**,

Thank you for your recent email and the editorial decision on our manuscript "*LATS1 controls CTCF chromatin occupancy and hormonal response of 3D-grown breast cancer cells*" (EMBOJ-2023-114519). We acknowledge the points that are appreciated in our work such as the similarities and differences in nuclear architecture and chromatin organization and the proposed model on how the observed differences in signaling and transcriptional regulation can be linked to the different cellular environments. However, we are respectfully disappointed by the decision not to submit our manuscript for peer review. We understand that your decision is based on the following statement: "*We are concerned however that the manuscript does not show or discuss if the 3D specific properties more faithfully recapitulate in vivo tumor cells and their physiological environment*".

We believe that your concern is justified, as we have not been clear enough in highlighting -through experimentation or appropriate discussion- the importance of spheroids as a system that can effectively recapitulate the situation of the breast cancer cell in its physiological environment. Therefore, we would like to have the opportunity to resubmit the manuscript addressing this issue.

In case you agree, the new version will include new data and analysis, as well as a broader and improved discussion, on how the 3D spheroids better recapitulate the breast tumor condition.

Specifically, we will incorporate new RNA-seq data supporting that gene expression pattern in 3D spheroids is more similar to that occurring in estrogen receptor positive (ER⁺) patient derived xenografts (PDXs) than its 2D counterpart. To address this point, we use the Mutant or Translocated Estrogen Receptor Alpha (MOTERA) signature, composed of 24 genes characteristic of the presence of an active ER in PDXs (Gou et al., *Cancer Res* 2021). By using this approach, 3D grown cells showed increased expression of 58% of the MOTERA signature genes when compared to 2D cells (14 out of 24 genes, see Figure 1). Next, we asked whether the sites where the progesterone receptor (PR) binds only in 3D cells with hormone (3D-exclusive set) are also occupied in PDXs. By using new ChIP-seq data performed in two different ER⁺/PR⁺ PDXs (T99 and HCI-005) and under different conditions, we found that effectively, PR binds in PDX to the same regions as in 3D cells (Figure 2). Moreover, ER and androgen receptor (AR) binding is also detected in both PDXs, in the presence of their respective ligands, E2 and DHT (Figure 2). Therefore, nuclear receptor binding in the tumor is efficiently recapitulated by 3D cells.

IBMB-CSIC
Parc Científic de Barcelona
C/ Baldori Reixac, 4-8
Torre R, pl. 3
08028 BARCELONA
TEL.: +34 93 402 9060
FAX: +34 93 403 4979

MINISTERIO
DE CIENCIA
E INNOVACIÓN

Instituto de Biología Molecular de Barcelona
(IBMB)

Thus, hormone-dependent regulation of key genes (Figure 1) as well as hormone-receptor binding *in PDXs* (Figure 2) are conserved in 3D cells, highlighting the importance of this system as a breast cancer model.

In addition, we will discuss our results in the context of the currently available literature in PDXs (PMID: 27063403, PMID: 25453261, PMID: 28647083, PMID: 33669619 among others), emphasizing the similarities that support the use of this 3D system as a tool that mirrors the physiological situation of the tumor cell.

We believe that these new results will increase the physiological relevance of our manuscript. Therefore, we would very much appreciate if you would consider to evaluate a new version of our manuscript incorporating the additional results and an updated discussion.

Looking forward to hearing from you,

Yours sincerely

Guillermo Vicent

Guillermo P. Vicent, PhD
Chromatin and Gene Expression Lab (PBA33)
Structural and Molecular Biology Department
Molecular Biology Institute of Barcelona, IBMB-CSIC
C/ Baldiri Reixac, 4-8; E-8028
Barcelona, Spain
@: gvmbmc@ibmb.csic.es

IBMB-CSIC
Parc Científic de Barcelona
C/ Baldiri Reixac, 4-8
Torre R, pl. 3
08028 BARCELONA
TEL.: +34 93 402 9060
FAX: +34 93 403 4979

Dear Dr. Vicent,

thank you for coming back to me so quickly. The emails you received were only an acknowledgement of your appeal by the journal and not yet an invitation for re-submission. I agree with you though that the new data and sections in the manuscript address my concerns and have therefore indeed decided to send the manuscript for peer review. To make sure that the referees receive the complete revised manuscript could you please re-submit using the link below.

Yours sincerely,

Cornelius Schneider, PhD
Editor
The EMBO Journal
c.schneider@embojournal.org

We realize that it is difficult to revise to a specific deadline. In the interest of protecting the conceptual advance provided by the work, we recommend a revision within 3 months (28th Aug 2023). Please discuss the revision progress ahead of this time with the editor if you require more time to complete the revisions. Use the link below to submit your revision:

Instituto de Biología Molecular de Barcelona
(IBMB)

Barcelona, July 19th 2023

Dear Dr. Cornelius Schneider
Editor
The EMBO Journal

I hope this e-mail finds you well. I would like to express my gratitude for your recent email and for sharing the evaluations of our manuscript submitted to EMBO J (EMBOJ-2023-114519R1) with us. Your feedback, as well as the input from the three reviewers, is greatly appreciated.

After carefully reviewing the reviewers' comments, I would like to outline our plans for addressing the majority of their concerns. However, I must note that while we are committed to completing the revisions within a timeframe of less than six months, I cannot provide an exact deadline at this stage. It is important to consider that the upcoming summer break period, from mid-July to the end of August in Spain, will impact the timeline of our proposed experiments and analyses.

Nevertheless, I would like to highlight the experiments and analyses that we plan to include in the potential revised version of our manuscript. These are as follows:

REVIEWER 1

1-Bioinformatics analyses: We fully agree with Reviewer 1's suggestions regarding the bioinformatics analyses outlined in major points 1, 3, and 7. We will incorporate these analyses into our potential revised manuscript accordingly. However, we would like to note that we have not yet obtained access to the data from the paper by Li et al. (Cancer Cell 2018) as referred to by the reviewer. Nevertheless, we are confident that we can address this concern appropriately.

2-Experimental part: Reviewer 1 requests a ChIP-seq analysis of a component of Cohesins to explain the disruption of TADs in the 2D versus 3D conditions. We appreciate the reviewer's interest in this aspect. However, based on our current results (Figure 3A, D, and E), we have observed that overall, the 2D and 3D TADs remain largely unchanged, with approximately 90% of TADs showing no significant differences. The remaining 10% exhibit moderate changes, primarily displaying a trend towards fusion in

3D. This lack of significant change in TADs can be attributed to the persistence of 20-30% of CTCF and/or the Cohesin complex binding in both conditions.

In fact, we have performed ChIP-seq experiments for RAD21 (a component of the Cohesin complex) in 2D and 3D conditions, and our results demonstrate no significant differences in RAD21 content between fused and random TADs. We have included these findings in **Figure 1** (attached for the Editor). It is important to note that these ChIP-seq experiments were conducted under slightly different growth conditions (absence of phenol red and presence of charcoalized serum) compared to the genome architecture assays depicted in Figure 3 but similar to those used in Figure 6. Considering the above, we can cautiously discuss these new results and the potential involvement of Cohesin components in our systems in a potential revised version of our manuscript.

3-Contribution of LATS2: We agree with Reviewer 1's proposal to consider the potential involvement of LATS2 in the observed effects. However, measuring the activation of LATS2 presents challenges due to the lack of commercial antibodies specifically detecting the phosphorylation at T1041 resulting from the activation of the Hippo pathway via MST1/2. We propose using an antibody (PA5-117227 from Thermo Fisher) that recognizes LATS1 and LATS2 phosphorylated at T1079 and T1041, respectively, in a background of LATS1-depleted cells as shown in Figure 5C. This approach will allow us to visualize the putative activation of LATS2 exclusively in the 3D condition. LATS2 depletion by siRNA could also be attempted, but transfections in 3D-grown spheroids is still very challenging and the creation of stable shRNA cell lines will be time-consuming and is unlikely to significantly alter the conclusions of our study. Therefore, we suggest evaluating LATS2 activation using the dual antibody (Thermo) in shLATS1 cells. If a positive signal is detected, we will clarify throughout the manuscript that part of the observed effect could be attributed to the activation of LATS2 in 3D.

4-Phosphorylation of CTCF: Reviewer 1 requests formal proof of CTCF phosphorylation in 3D. We propose conducting experiments similar to those performed by Luo et al. in 2020. We will immunoprecipitate CTCF and probe the resulting blot with an antibody (Cell Signaling #9614 RxxS7T) capable of detecting LATS1-mediated phosphorylations. We will compare the following conditions: 2D, 3D, and 3D with LATS1 inhibitor (TRULI). This approach will enable us to investigate the phosphorylation status of CTCF under these different growth conditions.

5-Cell proliferation and activation of LATS1: We can repeat the experiments measuring cell proliferation, LATS1 activation, and pFAK inhibition (Figure 1C, D, and E) using ER+

different breast cancer cell lines such as MCF-7 or BT-474. If LATS1 is found to be activated in 3D, we can expand the analysis to detect phosphorylated CTCF using phos-tag gels. Additionally, we can evaluate the response of progesterone-target genes comparing 2D and 3D conditions.

Additionally, we assure you that we will address all minor points raised, including panel corrections, improving the quality of immunofluorescence images, providing data on cell numbers, and optimizing Western blot presentations.

REVIEWER 2

1- *Points 1, 3, 4, 5, 7, and 9:* We appreciate Reviewer 2's request to discuss or include additional information on these points. We will incorporate the relevant discussions and supplementary data as suggested by the reviewer without any inconvenience.

2- Reviewer 2 suggests extending the analysis of 3D genes that coincide with the breast cancer signature, with a specific focus on genes involved in neurogenesis. We agree with this suggestion and will perform an extended analysis of these genes as recommended.

3-The reviewer does not consider this point relevant to support the conclusions of this part of the manuscript. However, we can explore the use of POSSUM to map the smaller compartments as a potential analysis. We leave it to the Editor's discretion to determine whether this analysis is relevant and should be included.

4- Reviewer 2 suggests increasing the size of the lettering in Figures 4E and F. We will ensure that the lettering is appropriately increased to improve the legibility of these figures.

5-*Regarding point 9:* In Figure appendix S4F, we demonstrate that the regions losing accessibility in 3D are highly enriched in CTCF and BORIS (CTCF-L), leading to the logical assumption that TRULI-induced recovery of CTCF binding will increase accessibility once again. Although we do not consider this point to be highly relevant to the overall message of our manuscript, to address this concern, we can discuss the dynamics of CTCF binding and accessibility in the context of the available literature. However, if the Editor considers this discussion insufficient, we are open to exploring an experimental approach to further support this aspect.

REVIEWER 3

1-*Statistical tests, replicates, and deposition details*: We will upload the required information, including the statistical tests used, details on replicates, and deposition details, to GEO shortly.

2-*Figure S1B*: The results can be presented as violin plots, as suggested by the reviewer.

3- *FACS analysis of 3D spheroids (Figure S2A)*: We attempted to perform this analysis; however, technically, it was challenging as the cells did not disperse sufficiently to pass through the FACS. When treated with trypsin for longer times to achieve single-cell suspensions, the results were not reliable due to an increased percentage of damaged cells. Therefore, we are unable to provide the requested FACS analysis for Figure S2A.

4-*Combined analysis of GO terms and KEGG (Figure 1F)*: We will expand on the combined analysis of GO terms and KEGG to further support the conclusions presented in our manuscript.

5-*Rewriting text on H3K27ac, H3K18ac, and Figure 3*: We will rewrite the text on page 6 to provide more precise information, as suggested by the reviewer. We will also ensure that the discussion avoids confusing correlation with causation when considering Hi-C compartments in Figure 3.

6-*Clarification of MNase and ATAC-seq information in Figure 2*: We will clarify the different information provided by MNase and ATAC-seq assays, specifically addressing the general compaction at a large scale (MNase) and the localized loss of accessibility provided (ATAC-seq).

7-*Discussion of H3K27me3 levels in 3D down-regulated genes (Figure EV2A)*: We propose discussing the implications of the small change in H3K27me3 levels observed in 3D down-regulated genes within the context of the available literature. We will explore potential mechanisms involved and whether this change could partially contribute to the observed lower gene activity.

8-*Addressing concerns about CTCF ChIP-seq (Figure 4)*: We respectfully disagree with the reviewer's suggestion that the lower signal observed in the 3D CTCF ChIP-seq could be attributed to technical issues associated with performing ChIP experiments on 3D spheroids. We believe this is not the case due to the following reasons:

i) The size and quality of the chromatin used in both 2D and 3D conditions is similar, as demonstrated by the corresponding agarose gel images (see **Figure 2A**, attached for the Editor). This similarity in chromatin size indicates that the technical aspects of chromatin preparation are comparable between the two conditions. In fact, the composition of the chromatin in 2D and 3D looks very similar as levels of Histone H1.2 and FOXA1 (pioneer factor which can bind nucleosomes) do not change between conditions (**Figure 2B**, attached for the Editor)

ii) The decrease in signal observed in 3D is specific to CTCF, as other ChIP-seq experiments targeting PR, H3K18ac, H3K27ac, H3K27me3, H3K9me3, and RNAPol II, among others, either exhibit small changes or an increase in 3D cells. Furthermore, in our ChIP-seq experiments performed in 3D, CTCF regains binding to the genome in the presence of the LATS1 inhibitor (TRULI), as shown in Figure 5D-E. These results strongly support the specific nature of the observed decrease in CTCF binding in 3D.

Additionally, in our chromatin fractionation assays conducted in 2D and 3D-grown cells (Appendix Figure S4A), we have observed that approximately 50% less CTCF is bound to the chromatin fraction in 3D compared to the 2D condition. This finding -by a different technical approach- further supports the decreased CTCF binding observed in 3D samples.

While we agree that the use of spike-in controls, such as *Drosophila* chromatin, in ChIP-seq experiments is generally recommended to compensate for potential material loss during immunoprecipitation, we believe that the differences observed in our CTCF ChIP-seq are substantial enough that the inclusion of spike-in controls would not alter the main conclusions of our study. However, if the Editor considers it necessary to conduct additional experiments, such as monitoring several CTCF binding sites by ChIP-qPCR using spike-in controls and other control regions, we are open to performing them.

9-Furthermore, in response to the reviewer's suggestion, we will include a protein not phosphorylated by LATS as a control in Figure 5A.

10-The reviewer suggests conducting transfections of CTCF with mutations in T374 and S402, which are the phosphorylation sites targeted by LATS1, to investigate their impact on chromatin binding and genome structure. However, as mentioned above, performing transfections on T47D cells grown as 3D spheroids poses challenges in achieving uniform incorporation of the DNA construct among the spheroid cells. This lack of uniformity in transfection may compromise the reliability of the results obtained, and therefore, will not provide us valuable information to support the raised hypotheses.

By conducting these additional investigations, we believe that we can effectively address the concerns raised by the three reviewers and strengthen the overall quality of our manuscript.

Best Regards,

Guillermo Vicent
IBMB-CSIC

Dear Dr. Vincent,

Thank you very much for sending the preliminary point-by-point reply to the referee comments. I have studied them carefully and have also discussed the proposed experiments with my colleagues. I am sorry to say that we have decided not to pursue the publication of the manuscript at The EMBO Journal but would suggest transfer to our sister journal EMBO Reports.

We appreciate that you propose a large body of revision experiments that will address many of the concerns raised by all three referees. We have also noted though that there are several points raised especially by referee #3 that you do not plan to address experimentally. Your response convincingly explains the technical difficulties that prevent you from performing the required experiments. Nonetheless, given the overall critical assessment of the manuscript by referee #3 we do not believe that the proposed revisions will lead to a positive final assessment by this referee. To avoid a prolonged and unproductive revision process we have therefore decided to not pursue publication of this manuscript at the EMBO Journal.

I have shared your manuscript with my colleague Esther Schnapp at EMBO Reports. She would be interested in the manuscript and would invite the proposed revisions if you decide to transfer to EMBO reports. EMBO Reports would not require the ChIP-seq analysis of a component of Cohesins, would not ask you to perform experiments regarding the optional points raised by referee 2 point 2 (as per authors PBP) and you would also not need to include spike in controls as requested by referee #3. If you have any further questions, don't hesitate to contact Esther Schnapp (e.schnapp@emboreports.org) for further information.

Thank you for giving us the opportunity to consider this manuscript. I regret that I could not offer more positive news at this time, and I hope that you consider our transfer offer.

Yours sincerely,

Cornelius Schneider

Cornelius Schneider, PhD
Editor
The EMBO Journal
c.schneider@embojournal.org

Referee #1:

In the manuscript entitled "The Hippo kinase LATS1 controls CTCF chromatin occupancy and the hormonal response of three-dimensionally grown breast cancer cells" the authors characterize the physical, biological, and genomic differences of ER+ T47D cells growing in 3D conditions and propose LATS1 kinase as a key driver of the architectural changes and increase expression of hormone regulated genes in ER+ breast cancer. Furthermore, these studies suggest the 3D platform as a finer tune system to investigate response to hormonal cues. Some additional analysis and experiments could be performed to support and further validate their findings.

Major comments.

- 1- The authors suggest that cell identity is maintained in cell grown in 3D with subtle differences in chromatin architecture and super enhancer landscape. To elucidate the main differences the authors should discuss the identified genes regulated by unique super-enhancers as well as those transitioning from compartment A to B.
- 2- The authors found that "changes in CTCF does not always lead to changes in TADs boundaries" which agrees with other reports suggesting that ~20% of TAD boundaries remain stable upon loss of CTCF. Changes in other proteins such as Cohesin and Cohesin loading factors Nipbl results in TAD disruption. The authors should perform ChIP-seq for Cohesin subunits to study in greater the effect in TAD disruption between 2D and 3D.
- 3- The ATAC-seq experiments showed reduced chromatin accessibility between 3D and 2D, but analyses are underdeveloped. Further analyses such as 1) transcription factor motif analysis, 2) identification of the genes that lost/gain accessible regions in 3D compared to 2D would improve and 3) their gene expression profile should be included.
- 4- Figure 5B-C aim to provide a link between inhibition of LATS1 kinase activity and CTCF binding to chromatin. Figure 5B (right panel) clearly showed the recovered binding of CTCF by more than 2-fold increase after treatment with TRULI. However, in panel C, after knockdown of the protein with two different shRNA (equally effective reducing LATS1 protein levels) the increase in CTCF binding is more modest and not significant in 2 of the 3 sites for the shLATS1. Do the authors consider the contribution of LATS2? LATS1/2 share 85% homology in the kinase domain and TRULI inhibit the activity of both LATS1/2.

- 5- A major finding from this paper suggest that LATS phosphorylates CTCF and induce displacement from chromatin in 3D grown cells. While inhibition or depletion of LATS restores CTCF binding to target genes, a formal demonstration of direct phosphorylation is lacking. The authors couldn't detect the changes in CTCF proliferation after inhibition of LATS kinase activity with TRULI. A formal prove that LATS phosphorylates CTCF in 3D should be provided.
- 6- To increase the robustness of the findings, some of the key experiments (i.e. Figure 5A-C, Figure 1C, D, E; Figure 2A) should be performed in another ER+ BC cell line.
- 7- Despite the success of CDK4/6 inhibition in combination with endocrine therapy, around 80% of patients develop resistance after 12 to 36 months. One of the proposed mechanisms is loss of the tumor suppressor FAT1 and activation of the hippo pathway and YAP activity¹. Can the authors comment on the progesterone receptor status of these patients and whether your results and model system could be useful finding new therapeutic alternatives for these patients?

Minor comments:

- 1- In figure 1C the authors showed the increased proliferative capacity of T47D cells growing in 3D conditions respect to 2D. However, the GSEA presented in figure 1F showed a decreased expression in the 3D conditions of the proliferative signaling pathways such as MAPK, RAS, estrogen signaling as well as the hippo pathway. On the other hand, figure EV1B shows increase in ER regulated gene expression such as CXCL12. This is a bit confusing. I believe the figure should be log₂FC 2D vs 3D (Figure 1F, middle panel) instead of log₂FC 3D vs 2D.
- 2- While ChIP-qPCR results from Figure 5B showed recovered binding of CTCF after TRULI treatment, the WB results are not very clear considering that total amount of LATS is also less in the 3D conditions + TRULI.
- 3- The rational for using T47D cells a main model system is unclear (end of first paragraph of results).
- 4- No data is provided on the average number of cells (120) in 3D after 10 days.
- 5- IFs from Figure 1D are not clear.
- 6- This reviewer does not agree with the claim that H3K27AC and H3K18ac don't change in 3D versus 2D. Indeed, the presented venn diagram argues that there are big changes.

REFERENCE

1. Li, Z. et al. Loss of the FAT1 Tumor Suppressor Promotes Resistance to CDK4/6 Inhibitors via the Hippo Pathway. *Cancer Cell* 34, 893-905 e8 (2018).

Referee #2:

The manuscript describes differences in breast cancer cells grown in 2D culture conditions versus 3D spheroids. Authors report differences in cellular parameters, including nuclear size, as well as an array of epigenetic changes. Patterns of gene expression are dramatically different between the two conditions. Among these differences is the activation of the Hippo pathway and the activation of the LATS1 kinase in cells grown as spheroids. Interestingly, the authors show that activation of LATS1 results in phosphorylation of CTCF, leading to a dramatic re-distribution of this protein and subsequent changes in 3D chromatin architecture.

The manuscript reports very interesting observations for both the breast cancer field as well as for those interested in basic biology of nuclear organization. Little is known about how CTCF binding is regulated during cell differentiation and the finding of a direct involvement of LATS1 in this process connects signaling events in the cytoplasmic membrane to changes in 3D chromatin organization and gene expression for the first time to my knowledge. The manuscript is appropriate for publication in *EMBO J*. I only have a few mostly minor comments that the authors should address.

1. The authors find an increase in nuclear volume of T47D cells grown in 3D versus 2D, but a decrease in nuclear diameter. This is surprising and authors may want to try to explain the observation in more detail to avoid confusing the reader.
2. T47D grown in 3D cultures upregulate genes involved in neurogenesis. Does this mean that the cells differentiate into a neuronal lineage under these conditions? Do MCF10A express the same genes when grown in these conditions or is this a characteristic of tumor cells? Are the same genes involved in neurogenesis also active in actual breast tumors? The analysis described for the MOTERA genes does not seem to address this question.
3. Approximately 80% of superenhancers are the same in 3D and 2D T47D cultured cells, indicating that 20% are different. Authors may want to reconsider the conclusions from these findings. First, a difference of 20% could be very significant in this context. I, personally, would be dancing on the streets if I received a salary rise of 20%. Second, the conclusion that "cells basically maintain their cell identity in both conditions" seems premature. Expression of 4 genes is all that is required to make any cell of the body into an iPSC.
4. Figure 2A. Regions of the nucleus that stain more brightly with DAPI correspond to chromocenters formed by interactions between pericentromeric regions of individual chromosomes. Therefore, the observed alterations may reflect changes in the numbers of chromosomes that come together at centromeres and may be unrelated to changes in chromatin condensation.

5. Figure 2B. Earlier in the manuscript, the authors showed that the nuclei of 3D grown cells have a smaller volume, suggesting that the same chromatin in 2D and 3D cells is packed in a smaller space. It would be interesting to consider this in the context of the MNase results. Keeping in mind that levels of H3K9me3 are similar between 2D and 3D cells, could the lower accessibility to MNase reflect the inability of the enzyme to reach parts of the nucleus, rather than changes in chromatin condensation at the nucleosome level?

6. Figures 3A and 3B. If, by definition, the A compartment is active chromatin and the B compartment is inactive chromatin, changes in gene expression should be reflected in compartmental changes. One explanation for why this would not happen is that the compartments are defined at low resolution, 1 Mb in Figure 3A. Figure 3B probably better reflects reality and I would delete Figure 2A from the manuscript if I were the authors. Analyzing compartments at a resolution higher than 100 kb would involve deeper sequencing of the Hi-C libraries, which is probably not worth the expense required. Authors may be able to map smaller compartments with their existing data using POSSUMM (PMID 37280210), although this is not critical to support the conclusions of this part of the manuscript.

7. The statement in page 9 "almost all TADs are bookmarked by CTCF (Dixon et al., 2012), suggesting an important functional role in TAD formation and maintenance" is not exactly correct. Instead, Dixon et al concluded in their 2012 manuscript that "We have identified multiple factors that are associated with the boundary regions separating topological domains, including the insulator binding factor CTCF, housekeeping genes and SINE elements. The association of housekeeping genes with boundary regions extends previous studies in yeast and insects and suggests that non-CTCF factors may also be involved in insulator/barrier functions in mammalian cells". This could explain why, in some cases (Appendix Figure S4E), loss of CTCF does not impact TAD structure.

8. Figures 4E and 4F. Authors may want to use a larger font in the interaction scales, which are very difficult to read.

9. The recovery of CTCF binding after incubation of 3D cells with TRULI is a really nice result. Is it possible that the same treatment also increases the ATAC-seq peaks in 3D cells to a number similar to that found in 2D cells?

Referee #3:

In 3D-grown cells the authors clearly show that, as perhaps is to be expected, the presence of a less stiff environment is sensed, integrated, and transduced leading to repression of the focal adhesion pathway and activation of the Hippo pathway. They conclude that altered levels of nuclear YAP, via the activity of the LATS1 kinase, enhances progesterone-induced progesterone receptor binding in 3D cells. LATS kinases are already known to be modulators of nuclear hormone receptors. LATS1 is also suggested to phosphorylate CTCF, leading to a massive loss of binding and consequent changes in TADs. This aspect, though intriguing, is underdeveloped.

The manuscript doesn't hang together as a whole. The later part examining the effect of LATS1 and YAP on gene regulation, especially by the progesterone receptor, is much better than the earlier results (Figs 1-4). The earlier part of the manuscript presents a series of correlative general assays of chromatin structure. These are often mis-interpreted, or not correctly controlled - for example the authors present ATAC-seq data as a confirmation of differences in MNase sensitivity of 2D and 3D grown cells. These two assays probe very different aspects of chromatin structure. MNase detects the general compaction of large-scale chromatin fibres - for example linker histone density or the presence of constitutive heterochromatin. ATAC-seq measures the focal loss of specific nucleosomes at sites of transcription factor binding. The authors also frequently confuse correlation with causation.

I found having additional data presented in either Appendix Supplementary Figures and EV Figures extremely confusing to follow.

Specific points:

Fig S1B. The data points, and/or data distribution for these graphs should be shown (e.g. violin plots). How many biological replicates of these data were performed, the number of data points, and details of the statistical tests used also need to be described.

Fig S2A. The altered cell cycle of cells grown in 2D and 3D needs to be better quantified - i.e. by FACs. This will put the changes in nuclear parameters into a better perspective.

Fig 1F The GO terms in EV1 data makes sense, but many of the KEGG pathways in Fig 1F less so - the current discussion cherry picks only those pathways that fit the authors hypothesis.

Pg6. Bottom of pg 6. H3K27ac and H3K18ac histone modifications are NOT epigenetic. They are unstable marks that track with gene expression. Profiling histone marks also does not show that the gene expression changes are "due" to histone

modifications - it just determines if there is a correlation. The authors should be more precise in the terms they use.

EV2A. The small quantitative changes in H3K27me3 for 3D down genes is unconvincing. What does a small change in H3K37me3 for down-regulated genes mean? Polycomb (H3K27me3) is digital not analogue - genes are either polycomb coated or they are not. I could not find details of the number of biological replicates performed for ChIP-seq, how replicate data were handled statistically, nor the data deposition details (Accession numbers for GEO).

Figure 2. MNase and ATAC-seq are mis-interpreted - they probe very different aspects of chromatin structure. MNase detects the general compaction of large-scale chromatin fibres - e.g. linker histone density or the presence of constitutive heterochromatin. ATAC-seq measures the focal loss of specific nucleosomes at sites of sequence-specific transcription factor binding; the nucleosome-depleted region at transcription start sites and at CTCF sites. As for ChIP-seq, no details are given of how many biological replicate ATAC-seq datasets were generated, how they were treated statistically, and the public Accession numbers.

Figure 3 and pg 9. The authors confuse correlation and causation in considering Hi-C compartments, stating that "These findings suggest that in 3D condition there are gradual transitions between the A and B compartments that sustain the changes detected in gene expression". It is gene expression and chromatin states such as H3K27me3 and H3K9me3 domains that determine A and B compartments, not the other way around.

Figure 4. 75% loss CTCF binding across the genome is implausible unless there is a very widespread increase in DNA methylation to block CTCF binding, or as the authors go on to suggest, massive post-translational modification of CTCF that prevents its binding. The authors should have used a spike in control in their ChIP-sequencing to be confident of reduced CTCF binding genome-wide. Otherwise, this result could simply reflect technical issues - e.g. with chromatin solubility, cross-linking or fragmentation. Based on the data in Figs 5 and 6, the authors should show whether the ATAC-seq peaks that correspond to the nucleosome depleted region underneath the sites of bound CTCF are particular affected.

Figure 5. The authors present data in Fig 5A suggesting that all CTCF in the 3D cells is phosphorylated. There is no control for a protein that they do not expect to be phosphorylated by LATS1. CTCF-ChIP-seq +/-the LATS inhibitor should similarly be controlled with a spike in control (such as mouse chromatin). This is important given that the authors state that the effect of a LATS1 inhibitor on CTCF phosphorylation could not be detected. A massive loss of CTCF across the genome, as a consequence of LATS1-mediated phosphorylation, seems at odds with the site-selective loss of CTCF as a result of LATS-mediated phosphorylation, reported by Luo et al., 2020. This aspect of the manuscript is under-developed - e.g. the authors should transfect into their cells CTCF with mutations at T374 and S402 that block this phosphorylation. This should restore binding and 3D genome organisation.

** As a service to authors, EMBO Press provides authors with the possibility to transfer a manuscript that one journal cannot offer to publish to another EMBO publication or the open access journal Life Science Alliance launched in partnership between EMBO Press, Rockefeller University Press and Cold Spring Harbor Laboratory Press. The full manuscript and if applicable, reviewers' reports, are automatically sent to the receiving journal to allow for fast handling and a prompt decision on your manuscript. For more details of this service, and to transfer your manuscript please click on Link Not Available. **

Point by point response to the reviewers

Referee #1 (Report for Author)

In the manuscript entitled "The Hippo kinase LATS1 controls CTCF chromatin occupancy and the hormonal response of three-dimensionally grown breast cancer cells" the authors characterize the physical, biological, and genomic differences of ER+ T47D cells growing in 3D conditions and propose LATS1 kinase as a key driver of the architectural changes and increase expression of hormone regulated genes in ER+ breast cancer. Furthermore, these studies suggest the 3D platform as a finer tune system to investigate response to hormonal cues. Some additional analysis and experiments could be performed to support and further validate their findings.

Major comments.

1. The authors suggest that cell identity is maintained in cell grown in 3D with subtle differences in chromatin architecture and super enhancer landscape. To elucidate the main differences the authors should discuss the identified genes regulated by unique super-enhancers as well as those transitioning from compartment A to B.

R: As proposed by the reviewer, we carry out a more comprehensive analysis of the Super-Enhancers (SEs). Our analysis uncovers that a majority of SEs exhibit persistence across both the 2D and 3D conditions (EV2D). However, a closer examination using a proximity-based script (Hnisz et al., 2013) of the specific genes under the regulation of these SEs, reveals that there are 129 genes regulated by specific SEs in 2D, while 95 genes are exclusively regulated under the 3D condition (new Figure EV3A).

The 2D-exclusive genes are related to terms like abnormal embryo size, abnormal development (body size), and hematopoiesis, which are categories closely linked to the Hippo pathway (as reported in Yu et al., 2015 (see new Figure EV3C). Conversely, in the case of genes exclusively regulated in the 3D condition by SEs, their numbers are smaller, and many of them appear to be artifacts, as illustrated with the HSPA8 gene (see new Figure EV3B). This hampers the identification of significant associated terms to this category.

Furthermore, the number of SEs that undergo a transition from state A to state B in the 3D condition, is quite limited. This constraint makes it challenging to extract comprehensive information regarding the pathways and biological processes involved. In the new version of the manuscript, we have included snapshots from the genome browser for two 3D down-regulated genes, SLC26A7 and RUNX1T1, and two 3D up-regulated genes, PI15 and ITGAL (see new Figure EV3C). These examples highlight the most noticeable transition between the 2D and 3D for these SEs, involving a change from state A to less A or towards an emergent state B. (see new Figure EV3C)

In the new version of the manuscript, new text in the discussion section was added (Page 8, line 31):

"To address this point, by using a proximity-based script (Hnisz et al., 2013), we found 129 and 95 genes associated to 2D and 3D SEs, respectively (Figure EV3A). The 2D-exclusive genes are related to terms like abnormal embryo size, abnormal development, and hematopoiesis, which are closely linked to the Hippo pathway (as reported in (Yu et al., 2015)) (Figure EV3C). Conversely, in the case of genes exclusively regulated in the 3D condition by SEs, many of them appear to be artifacts, as illustrated with the HSPA8 gene (Figure EV3B). This hampers the identification of significant associated terms.

When attempting to establish a connection between super enhancers and alterations in genome structure, such as the A to B transition, it becomes evident that the quantity of SEs is reduced, and this transformation exhibits subtler changes, predominantly shifting from A to less degree of A, or toward emerging B, and vice versa (Figure EV3D). Snapshots from the genome browser illustrating these transitions for two 3D down-regulated genes, SLC26A7 and RUNX1T1, and two 3D up-regulated genes, PI15 and ITGAL are shown (Figure EV3D)."

2. The authors found that "changes in CTCF does not always lead to changes in TADs boundaries" which agrees with other reports suggesting that ~20% of TAD boundaries remain stable upon loss of CTCF. Changes in other proteins such as Cohesin and Cohesin loading factors Nipbl results in TAD disruption. The authors should perform ChIP-seq for Cohesin subunits to study in greater the effect in TAD disruption between 2D and 3D.

R: We appreciate the reviewer's comment on this aspect. Based on our current results (Figure 3A, D, and E), we have observed that overall, the 2D and 3D TADs remain largely unchanged, with approximately 90% of TADs showing no significant differences. The remaining 10% exhibit moderate changes, primarily displaying a trend towards fusion in 3D. This lack of significant change in TADs can be attributed to the persistence of CTCF and/or the Cohesin complex binding in both conditions.

To address the point raised by the reviewer, we have performed ChIP-seq experiments for RAD21 (a component of the Cohesin complex) in 2D and 3D conditions, and our results showed no significant differences in RAD21 content between fused and random TADs. We have included these findings in new

Appendix Figure S5 and we cautiously discussed these new results and the potential involvement of Cohesin components in our systems.

We've added the new results as new Appendix Figure S5, and their description as a new paragraph (**Page 12, line 12**). See below:

“To assess whether changes in the 3D Topologically Associated Domains (TADs) are attributable to differential loading of cohesin components in these regions, we conducted RAD21 ChIP-seq experiments under 2D and 3D growth conditions. Our findings indicate that there are no significant alterations in RAD21 occupancy within both fused and random TADs (Appendix Figure S5)”

3. The ATAC-seq experiments showed reduced chromatin accessibility between 3D and 2D, but analyses are underdeveloped. Further analyses such as 1) transcription factor motif analysis, 2) identification of the genes that lost/gain accessible regions in 3D compared to 2D would improve and 3) their gene expression profile should be included.

R: *As recommended by the reviewer, we conducted a more in-depth analysis of the ATAC-seq assays. We identified the regions that become more accessible in 2D and 3D as well as the DNA binding motifs (new EV4A-B). Our results suggest that the sites that become more accessible in 2D and 3D are enriched in CTCF, TEAD, FOXA1 and PR motifs (new Figure EV4B). However, when we cross-checked our data with ChIP-seq database (Zheng et al., 2019), our word cloud plots (new EV4B-C, lower panels) give a different result: the sites that open in 3D are enriched in the estrogen pathway through the estrogen receptor (ER) itself, as well as its pioneer factors FOXA1 and GATA3; the cofactor GREB1 and the coactivator P300, which could explain the increased proliferation observed in 3D cells (Figure 1C). In contrast, the sites that become closed in 3D (2D-exclusive) are enriched in CTCF, an architectural factor that we have shown to be displaced when we grow the cells as spheroids (new Figure 4A). This would imply that in 3D a more estrogenic program is turned on and CTCF is displaced, whether these two events are connected require further investigation. As also recommended by the reviewer, we analyze the chromatin accessibility of 2D and 3D-exclusive genes. We found that the gene body of those genes activated in 2D are more accessible in 2D than in their 3D counterpart; surprisingly, the 3D-up genes failed to show increased accessibility in 3D condition (new EV4D). However, we found increased accessibility in distantly located regulatory regions (enhancers) associated to 166 genes up-regulated in 3D. To illustrate this finding, the TN2Cx and PTPRD genes are shown.*

In the new version of the manuscript we have discussed these new results (page 10, line 15 and new EV4E).

“To characterize the regions that become more accessible in 2D and 3D we conducted a more in-depth analysis of the ATAC-seq (EV4A-B). Our results suggest that the sites that become more accessible in 2D and 3D are enriched in CTCF, TEAD, FOXA1 and PR motifs (EV4B). However, when we combined our analysis with available ChIP-seq data (Zheng et al., 2019), word cloud plots (EV4B-C, lower panels) indicated that open sites in 3D are enriched in the estrogen pathway through the ER itself, alongside pioneer factors FOXA1 and GATA3; the cofactor GREB1 and the coactivator P300, which could explain the increased proliferation observed in 3D cells (Figure 1C). In contrast, the sites that become closed in 3D (2D-exclusive) are enriched in CTCF, an architectural factor that we have shown to be displaced when we grow the cells as spheroids (Figure 4A). This would imply that in 3D cells a more estrogenic program is turned on along CTCF is displaced, if these two events are connected require further investigation.

Furthermore, the chromatin accessibility of 2D and 3D-exclusive genes was also assayed, and we detected changes. As expected, 2D-up genes exhibit greater accessibility within the gene body in 2D condition compared to their 3D counterpart, whereas in 3D condition the expected inverse trend is not observed (EV4D). However, we found increased accessibility in distantly located regulatory regions (enhancers) associated to 166 genes up-regulated in 3D. To illustrate this finding, the TN2Cx and PTPRD genes are shown (EV4E).”

4. Figure 5B-C aim to provide a link between inhibition of LATS1 kinase activity and CTCF binding to chromatin. Figure 5B (right panel) clearly showed the recovered binding of CTCF by more than 2-fold increase after treatment with TRULI. However, in panel C, after knockdown of the protein with two different shRNA (equally effective reducing LATS1 protein levels) the increase in CTCF binding is more modest and not significant in 2 of the 3 sites for the shLATS1. Do the authors consider the contribution of LATS2? LATS1/2 share 85% homology in the kinase domain and TRULI inhibit the activity of both LATS1/2.

R: *To address the point highlighted by the reviewer, we used a specific antibody that recognizes phosphorylation at T1079 and T1041 present in both LATS1 and LATS2 (PA5-117227 from Thermo Fisher), and performed blots on control and LATS1-depleted cells (shControl and shLATS1, respectively). The dual phospho-LATS1/2 signal in 3D cells was significantly reduced when LATS1 is absent. Therefore, we*

conclude that, although possibly not entirely, the majority of LATS activity in our system corresponds to LATS1 (Figure for the reviewer, see below).

We incorporate the new results in Figure EV5D and we added the paragraph below in the new version of the manuscript (page 15, line 7):

“The potential involvement of LATS2 was assessed using a specific antibody that recognizes phosphorylation at T1079 and T1041 present in both LATS1 and LATS2, in wild-type cells and LATS1-depleted cells. The phosphorylation signal in 3D cells decreased proportionally with the reduction of LATS1, supporting that LATS1 is the Hippo kinase involved in our system (EV5C).

5. A major finding from this paper suggest that LATS phosphorylates CTCF and induce displacement from chromatin in 3D grown cells. While inhibition or depletion of LATS restores CTCF binding to target genes, a formal demonstration of direct phosphorylation is lacking. The authors couldn't detect the changes in CTCF proliferation after inhibition of LATS kinase activity with TRULI. A formal prove that LATS phosphorylates CTCF in 3D should be provided.

R: To answer the point raised by this reviewer and to add a formal proof that LATS1 phosphorylates CTCF in 3D cells, we performed immunoprecipitation of CTCF in 3D grown cells treated or not with TRULI and then probed with specific antibodies recognizing the phosphorylated RXXpS/T epitope found in CTCF T374 and S402 as used in Luo et al., Science Advances 2020. Our new results prove that the p-CTCF signal is reduced by 40% in the presence of the LATS inhibitor, TRULI (new Figure EV5A).

We've added the new results as Figure EV5A, and their description as a new paragraph (Page 14, line 10). See below:

To have a formal proof that LATS1 phosphorylates CTCF in 3D cells, we performed immunoprecipitation of CTCF in 3D-grown cells treated or not with TRULI and then probed with specific antibodies recognizing the phosphorylated RXXpS/T residues that match the changes observed in T374 and S402 as previously described (Luo et al., 2020). Our results showed that the phospho-CTCF signal in 3D is reduced by 40% in the presence of TRULI, confirming the raised hypothesis (EV5A).

6. To increase the robustness of the findings, some of the key experiments (i.e. Figure 5A-C, Figure 1C, D, E; Figure 2A) should be performed in another ER+ BC cell line.

R: As suggested by the reviewer, we repeated the assays in the ER+ MCF-7 cell line. In MCF-7, although the total levels of YAP are significantly lower than in T47D, we observed an increase in LATS phosphorylation in the 3D context compared to 2D (new Appendix Figure S1C).

We've added the new results as new panel in Appendix Figure S1C, and their description as a new paragraph (Page 6, line 11). See below:

“Increased levels of phospho-LATS (part of the Hippo pathway) in response to 3D cell growth was also observed in ER+ MCF-7 cells. However, this activation was less pronounced in comparison to T47D cells (Appendix Figure S1C compared to Figure 5B). Notably, there was no significant increase in p-YAP or decrease in p-FAK in MCF-7 cells, where YAP expression notably remained low in comparison to other mammary cell lines (Appendix Figure S1C). This observation indicates a partial conservation of the cytosolic activation of the Hippo pathway in response to 3D in MCF-7 cells. (Appendix Figure S1C). Additionally, it is worth mentioning that reduced ER levels in MCF-7 3D cells (Appendix Figure S1C) had an impact on their proliferative capacity (data not shown).”

7. Despite the success of CDK4/6 inhibition in combination with endocrine therapy, around 80% of patients develop resistance after 12 to 36 months. One of the proposed mechanisms is loss of the tumor suppressor FAT1 and activation of the hippo pathway and YAP activity¹. Can the authors comment on the progesterone receptor status of these patients and whether your results and model system could be useful finding new therapeutic alternatives for these patients?

R: In response to the reviewer's query, we compared gene expression data from Li et al.'s study (Cancer Cell 2018) with our 3D breast cancer cell model. We found that cells resistant to CDK4/6 and endocrine therapy exhibit low levels of FAT1 and activation of the Hippo pathway (Li et al., 2018), consistent with 3D-grown cells (new Appendix Figure S12B). Additionally, PR status was lower in resistant cells (new Appendix Figure S12A), resulting in reduced expression of progestin-dependent genes for both activation and inhibition, as depicted in new Appendix Figure S12D. Thus, while our 3D model partially recapitulates resistant cell characteristics, it doesn't fully replicate the reduced hormonal response observed in FAT1 knockout cells.

We incorporate the following text in the Discussion section (**page 23, line 34**):

“To assess whether our 3D system recapitulate the resistance of breast cancer cells to CDK4/6 inhibitors and endocrine therapy, which is associated with low FAT1 levels and Hippo pathway activation, we conducted a comparative analysis of gene expression profiles of 3D T47D cells with those from a previous study (Li et al., 2018). Our analysis revealed that cells resistant to CDK4/6 inhibitors and endocrine therapy exhibited reduced levels of FAT1 and demonstrated activation of the Hippo pathway, which matched the characteristics observed in our 3D-grown cells (as depicted in Appendix Figure S12). However, the resistant cells had lower progesterone receptor (PR) status (Appendix Figure S12A), leading to decreased expression of genes controlled by progesterin, both in terms of activation and repression (Appendix Figure S12D). Thus, although our 3D model partially replicated the characteristics of resistant cells, it did not entirely reproduce the diminished hormonal response observed in cells lacking FAT1 expression (Appendix Figure S12E). In fact, in our 3D model, this response was enhanced (Figure 6A)”

1. Li, Z. et al. Loss of the FAT1 Tumor Suppressor Promotes Resistance to CDK4/6 Inhibitors via the Hippo Pathway. *Cancer Cell* 34, 893-905 e8 (2018).

Minor comments:

1. In figure 1C the authors showed the increased proliferative capacity of T47D cells growing in 3D conditions respect to 2D. However, the GSEA presented in figure 1F showed a decreased expression in the 3D conditions of the proliferative signaling pathways such as MAPK, RAS, estrogen signaling as well as the hippo pathway. On the other hand, figure EV1B shows increase in ER regulated gene expression such as CXCL12. This is a bit confusing. I believe the figure should be log2FC 2D vs 3D (Figure 1F, middle panel) instead of log2FC 3D vs 2D.

R: The reviewer was correct and the KEGG data in Figure 1F were not properly statistically controlled; thus, we have repeated the analyses in the new version of our manuscript. On the one hand, the up genes in 3D are enriched in Ribosome, Mineral Absorption and Amino acid Metabolism terms (new Figure 1F, right panel). We have discussed these new results in the context of the effect of environmental stiffness on these parameters. On the other hand, the repressed genes show an enrichment in genes involved in Transcriptional deregulation in Cancer and the Hippo and Apelin pathways, among others (new Figure 1F, right panel). The understanding of the Apelin pathway's role in breast cancer is limited, with only one report linking high levels of Apelin to postmenopausal breast cancer (Salman et al., 2016); but it is clear that genes involved in vasodilation and muscle contraction are repressed, which could be explained as an adaptation to a softer environment found in 3D. The repression of these pathways could also be linked to the previously mentioned Mineral Absorption category found in 3D up genes, as Ca²⁺ can regulate vascular smooth muscle cells contractility (Brozovich et al., 2016).

We incorporated the new Figure 1F (right panel) and we added the paragraph below in the new version of the manuscript (**page 6, line 35**):

“Further analysis of KEGG exclusive 3D pathways showed terms related to Ribosome, Mineral Absorption and Amino acid Metabolism for 3D up-genes (Figure 1F, right panel). The physical characteristics of the surrounding structure play a pivotal role in coordinating this regulatory interplay. Concerning Amino acid Metabolism, the stiffness of the extracellular matrix regulates the activation of the Hippo pathway and, consequently, the nuclear localization of YAP/TAZ. YAP/TAZ play a critical role in controlling genes involved in amino acid synthesis, transport and metabolism (Ge et al., 2021). Therefore, in response to the mechanical properties of their environment, cells adjust amino acid metabolism to acquire the energy necessary for specific cellular functions, such as enhanced cell proliferation, as illustrated in Figure 1C. Extracellular matrix stiffness also influences mineral absorption, as previously documented (Derricks et al., 2015). Cells on softer substrates (4 kPa) demonstrate increased responsiveness to VEGF and Ca²⁺ dynamics, while cells on stiffer substrates (125 kPa) exhibit a diminished response.

The enrichment of genes related to Ribosomes suggests that protein synthesis is differentially regulated in 3D environments. This may be attributed to the unique demand for specialized proteins involved in cell-cell interactions, the integration of external signals, and mechano-transduction, especially in cells growing as spheres with low stiffness. In summary, the regulation of genes associated with Amino acid Metabolism, Mineral Absorption, and Ribosomes in 3D environments underscores the significant influence of extracellular matrix stiffness on various cellular processes. This intricate interplay allows cells to adapt to their mechanical surroundings, ultimately affecting crucial aspects of cell biology and physiology.

The 3D-repressed genes exhibit an enrichment in genes associated with Transcriptional deregulation in Cancer, the Hippo and Apelin pathways, among others. The Apelin pathway is not well-documented in breast cancer, with only one report associating high levels of Apelin with postmenopausal breast cancer (Salman et al., 2016). However, genes involved in vasodilation and muscle contraction are significantly repressed, which could be explained as an adaptation to the softer 3D environment. The suppression of these pathways could also be linked to the up-regulation of Mineral Absorption pathway, as calcium ions (Ca²⁺) can regulate the contractility of vascular smooth muscle cells (Brozovich et al., 2016)”

2. While ChIP-qPCR results from Figure 5B showed recovered binding of CTCF after TRULI treatment, the WB results are not very clear considering that total amount of LATS is also less in the 3D conditions + TRULI.

R: We agree with the reviewer, and we have replaced phospho-LATS with new phospho-YAP blots, which is a more indicative reporter of TRULI-mediated inhibition of the Hippo pathway (Kastan et al., 2021). According to Kastan et al.'s original paper (2021), TRULI impedes YAP phosphorylation at S127 without impacting on LATS phosphorylation in MCF10A cells (Figure 3 in Kastan et al., 2021). While observed variations might be attributed to the cell type used, we decided to employ a more reliable indicator of LATS activity, namely phosphorylated YAP. Consequently, we have replaced the initial blots in Figure 5B with new ones to demonstrate TRULI's effect on YAP phosphorylation in T47D cells.

3. The rationale for using T47D cells as a main model system is unclear (end of first paragraph of results).

R: To address this point, we incorporate the following text in the Results section (**page 5, line 12**):

"In this study, our primary focus was on PR+ T47D cells as a model system. These cells are known for their robust responsiveness to the hormone progesterone, and their genomic structure, the PR cisome, and the associated signaling pathway network have been extensively characterized (Ballare et al., 2013; Le Dily et al., 2014; Le Dily et al., 2019; Vicent et al., 2006; Vicent et al., 2010; Zaurin et al., 2021)"

4. No data is provided on the average number of cells (120) in 3D after 10 days.

R: We provide the data requested by the reviewer (**page 5, line 29**)

Within 10 days of culture the 3D cells formed spheroids with an average diameter of 100 μ m, typically comprising 119 \pm 12 cells (Figure 1C, right panel).

5. IFs from Figure 1D are not clear.

R: As recommended by the reviewer, we improved the quality of the IFs in Figure 1D.

6. This reviewer does not agree with the claim that H3K27Ac and H3K18ac don't change in 3D versus 2D. Indeed, the presented venn diagram argues that there are big changes.

R: We agree with the reviewer, and we changed the paragraph describing the possible role of K27 and K18 acetylation (**page 7, line 34**) to:

"In general, a notable variation, ranging from 20% to 44%, was observed in the two marks when cells transitioned from 2D to 3D culture conditions (Figure EV1A-B). These alterations were particularly concentrated in genes influenced by the growth environment. Consequently, a substantial reduction in H3K27ac was detected for genes down-regulated and up-regulated in 3D (Appendix Figure S3C), along with an increase in H3K18ac for genes up-regulated in 3D (Appendix Figure S3D, right panel), as compared to their 2D counterparts"

REFERENCE

1. Li, Z. et al. Loss of the FAT1 Tumor Suppressor Promotes Resistance to CDK4/6 Inhibitors via the Hippo Pathway. *Cancer Cell* 34, 893-905 e8 (2018).

Referee #2 (Report for Author)

The manuscript describes differences in breast cancer cells grown in 2D culture conditions versus 3D spheroids. Authors report differences in cellular parameters, including nuclear size, as well as an array of epigenetic changes. Patterns of gene expression are dramatically different between the two conditions. Among these differences is the activation of the Hippo pathway and the activation of the LATS1 kinase in cells grown as spheroids. Interestingly, the authors show that activation of LATS1 results in phosphorylation of CTCF, leading to a dramatic re-distribution of this protein and subsequent changes in 3D chromatin architecture.

The manuscript reports very interesting observations for both the breast cancer field as well as for those interested in basic biology of nuclear organization. Little is known about how CTCF binding is regulated during cell differentiation and the finding of a direct involvement of LATS1 in this process connects signaling events in the cytoplasmic membrane to changes in 3D chromatin organization and gene expression for the first time to my knowledge. The manuscript is appropriate for publication in EMBO J. I only have a few mostly minor comments that the authors should address.

1. The authors find an increase in nuclear volume of T47D cells grown in 3D versus 2D, but a decrease in nuclear diameter. This is surprising and authors may want to try to explain the observation in more detail to avoid confusing the reader.

R: To address this point, we incorporate the paragraph below to the Results section (**page 18, line 35**):
“Morphologically, T47D cells grown as spheroids in Matrigel, presented an increase in nuclear volume coupled with a reduction in nuclear diameter (Appendix Figure S1B). These seemingly conflicting alterations might be attributed to the underlying physicochemical modifications that dictate the irregular shapes of the cell nucleus. These processes encompass mechanical forces acting through microtubules, actin filaments, and the osmotic pressure within the cytoplasm (Kim et al., 2015)”

2. T47D grown in 3D cultures upregulate genes involved in neurogenesis. Does this mean that the cells differentiate into a neuronal lineage under these conditions? Do MCF10A express the same genes when grown in these conditions or is this a characteristic of tumor cells? Are the same genes involved in

neurogenesis also active in actual breast tumors? The analysis described for the MOTERA genes does not seem to address this question.

R: To address the interesting point raised by the reviewer, we conducted a comparative analysis with non-tumorigenic MCF10A cells cultured either as monolayer or as 3D spheres (Maguire et al., 2016). Upon examining the genes upregulated in the 3D environment in both cell lines and categorizing them by tissue-associated genes, we observed a consistent enrichment of brain-associated genes, followed by those associated with the lung, mammary gland, and blood (new Appendix Figure S11A). This suggests that the culture conditions may play a significant role in gene expression patterns, rather than being solely dependent on tumor or non-tumorous state.

However, upon closer examination of the expression of up-regulated genes in the 3D environment and their clustering, distinctive differences became evident. Notably, in cluster 2, we identified an exclusive overexpression of the term "nervous system development" in T47D cells compared to MCF10A cells (new Appendix Figure S11B). Additionally, we observed clear variations in the expression of genes linked to cell and focal adhesion, particularly in cluster 3, which turned out to be distinct between tumor and non-tumoral cells cultured in the 3D environment (new Appendix Figure S11B). Therefore, the distinct expression of neurogenesis-related genes in a 3D environment appears to be linked to a more cancerous behavior. However, further investigations, involving a larger variety of both tumor and normal cells, are needed to confirm this observation.

As proposed by the reviewer, we have tried to determine if these genes associated with neurogenesis/nervous system development were enriched in tumor samples. However, the heterogeneity of the samples and the different types of breast cancer (ER and HER2 status and their combinations) made these analyses yield inconclusive results. More experiments and computational analyses will be necessary to address this intriguing question highlighted by the reviewer.

We incorporated the following text to the new version of the manuscript (**page 19, line 35**):

“As T47D cells exhibit an upregulation of genes related to neurogenesis (EV1A), we sought to determine whether this phenomenon was exclusive to breast cancer cells. To address this, we conducted a comparative analysis using non-tumorigenic MCF10A cells cultured as monolayers and as 3D spheres (Maguire et al., 2016). When we examined the genes upregulated in the 3D environment in both cell lines and categorized them by tissue-associated genes, we consistently found an enrichment of brain-associated genes, followed by those associated with the lung, mammary gland, and blood (see Appendix Figure S11A). This suggests that culture conditions have a significant influence on gene expression patterns, rather than being solely dependent on tumor or non-tumorous state.

However, upon closer examination of the expression of up-regulated genes in the 3D environment and their clustering, distinctive differences emerged. Notably, in cluster 2, we identified an exclusive overexpression of the term "nervous system development" in T47D cells compared to MCF10A cells (see Appendix Figure S11B). Additionally, we observed significant variations in the expression of genes related to cell and focal adhesion, particularly in cluster 3, which turned out to be distinct between tumor and non-tumoral cells cultured in the 3D environment (see Appendix Figure S11B). Therefore, the distinct expression of neurogenesis-related genes in a 3D environment appears to be associated with a more cancerous behavior. However, further investigations involving a broader range of both tumor and normal cells are required to confirm this observation”

3. Approximately 80% of super-enhancers are the same in 3D and 2D T47D cultured cells, indicating that 20% are different. Authors may want to reconsider the conclusions from these findings. First, a difference of 20% could be very significant in this context. I, personally, would be dancing on the streets if I received a salary rise of 20%. Second, the conclusion that "cells basically maintain their cell identity in both conditions" seems premature. Expression of 4 genes is all that is required to make any cell of the body into an iPSC.

R: *We have rephrased the paragraph describing the connection between super-enhancers (SEs) and cell identity and we incorporated new analysis using a proximity-based script that connects 2D and 3D SEs with their predicted target genes (Hnisz et al., 2013) (page 8, line 24)*

"The overlap between the super-enhancers detected in 2D and 3D according to the H3K27ac signal (579 and 564, respectively) was close to 80% (Figure EV2C). This observation would support that cells fundamentally maintain their regulatory gene network in both of these environments (Figure EV2D). Nonetheless, it's essential to acknowledge that 20% of super-enhancers are exclusively activated in the 3D environment. The question of whether these variances have implications for the alteration of cell identity remains open.

To address this point, by using a proximity-based script (Hnisz et al., 2013), we found 129 and 95 genes associated to 2D and 3D SEs, respectively (Figure EV3A). The 2D-exclusive genes are related to terms like abnormal embryo size, abnormal development, and hematopoiesis, which are closely linked to the Hippo pathway (as reported in (Yu et al., 2015)) (Figure EV3A). Conversely, in the case of genes exclusively

regulated in the 3D condition by SEs, many of them appear to be artifacts, as illustrated with the HSPA8 gene (Figure EV3B). This hampers the identification of significant associated terms.

When attempting to establish a connection between super enhancers and alterations in genome structure, such as the A to B transition, it becomes evident that the quantity of SEs is reduced, and this transformation exhibits subtler changes, predominantly shifting from A to less degree of A, or toward emerging B, and vice versa (Figure EV3C). Snapshots from the genome browser illustrating these transitions for two 3D down-regulated genes, SLC26A7 and RUNX1T1, and two 3D up-regulated genes, PI15 and ITGAL are shown (Figure EV3C)."

4. Figure 2A. Regions of the nucleus that stain more brightly with DAPI correspond to chromocenters formed by interactions between pericentromeric regions of individual chromosomes. Therefore, the observed alterations may reflect changes in the numbers of chromosomes that come together at centromeres and may be unrelated to changes in chromatin condensation.

R: *While it's possible that a varying number of centromeres could aggregate in these cells (Figure 2A), we believe such an occurrence wouldn't necessarily result in increased DAPI brightness. Our reasoning is based on the notion that maintaining a consistent degree of compaction implies that an increased convergence of centromeres (or other genomic regions) wouldn't inherently translate into intensified foci. Rather, it would likely result in a broader area of DNA accumulation.*

What we've observed is that the brightest DAPI areas in 3D cultured cells exhibit higher intensity levels compared to the brightest areas in 2D cells. This suggests a higher chromatin content per area in these regions, leading us to conclude they correspond to chromatin condensation

5. Figure 2B. Earlier in the manuscript, the authors showed that the nuclei of 3D grown cells have a smaller volume, suggesting that the same chromatin in 2D and 3D cells is packed in a smaller space. It would be interesting to consider this in the context of the MNase results. Keeping in mind that levels of H3K9me3 are similar between 2D and 3D cells, could the lower accessibility to MNase reflect the inability of the enzyme to reach parts of the nucleus, rather than changes in chromatin condensation at the nucleosome level?

R: *Although surprising, what we observed is that the nuclei of 3D grown cells have larger volumes (Appendix Figure S1B) as well as increased sphericity and surface area, but they undergo decrease MNase cuts under the same incubation conditions than 2D cells. Therefore, we would tend to exclude the possibility that the enzyme could not reach its targets as the nuclear volumes are larger than in 2D cells. Thus, the data in Figures 2A and 2B suggested that it is the local condensation of chromatin that makes it less accessible to MNase rather than a global hindrance related to altered nuclear volumes.*

6. Figures 3A and 3B. If, by definition, the A compartment is active chromatin and the B compartment is inactive chromatin, changes in gene expression should be reflected in compartmental changes. One explanation for why this would not happen is that the compartments are defined at low resolution, 1 Mb in Figure 3A. Figure 3B probably better reflects reality and I would delete Figure 2A from the manuscript if I were the authors. Analyzing compartments at a resolution higher than 100 kb would involve deeper sequencing of the Hi-C libraries, which is probably not worth the expense required. Authors may be able to map smaller compartments with their existing data using POSSUMM (PMID 37280210), although this is not critical to support the conclusions of this part of the manuscript.

R: *As the reviewer suggested, we removed Figure 3A and we changed the rest of the text, figures and legends accordingly.*

7. The statement in page 9 "almost all TADs are bookmarked by CTCF (Dixon et al., 2012), suggesting an important functional role in TAD formation and maintenance" is not exactly correct. Instead, Dixon et al concluded in their 2012 manuscript that "We have identified multiple factors that are associated with the boundary regions separating topological domains, including the insulator binding factor CTCF, housekeeping genes and SINE elements. The association of housekeeping genes with boundary regions extends previous studies in yeast and insects and suggests that non-CTCF factors may also be involved in insulator/barrier functions in mammalian cells". This could explain why, in some cases (Appendix Figure S4E), loss of CTCF does not impact TAD structure.

R: *The reviewer is right, we changed the paragraph in Results section (page 11, line 22) to:*

"Since the discovery of TADs, it became clear that the boundary regions separating topological domains are enriched in CTCF along with the structural maintenance of chromosomes (SMC) cohesin complex, housekeeping genes and SINE elements (Barutcu et al., 2018; Dixon et al., 2012; Nora et al., 2012)"

and in the Discussion section (page 22, line 13):

"As we previously mentioned, the association of housekeeping genes with boundary regions extends previous studies in yeast and insects (Duan et al., 2010; Ulianov et al., 2016) and suggests that non-CTCF factors may also be involved in insulator/barrier functions in mammalian cells."

8. Figures 4E and 4F. Authors may want to use a larger font in the interaction scales, which are very difficult to read.

R: *As suggested, we changed the size of the font in Figures 4E and 4F.*

9. The recovery of CTCF binding after incubation of 3D cells with TRULI is a really nice result. Is it possible that the same treatment also increases the ATAC-seq peaks in 3D cells to a number similar to that found in 2D cells?

R: *The experiment suggested by the reviewer is indeed intriguing, but due to economic restrictions, we've had to prioritize the number of experiments involving massive sequencing. As the reviewer may have noticed, we have incorporated new genomic ChIP-seq data for RAD21 and CTCF with spike-in controls (new Appendix Figures S5 and S7). Regrettably, given these economic limitations and considering that this point is a minor concern for the reviewer, we were unable to carry out the ATAC-seq + TRULI analysis, as we believe it does not significantly impact the main message of our manuscript. Nonetheless, we conducted a more comprehensive analysis of the ATAC-seq assays, which included: i) an analysis of transcription factor motifs, ii) identification of genes with altered accessible regions in the 3D compared to the 2D context, and iii) an assessment of their gene expression profiles (new Figure EV4).*

Referee #3 (Report for Author)

In 3D-grown cells the authors clearly show that, as perhaps is to be expected, the presence of a less stiff environment is sensed, integrated, and transduced leading to repression of the focal adhesion pathway and activation of the Hippo pathway. They conclude that altered levels of nuclear YAP, via the activity of the LATS1 kinase, enhances progesterone-induced progesterone receptor binding in 3D cells. LATS kinases are already known to be modulators of nuclear hormone receptors. LATS1 is also suggested to phosphorylate CTCF, leading to a massive loss of binding and consequent changes in TADs. This aspect, though intriguing, is underdeveloped.

The manuscript doesn't hang together as a whole. The later part examining the effect of LATS1 and YAP on gene regulation, especially by the progesterone receptor, is much better than the earlier results (Figs 1-4). The earlier part of the manuscript presents a series of correlative general assays of chromatin structure. These are often mis-interpreted, or not correctly controlled - for example the authors present ATAC-seq data as a confirmation of differences in MNase sensitivity of 2D and 3D grown cells. These two assays probe very different aspects of chromatin structure. MNase detects the general compaction of large-scale chromatin fibres - for example linker histone density or the presence of constitutive heterochromatin. ATAC-seq measures the focal loss of specific nucleosomes at sites of transcription factor binding. The authors also frequently confuse correlation with causation.

I found having additional data presented in either Appendix Supplementary Figures and EV Figures extremely confusing to follow.

Specific points:

Fig S1B. The data points, and/or data distribution for these graphs should be shown (e.g. violin plots). How many biological replicates of these data were performed, the number of data points, and details of the statistical tests used also need to be described.

R: *As suggested by the reviewer the data in Appendix Figure S1B are now presented as violin plots. The number of biological replicates, data points as well as the statistical tests used are explained in the new version of Extended Bioinformatics methods. The raw sequencing data from this study (ChIP- seq, ATAC- seq, RNA-seq and Hi-C) were submitted to the NCBI Gene Expression Omnibus (GEO; <http://www.ncbi.nlm.nih.gov/geo/>) repository, and the accession number is GSE247777.*

Fig S2A. The altered cell cycle of cells grown in 2D and 3D needs to be better quantified - i.e. by FACs. This will put the changes in nuclear parameters into a better perspective.

R: *As suggested by the reviewer, we conducted the BrdU staining protocol followed by flow cytometry to determine cell cycle distribution pattern in 2D and 3D-grown T47D cells. It's important to note that isolating the spheroids from matrigel and dispersing the cell clusters in 3D to prevent FACS blockage, presented significant challenges.*

Although we were able to isolate the cells from spheroids, both the cell number (4.7-fold less in 3D) and the BrdU staining profile (see Figure for reviewer below) turned out to be different between both conditions, most

likely due to the stress to which the 3D cells were subjected. Specifically, in the 3D cell setting, only 2100 events can be reliably monitored in the flow cytometry analysis, contrasting the standard 10,000 events typically observed.

Under these sub-optimal conditions, no difference in the percentage of cells in S-phase between 2D and 3D were found (approximately 11% for both conditions) (see Figure for reviewer below). Therefore, although we agree with the reviewer that a direct cell cycle assay in both conditions would complete the cell viability results (Figure 1C), our current data doesn't possess the level of robustness required for its incorporation into the revised manuscript. If the reviewer considers that this result should be incorporated or discussed in the text, we are willing to do so.

Fig 1F The GO terms in EV1 data makes sense, but many of the KEGG pathways in Fig 1F less so - the current discussion cherry picks only those pathways that fit the authors hypothesis.

R: *The reviewer was correct, and it appears that the appropriate statistical stringency was not applied to the KEGG data presented in the previous version of our manuscript; thus, we have repeated the analyses. The new results showed that, on one hand, the up genes in 3D are enriched in Ribosome, Mineral Absorption and Amino acid Metabolism terms (new Figure 1F, right panel). We have discussed these new results in the context of the effect of environmental stiffness on these parameters. On the other hand, the repressed genes showed an enrichment in genes involved in Transcriptional deregulation in Cancer and the Hippo and Apelin pathways, among others (new Figure 1F, right panel). The understanding of the Apelin pathway's role in breast cancer is limited, with only one report linking high levels of Apelin to postmenopausal breast cancer (Salman et al., 2016); but it is clear that genes involved in vasodilation and muscle contraction are repressed, which could be explained as an adaptation to a softer environment found in 3D. The repression of these pathways could also be linked to the previously mentioned Mineral Absorption category found in 3D up genes, as Ca²⁺ can regulate vascular smooth muscle cells contractility (Brozovich et al., 2016).*

We incorporated the new Figure 1F (right panel) and we added the paragraph below in the new version of the manuscript (page 6, line 35):

“Further analysis of KEGG exclusive 3D pathways showed terms related to Ribosome, Mineral Absorption and Amino acid Metabolism for 3D up-genes (Figure 1F, right panel). The physical characteristics of the surrounding structure play a pivotal role in coordinating this regulatory interplay. Concerning Amino acid Metabolism, the stiffness of the extracellular matrix regulates the activation of the Hippo pathway and, consequently, the nuclear localization of YAP/TAZ. YAP/TAZ play a critical role in controlling genes involved in amino acid synthesis, transport and metabolism (Ge et al., 2021). Therefore, in response to the mechanical properties of their environment, cells adjust amino acid metabolism to acquire the energy necessary for specific cellular functions, such as enhanced cell proliferation, as illustrated in Figure 1C. Extracellular matrix stiffness also influences mineral absorption, as previously documented (Derricks et al., 2015). Cells on softer substrates (4 kPa) demonstrate increased responsiveness to VEGF and Ca²⁺ dynamics, while cells on stiffer substrates (125 kPa) exhibit a diminished response.

The enrichment of genes related to Ribosomes suggests that protein synthesis is differentially regulated in 3D environments. This may be attributed to the unique demand for specialized proteins involved in cell-cell interactions, the integration of external signals, and mechano-transduction, especially in cells growing as spheres with low stiffness. In summary, the regulation of genes associated with Amino acid Metabolism, Mineral Absorption, and Ribosomes in 3D environments underscores the significant influence of extracellular matrix stiffness on various cellular processes. This intricate interplay allows cells to adapt to their mechanical surroundings, ultimately affecting crucial aspects of cell biology and physiology.

The 3D-repressed genes exhibit an enrichment in genes associated with Transcriptional deregulation in Cancer, the Hippo and Apelin pathways, among others. The Apelin pathway is not well-documented in breast cancer, with only one report associating high levels of Apelin with postmenopausal breast cancer (Salman et al., 2016). However, genes involved in vasodilation and muscle contraction are significantly repressed, which could be explained as an adaptation to the softer 3D environment. The suppression of these pathways could also be linked to the up-regulation of Mineral Absorption pathway, as calcium ions (Ca²⁺) can regulate the contractility of vascular smooth muscle cells (Brozovich et al., 2016)”

Pg6. Bottom of pg 6. H3K27ac and H3K18ac histone modifications are NOT epigenetic. They are unstable marks that track with gene expression. Profiling histone marks also does not show that the gene expression changes are "due" to histone modifications - it just determines if there is a correlation. The authors should be more precise in the terms they use.

R: We fully agree with the reviewer, thus, to be more precise on this point, we have changed the term "epigenetic" and replaced it with the description of the corresponding histone mark. These changes have been made on **Page 7, line 31**: “Next, we asked whether the changes in gene expression that we observed in 3D grown cells (Figure 1F) were due to changes in histone acetylation. To this end, cells grown in 2D and 3D were subjected to ChIP-seq of H3K27ac and H3K18ac, two marks strongly associated to gene expression (Ferrari et al., 2012; Wang et al., 2008).”

EV2A. The small quantitative changes in H3K27me3 for 3D down genes is unconvincing. What does a small change in H3K37me3 for down-regulated genes mean? Polycomb (H3K27me3) is digital not analogue - genes are either polycomb coated or they are not. I could not find details of the number of biological replicates performed for ChIP-seq, how replicate data were handled statistically, nor the data deposition details (Accession numbers for GEO).

R: We agree with the reviewer that this small increase of H3K27me3 close to the TSS of 3D down-regulated genes cannot be easily explained.

In the new version of the text we have discussed with caution the increase in H3K27me3 detected in 3D down genes by adding the following paragraph (**page 8, line 16**):

“Given that Polycomb operates through a digital mechanism with two opposing expression states signifying its presence or absence on target genes (Menon et al., 2021), assigning the modest increase in H3K27me3 detected in 3D down genes directly to its final impact on transcription is challenging. Instead, it is more likely a complex interplay of various histone modifications, along with other transcriptional and architectural factors, that contributes to this phenomenon”

The number of biological replicates, data points as well as the statistical tests used are explained in the new version of Extended Bioinformatics methods. The raw sequencing data from this study (ChIP-seq, ATAC-seq, RNA-seq and Hi-C) were submitted to the NCBI Gene Expression Omnibus (GEO); <http://www.ncbi.nlm.nih.gov/geo/> repository and the accession number is GSE247777.

Figure 2. MNase and ATAC-seq are mis-interpreted - they probe very different aspects of chromatin structure. MNase detects the general compaction of large-scale chromatin fibres - e.g. linker histone density or the presence of constitutive heterochromatin. ATAC-seq measures the focal loss of specific nucleosomes at sites of sequence-specific transcription factor binding; the nucleosome-depleted region at transcription start sites and at CTCF sites. As for ChIP-seq, no details are given of how many biological replicate ATAC-seq datasets were generated, how they were treated statistically, and the public Accession numbers.

R: To address the point highlighted by the reviewer, we have rephrased the text where we introduce the accessibility experiments mediated by MNase and ATAC-seq (Figure 2).

In **page 9, line 19**, in the new version of the manuscript, we add the following paragraph, “In order to assess chromatin accessibility within the cell nucleus of 2D and 3D-grown cells, we conducted a comprehensive approach that includes: i) MNase experiments, which detect the compaction of chromatin fibers at a large scale, potentially indicating changes in the presence of linker histones and heterochromatin, and ii) ATAC-seq experiments, which measure local nucleosome depletion at sites where sequence-specific transcription factors, such as CTCF, bind, as well as the nucleosome-depleted region at transcription start sites (TSS)”

In **page 10, line 11**, we added: "Hence, the chromatin structure in 3D-grown T47D cells exhibited a noticeably less accessible conformation, both at the level of larger fibers and at nucleosome resolution, in contrast to the 2D cells. These findings align with the results from gene expression analysis and imaging studies (Figure 1F and Figure 2A-C)"

In the Discussion section (**page 19, line 22**): "MNase and ATAC-cleavage experiments provided confirmation that chromatin in 3D cultured cells is characterized by increased compaction, affecting both the larger chromatin fibers and the localized regions where transcription factors interact with nucleosomes. This heightened compaction results in reduced accessibility (Figure 2B-C)"

All ChIP-seq, RNA-seq, and ATAC-seq as well as Hi-C experiments were conducted in duplicate for each target protein (ChIP) and experimental condition (2D and 3D). The bioinformatics protocols employed to process and statistically analyze the data are detailed in the Extended Bioinformatics Methods section. All data presented in this study has been deposited at the GEO repository (accession number: GSE247777).

Figure 3 and pg 9. The authors confuse correlation and causation in considering Hi-C compartments, stating that "These findings suggest that in 3D condition there are gradual transitions between the A and B compartments that sustain the changes detected in gene expression". It is gene expression and chromatin states such as H3K27me3 and H3K9me3 domains that determine A and B compartments, not the other way around.

R: We agree with the reviewer and we changed the paragraph in **page 11, line 14** to:

"These findings suggest that in 3D condition the changes detected in gene expression sustain gradual transitions between the A and B compartments"

Figure 4. 75% loss CTCF binding across the genome is implausible unless there is a very widespread increase in DNA methylation to block CTCF binding, or as the authors go on to suggest, massive post-translational modification of CTCF that prevents its binding. The authors should have used a spike in control in their ChIP-sequencing to be confident of reduced CTCF binding genome-wide. Otherwise, this result could simply reflect technical issues - e.g. with chromatin solubility, cross-linking or fragmentation. Based on the data in Figs 5 and 6, the authors should show whether the ATAC-seq peaks that correspond to the nucleosome depleted region underneath the sites of bound CTCF are particular affected.

-Regarding the point raised by the reviewer: "whether the ATAC-seq peaks that correspond to the nucleosome depleted region underneath the sites of bound CTCF are particular affected"

R: As recommended by the reviewer, we conducted a more in-depth analysis of the ATAC-seq assays. We identified the regions that become more accessible in 2D and 3D as well as the DNA binding motifs (new EV4A-B). Our results suggest that the sites that become more accessible in 2D and 3D are enriched in CTCF, TEAD, FOXA1 and PR motifs (new EV4B). However, when we cross-checked our data with ChIP-seq database (Zheng et al., 2019) (new EV4B), our word cloud plots (new EV4B-C, lower panels) give different results: the sites that open in 3D are enriched in the estrogen pathway through the estrogen receptor (ER) itself, as well as its pioneer factors FOXA1 and GATA3; the cofactor GREB1 and the coactivator P300, which could explain the increased proliferation observed in 3D cells (Figure 1C). In contrast, the sites that become closed in 3D (2D-exclusive) are enriched in CTCF, the architectural factor that we have shown to be displaced when we grow the cells as spheroids (new Figure 4A). This would imply that in 3D a more estrogenic program is turned on and CTCF is displaced, whether these two events are connected require further investigation.

In the new version of the manuscript, we add the following text in **page 10, line 15**:

"To characterize the regions that become more accessible in 2D and 3D we conducted a more in-depth analysis of the ATAC-seq (EV4A-B). Our results suggest that the sites that become more accessible in 2D and 3D are enriched in CTCF, TEAD, FOXA1 and PR motifs (EV4B). However, when we combined our analysis with ChIP-seq data (Zheng et al., 2019), word cloud plots (EV4B-C, lower panels) indicated that open sites in 3D are enriched in the estrogen pathway through the ER itself, alongside pioneer factors FOXA1 and GATA3; the cofactor GREB1 and the coactivator P300, which could explain the increased proliferation observed in 3D cells (Figure 1C). In contrast, the sites that become less accessible in 3D (2D-exclusive) are enriched in CTCF, an architectural factor that we have shown to be displaced when cells were grown as spheroids (Figure 4A). This would imply that in 3D cells a more estrogenic program is turned on along CTCF is displaced, if these two events are connected require further investigation."

Figure 5. The authors present data in Fig 5A suggesting that all CTCF in the 3D cells is phosphorylated. There is no control for a protein that they do not expect to be phosphorylated by LATS1. CTCF-ChIP-seq +/-the LATS inhibitor should similarly be controlled with a spike in control (such as mouse chromatin). This is important given that the authors state that the effect of a LATS1 inhibitor on CTCF phosphorylation could

not be detected. A massive loss of CTCF across the genome, as a consequence of LATS1-mediated phosphorylation, seems at odds with the site-selective loss of CTCF as a result of LATS-mediated phosphorylation, reported by Luo et al., 2020. This aspect of the manuscript is under-developed - e.g. the authors should transfect into their cells CTCF with mutations at T374 and S402 that block this phosphorylation. This should restore binding and 3D genome organisation.

-Regarding the reviewer comment "There is no control for a protein that they do not expect to be phosphorylated by LATS1. "

R: *In the new version of the manuscript we incorporated a new Phos-tag gel including BRG1 as loading control (new Figure 5A).*

-Regarding the reviewer comment "CTCF-ChIP-seq +/-the LATS inhibitor should similarly be controlled with a spike in control (such as mouse chromatin).

R: *We appreciate the reviewer's recommendation to include spike-in controls in our CTCF ChIP-seq experiments. Thus, we conducted these experiments -including biological replicas- in 2D, 3D, and 3D with the TRULI inhibitor, along with Drosophila spike-in controls (Active Motif). Our results, after spike-in correction, confirmed our previous findings. Notably, the shift from 2D to 3D occupancy, which was initially 75%, was reduced to 40% (new Appendix Figure S7).*

These changes can be attributed to the incorporation of spike-in controls. However, it is important to note that they could also result from adjustments made in the washing steps and in the DNA purification method. These adaptations became necessary as the ChIP-IT kit (Active Motif) initially employed in our experiments was no longer available due to budget constraints. In this context, prior studies have reported the influence of wash conditions on transcription factor binding (Kidder et al., Nat Immunol 12, 918–922, 2011). The average of all three CTCF ChIP-seq replicates obtained from 2D and 3D cells is depicted in new Appendix Figure S7D. Moreover, in spike-in-controlled CTCF ChIP-seq experiments conducted in the presence of the LATS inhibitor TRULI, CTCF binding was restored (Appendix Figure S7C).

*We've incorporated the new results as new Appendix Figure S7A-D, and discussed in **page 15, line 17**. See below:*

"The sensitivity of CTCF ChIP-seq experiments is notably influenced by the particular protocol employed. This results in varying outcomes regarding the detected 3D shift. Notably, using a stringent ChIP protocol, which includes multiple washes and the use of DNA purification columns (ChIP-IT, Active Motif), revealed a 75% displacement of CTCF when cells were cultured in a 3D environment, as depicted in Figure 4A.

In contrast, using a standard protocol with gentler wash steps, as outlined by Vicent et al., 2014 (Vicent et al., 2014), and incorporating an internal Drosophila spike-in control, reduced this displacement to 40% (Appendix Figure S7A-C). Notably, this reduction remained statistically significant when compared to cells cultured in a 2D monolayer setting. Furthermore, in spike-in-controlled CTCF ChIP-seq experiments conducted in the presence of the LATS inhibitor TRULI, CTCF binding was restored (refer to Appendix Figure S7C). These findings align with our earlier results; the average of all three CTCF ChIP-seq replicates obtained from 2D and 3D cells is depicted in Figure S7D"

-Regarding the reviewer comment "the authors should transfect into their cells CTCF with mutations at T374 and S402 that block this phosphorylation. This should restore binding and 3D genome organisation"

R: *To address the point highlighted by the reviewer, we transfected T47D cells with two CTCF constructs: the wild-type and the phospho-mimetic variant, which carries two mutations at T374E/S402E, targeting two LATS-dependent phosphorylation sites (Luo et al., Science Advances 2020). We assessed their ability to bind to the chromatin fraction. Constitutive phosphorylation at T374 and S402 resulted in a reduced CTCF binding compared to the wild-type CTCF, mirroring the situation in 3D (see Figure below and new EV5B).*

In addition, to incorporate a formal proof that LATS1 phosphorylates CTCF in 3D cells, we performed immunoprecipitation of CTCF in 3D-grown cells treated or not with TRULI and then probed with specific antibodies recognizing the phosphorylated RXXpS/T residues that match the changes observed in T374 and S402 as previously described (Luo et al., 2020). Our results showed that the phospho-CTCF signal in 3D is reduced by 40% in the presence of TRULI, confirming the raised hypothesis (see Figure below and EV5A).

It has been reported that the overexpression of CTCF impedes the proliferation and metastasis of breast cancer cells by deactivating the nuclear factor-kappaB pathway in breast cancer cells (Wu et al., Oncotarget 2017; PMID: 29212169). To elucidate the function T474 and S402, we conducted a functional assay by evaluating cell growth in T47D cells expressing both wild-type (WT) and a phospho-mimetic variant of CTCF T374E/S402E (see new EV5C and Figure below, left panels). The overexpression of WT CTCF in T47D cells not only inhibited growth but also triggered cell death, compared to the control transfection (new EV5C and Figure below). Conversely, in the T374E/S402E mutant in which chromatin binding is compromised (see EV5B and Figure below, right panel), this effect was not appreciated (compare WT vs. mutant in new EV5C). This suggests that LATS1-dependent phosphorylation of CTCF induced its displacement and participates in 3D cell growth, as demonstrated in Figures 1C and Appendix S8A-B.

These new results, are discussed in **page 14, line 16** in the new version of the manuscript

“To confirm that T374 and S402 of CTCF are the phosphorylation targets of LATS, and to assess their impact on chromatin binding in our experimental system, we transfected T47D cells with both wild-type and the phospho-mimetic T374E/S402E CTCF variant. Subsequently, we evaluated their chromatin binding abilities. The T374E/S402E mutant displayed a diminished capacity to bind to chromatin when compared to the wild-type CTCF (EV5B). These results confirm previous studies (Luo et al., 2020) and provide more precise insights into the LATS target sites.

It has been reported that the overexpression of CTCF impedes the proliferation and metastasis of breast cancer cells by deactivating the nuclear factor-kappaB pathway in breast cancer cells (Wu et al., 2017). To elucidate the function T474 and S402, we conducted a functional assay by evaluating cell growth in T47D cells expressing both wild-type (WT) and a phospho-mimetic variant of CTCF T374E/S402E (EV5C). The overexpression of WT CTCF in T47D cells not only inhibited growth but also triggered cell death, compared to the control transfection (EV5C). Conversely, in the T374E/S402E mutant in which chromatin binding is

compromised (see EV5B), this effect was not appreciated (compare WT vs. mutant in EV5C). This suggests that LATS1-dependent phosphorylation of CTCF induced its displacement and participates in 3D cell growth, as demonstrated in Figures 1C and Appendix S8A-B.”

And in **page 14, line 10**:

“To have a formal proof that LATS1 phosphorylates CTCF in 3D cells, we performed immunoprecipitation of CTCF in 3D-grown cells treated or not with TRULI and then probed with specific antibodies recognizing the phosphorylated RXXpS/T residues that match the changes observed in T374 and S402 as previously described (Luo et al., 2020). Our results showed that the phospho-CTCF signal in 3D is reduced by 40% in the presence of TRULI, confirming the raised hypothesis (EV5A)”

Dear Dr Vincent,

Thank you again for re-submitting the revised version of your manuscript "LATS1 controls CTCF chromatin occupancy and hormonal response of 3D-grown breast cancer cells" which was previously rejected after revision to the EMBO Journal. I have shared the revised manuscript with the original referees. Both referees #1 and #2 think that their comments have been addressed. Unfortunately, referee #3 which was more critical and raised several important concerns was not able to re-review the manuscript. We have therefore consulted with an external expert regarding your responses to the comments by referee #3. This advisor thinks that many concerns have been addressed. However, this advisor also remarks that the concerns raised by referee #3 regarding the findings reported in figure 5 are important and not addressed sufficiently:

"A massive loss of CTCF across the genome, because of LATS1-mediated phosphorylation, seems at odds with the site-selective loss of CTCF as a result of LATS-mediated phosphorylation, reported by Luo et al., 2020. This aspect of the manuscript is under-developed - e.g. the authors should transfect into their cells CTCF with mutations at T374 and S402 that block this phosphorylation. This should restore binding and 3D genome organisation. Here the referee is asking for a CTCF mutant which cannot be phosphorylated and overcome the effect of LATS-mediated phosphorylation, typically a CTCF T374A, S402A double mutant. Such a mutant is predicted to restore CTCF binding and 3D genome organisation, in cells lacking LATS1 or treated with an inhibitor of LATS. The mutant should also overcome the inhibition of gene expression and 3D growth. The authors have used a CTCF T374E, S402E mutant, which previously have been shown by Luo et al not to bind DNA. In my opinion, they should provide the experiment or at least test their model by performing the experiment. The experiment would also help addressing comment 5 by referees #1 more directly. "

We agree with the advisor that the response given here does not directly address the concern. We also think that this is an important point as the causal relationship between the 3D genome architecture and the site-specific phosphorylation of CTCF by LATS1 is a central finding of the manuscript (indeed constitutes the title of the manuscript). I would also like to remind you that as a matter of policy we do only allow a single round of major revisions as we do not want a protracted and frustrating review process with uncertain outcome.

Therefore, while we appreciate the considerable effort that has been undertaken, we cannot offer publication at the EMBO Journal under these circumstances. I am sorry that I cannot share better news here. I would like to repeat our transfer offer to EMBO Reports. Please contact my colleague Esther Schnapp (e.schnapp@emboreports.org) if you have any questions regarding the transfer.

Thank you in any case for the opportunity to consider this manuscript. I am sorry we cannot be more positive on this occasion, but we hope nevertheless that you will find our transfer suggestions helpful.

Best regards,

Cornelius Schneider

Cornelius Schneider, PhD
Editor | The EMBO Journal
c.schneider@embojournal.org

** As a service to authors, EMBO Press provides authors with the possibility to transfer a manuscript that one journal cannot offer to publish to another EMBO publication or the open access journal Life Science Alliance launched in partnership between EMBO Press, Rockefeller University Press and Cold Spring Harbor Laboratory Press. The full manuscript and if applicable, reviewers' reports, are automatically sent to the receiving journal to allow for fast handling and a prompt decision on your manuscript. For more details of this service, and to transfer your manuscript please click on Link Not Available. **
Please do not share this URL as it will give anyone who clicks it access to your account.

Dear Dr Schneider,

Thank you for your email with your comments on our EMBOJ-2023-114519R2-Q manuscript. We are very frustrated with your decision not to accept our manuscript, after all the effort and resources we have put into addressing the concerns of the 3 reviewers.

Both reviewer #3 and the external advisor propose the use of the phospho-dead T374A/S402A mutant to validate the importance of these sites in their binding to chromatin. Please, let me pinpoint our response to this specific issue raised by the reviewer:

1) We decided to address the response to reviewer #3 in a 2D system due to *the impossibility of carrying out the 3D transfection experiment*. The 3D spheroid transfection is a challenge that has not been successfully met to date. There is some optimism with mRNA (Ushida et al Micromachines 2020) but for DNA there is no solid and consensual protocol. For this and other projects we have tried electroporation and different transfection reagents such as the one from OZBiosciences but DNA entry is hampered by the matrigel and the little that enters does so in a very heterogeneous manner. For this reason, we decided to address the response to reviewer #3 in 2D cells.

2) *Inconsistency of targeting LATS1 in 2D cells*: Despite having the availability of a CTCF phospho-dead mutant (T374N/S402N), that was kindly provided Dr. Jianrong Lu from the Florida University (USA) (see below his e-mail from 28/07/2023), we have not used it. This is because in the 2D condition, the only condition where transfections proposed by the reviewer can be done, CTCF is already bound to its sites in the genome (Figures 4A and 5D) and p-LATS1 is very little or not activated (compare pYAP signal in Figure 1D and lane 2 in the new Figure for the Editor). Therefore, the use of LATS1 inhibitors or knockdowns in this condition does not make sense.

3) Given the impossibility of a transfectable cell system where LATS was activated, we chose to activate constitutively CTCF with the phospho-mimetic (new Figure EV5B). Our results showed that when T374 and S402 are phosphorylated CTCF binding to chromatin is significantly compromised (new Figure EV5B)

4) Although we believe that due to the technical limitations described above the experiment cannot be carried out, one possibility to address this point would be to treat T47D 2D cells with the Hippo regulator Latrunculin B (LatB) that increases the nuclear phosphorylation of LATS1 (see new Figure for the Editor) and test in that condition the binding of CTCF wild type and the phospho-dead mutant.

I hope you will take into account the effort and resources we have put in to have a complete and solid story and give us the opportunity to either discussed the limitation of using the CTCF phospho-dead mutant and/or to perform the LatB experiment. Therefore, we can finish this story that, in our opinion, deserves to be published in EMBO J.

I take this opportunity to wish you a Merry Christmas

Best regards

Guillermo P. Vicent

Dear Dr. Vicent,

Thank you for submitting your manuscript for consideration by the EMBO Journal.

As mentioned in my previous email I would like to invite you to prepare a revised manuscript based on the arguments detailed in your appeal letter and the proposed experiment in point 4.

When preparing your letter of response, please bear in mind that this will form part of the Review Process File, and will therefore be available online to the community. For more details on our Transparent Editorial Process, please visit our website:
<https://www.embopress.org/page/journal/14602075/authorguide#transparentprocess>

We generally allow three months as standard revision time. As a matter of policy, competing manuscripts published during this period will not negatively impact on our assessment of the conceptual advance presented by your study. However, we request that you contact the editor as soon as possible upon publication of any related work, to discuss how to proceed.

Thank you for the opportunity to consider your work for publication. I look forward to your revision.

Yours sincerely,

Cornelius Schneider

Cornelius Schneider, PhD
Editor
The EMBO Journal
c.schneider@embojournal.org

- a point-by-point response to the referees' comments, with a detailed description of the changes made (as a word file).
- a word file of the manuscript text.

- individual production quality figure files (one file per figure)

- a complete author checklist, which you can download from our author guidelines

(<https://www.embopress.org/page/journal/14602075/authorguide>).

- Expanded View files (replacing Supplementary Information)

We realize that it is difficult to revise to a specific deadline. In the interest of protecting the conceptual advance provided by the work, we recommend a revision within 3 months (16th Apr 2024). Please discuss the revision progress ahead of this time with the editor if you require more time to complete the revisions. Use the link below to submit your revision:

All editorial and formatting issues were resolved by the authors.

Dear Dr. Vicent,

I am pleased to inform you that your manuscript has been accepted for publication in the EMBO Journal.

Yours sincerely,

Cornelius Schneider, PhD
Editor
The EMBO Journal
c.schneider@embojournal.org
